



# Using OMPS-LP color ratio to extract stratospheric aerosol particle median radius and concentration with application to two volcanic eruptions

Yi Wang[1], Mark Schoeberl[1], Ghassan Taha[2]

[1]Science and Technology Corporation (STC), Columbia, MD, United States
[2]Morgan State University, Baltimore, MD, United States

*Correspondence to*: Yi Wang (yiwang@stcnet.com)

**Abstract.** We derive stratospheric aerosol microphysical parameters from Ozone Mapping Profiler Suite Limb Profiler (OMPS-LP) satellite measurements using aerosol extinction coefficient ratios at two wavelengths (the color ratio), which is sensitive to the particle radius, and concentration. We estimate various sources of uncertainty in this technique including extinction coefficient measurement error, sensitivity to the size distribution width assumption, and the OMPS-LP algorithm phase function error. We apply our algorithm to extinction coefficient measurements made by the Stratospheric Aerosol and Gas Experiment on the International Space Station (SAGE III/ISS) to verify our approach and find that our results are in good agreement. Our results also compare favorably to balloon borne particle size measurements and concentrations under ambient condition and 2019 Raikoke volcanic eruption assuming a log-normal particle size distribution width of 1.6. We also estimate the changes in aerosol median radius and concentration following the 2019 Raikoke and 2022 Hunga Tonga-Hunga Ha'apai volcanic eruptions and the result is consistent with other retrievals published in the literature.

## 1 Introduction

Stratospheric aerosols are of significant interest to the scientific community because of their impact on the climate through volcanic and pyroCb injections (e.g. Turco et al, 1982; Robock, 2000; Kremser et al., 2016; Fujiwara et al., 2020), and because of the possibility of altering their abundance for geoengineering purposes (Robock, 2014; NASEM, 2021). Stratospheric aerosols mostly consist of small sulfuric acid droplets that form in the high relative humidity lower stratosphere after the oxidation of sulfur containing trace gases (Seinfeld and Pandis, 2016). Aerosol particle sizes typically range from 0.01 to 1 µm radius. Stratospheric aerosols may also include volcanic ash (Vernier et al., 2016; Muser et al., 2020), meteoric smoke (Schneider et al., 2021; James et al. 2023) and wildfire smoke (Khaykin et al., 2020; Peterson et al., 2021). Larger ash and meteoric fragments ($> 1 - 10$ µm) settle out of the stratosphere within a few days, but smaller smoke and volcanic ash ($< 0.1$ µm) can provide condensation nuclei for sulfuric acid droplets (Mossop, 1963; 1964, Rosen et al., 1978). These result in black and brown aerosols that have been observed to 'self loft' through solar heating (Yu et al., 2019, Khaykin et al., 2020; Khaykin et al., 2022).



Stratospheric aerosol passive remote sensing began with simple solar occultation systems. Rosen et al. (1969) demonstrated the application of solar occultation methods using a balloon-system which was the precedent for the first satellite measurements of the stratospheric aerosol layer (Stratospheric Aerosol Measurement (SAM), SAM II, and Stratospheric Aerosol and Gas Experiment (SAGE) series, McCormick et al., 1979, 1982). In the SAGE III instrument, the aerosol extinction coefficient is

retrieved as a residual after removing of the ozone and other species at nine wavelengths, without any prior assumptions of the aerosol particle size (Thomason and Taha, 2003). Multi-wavelength measurements of SAGE III are often used to infer information of the particle size (Thomason et al., 2021; Wrana et al., 2021). The occultation technique has been extended to include stellar occultation (e. g. Global Ozone Monitoring by Occultation of Stars (GOMOS), Vanhellemont et al., 2016).

Newer approaches to satellite stratospheric aerosol measurements include lidar methods (e.g. Cloud-Aerosol LiDAR and Infrared Pathfinder Satellite Observations (CALIPSO), Vernier et al., 2009), and limb scattering methods used by the Scanning Imaging Absorption Spectrometer for Atmospheric Chartography (SCIAMACHY) (Malinina et al., 2018), the Optical Spectrograph and InfraRed Imaging System (OSIRIS) (Llewellyn et al., 2004) and the Ozone Mapping Profiler Suite Limb Profiler (OMPS-LP) on the Suomi National Polar-orbiting Partnership (S-NPP) (Flynn et al. 2014, Jaross et al., 2014). The

limb scattering method provides improved global coverage because it measures scattered light profiles along the atmospheric limb. For example, SAGE III on the International Space Station (SAGE III/ISS) requires about a month or more of occultations to provide near-global coverage, whereas OMPS-LP can provide equivalent coverage in a single day. However, unlike the solar osculation, limb scattering retrievals often requires a forward model calculation and aerosol particle assumptions for the aerosol retrieval. (Taha et al, 2021).


Exploiting OMPS-LP multiwavelength measurements to derive aerosol particle size information is important because there is no currently planned solar occultation mission beyond SAGE III/ISS, and the community will need to rely on OMPS-LP measurements to characterize changes in stratospheric aerosols. Despite the coverage advantages, limb scattering radiances are more complicated to interpret and have higher uncertainty due to variations in the scattering angle with orbit, variation in

scattering properties with particle size, and measurement contamination by surface reflected light. Yet, despite these drawbacks, the limb scattering techniques have provided new insights into the rapid evolution of stratospheric aerosols immediately following volcanic eruptions and PyroCB events (Taha et al., 2022, Khaykin et al., 2022, Wells et al., 2022, Gorkavyi et al., 2021).

The goal of this paper is to derive aerosol particle median radius and concentration from OMPS-LP aerosol extinction measurements. We use the color ratio approach, which relies on using the extinction measurements at two wavelengths. This approach has used by Bourassa et al. (2008b) with OSIRIS measurements although the idea dates to even earlier work by Yue and Deepak (1983). Our algorithm is similar to the color ratio method developed by Thomason and Vernier's (2013) for



SAGE II that has been also used for cloud identification in SAGE III/ISS data (Schoeberl et al., 2021; Kovilakam et al., 2023).
In the next section, we detail the algorithm and assess the uncertainties. We then check the algorithm by applying it to SAGE III/ISS data and validate the algorithm by comparing our results to *in situ* balloon observations. Finally, we apply the algorithm to the OMPS-LP measured background stratospheric aerosol distribution and two recent volcanic eruptions where aerosol extinction coefficients were anomalous.

## 2 Algorithm and uncertainty analysis

### 2.1 Aerosol particle radius and concentration retrieval algorithm

Our goal is to determine the aerosol particle median radius and concentration from extinction coefficient measurements. Our algorithm uses the aerosol extinction coefficient ($E$) from the L2 OMPS-LP products, V2.1 (Taha et al., 2021), which provides $E$ at multiple wavelengths. The $E$ at wavelength $\lambda$, is proportional to the scattering coefficient (1a)

$$E_\lambda = \int_0^\infty Q(r,\lambda)\pi r^2 PSD(r)\, dr \qquad (1a)$$

where $Q$ is the Mie scattering efficiency (equal to the extinction efficiency for non-absorbing particles), $r$ is the particle radius, and $PSD(r)$ is the particle size distribution, which is usually assumed to be a single mode log-normal distribution (Eq.3) where $N$, $r_0$ and $s$ are also defined. Eq. (1) can be approximated as the equivalent cross section ($\sigma$) times $N$.

$$E_\lambda \sim \sigma_\lambda(r_o, s, \theta) \cdot N \qquad (1b)$$

For a limb scattering measurements, the Mie scattering phase function $p(r,\lambda,\theta)$ where $\theta$ is the scattering angle is included in
the equivalent cross section (Bourassa et al., 2008a, b). For occultation measurements $p = 1$ and Eq. (1a) is the same as Wrana et al. (2021) Eq. 2. In this study, the color ratio (CR) is defined as the ratio between two extinction coefficient values.

$$CR = E_{\lambda 1}/E_{\lambda 2} \qquad (2)$$

For the CR, Eq. 1b shows that $N$ cancels out, and the CR is proportional to the Ångstrom exponent which is a function of the median radius. The Eq. 1b approximation works for both limb scattering and occultation measurements. In the OMPS-LP
V2.1 L2 algorithm, aerosol extinction coefficient is retrieved independently at six wavelength, 510, 600, 675, 745, 869, 997 nm (Taha et al., 2021) using the Chahine nonlinear relaxation method (Chahine, 1970) and the Gauss–Seidel limb scattering (GSLS) radiative transfer model (Loughman et al., 2015). More precisely, the current OMPS-LP aerosol algorithm computes the extinction coefficient at each OMPS measured wavelength assuming a fixed aerosol size distribution. A radiative transfer model then computes the scattered radiance as a first guess assuming background particle size distribution. The algorithm then
iterates the aerosol extinction coefficient using the Chahine (1970) method until the computed radiance nearly matches the observed radiance (Chen et al., 2018). The OMPS-LP aerosol extinction coefficient products have a vertical resolution of 1.6-1.8 km, reported every 1 km. Only data from the OMPS-LP central slit profiler is used in this study. It is important to note that the L2 V2.1 does not recompute median radius and number density for each iteration, instead it directly computes aerosol



extinction coefficient, ensuring a match with the observed radiance (Taha et al., 2021; Chen et al., 2018; Loughman et al.,
95    2018).

To demonstrate how using the CR provides information on the aerosol concentration and radius, we start with a series of
idealized extinction coefficient calculations. For these calculations, we simulate scattering process using the SASKTRAN
radiative transfer model (Bourassa et al., 2008a; Zawada et al., 2015) which includes Mie code that assumes a log-normal
aerosol size distribution with a width of 1.6 for stratospheric sulfate aerosols. This size distribution is consistent with *in situ*
stratospheric aerosol measurements (Deshler et al. 2003; Bourassa et al. 2008b) and will be discussed further below.

In Fig. 1 we show the how CR (510nm/869nm) and CR (745nm/869nm) varies for different aerosol particle sizes using
SASKTRAN's Mie code. We chose 869 rather than the 997 nm aerosol extinction coefficient for $\lambda_2$ because it is more accurate
and available for the whole mission (Taha et al., 2021). Fig. 1 shows that although both CR values decrease with increasing
particle size, at about 0.4 µm the CR asymptotes to one. This CR – median radius relationship allows us to infer the aerosol
particle radius up to ~0.4 µm. In general, the figure shows that the CR is more sensitive to the size if $\lambda_1$ and $\lambda_2$ are far apart;
however, if $\lambda_1$ is too short (e.g. < ~400 nm) Rayleigh scatter overwhelms the aerosol scattering signal at lower altitudes, and
the aerosol extinction coefficient can't be accurately measured. By calculating the aerosol extinction coefficient equivalent
cross-sections, and using the measured extinction coefficient ($E$) at one of the wavelengths (e.g., either 869 nm or 510nm), Eq.
(1b) can then be used to compute the aerosol number density. This method produces both consistent number density and
particle size and was used by Bourassa et al. (2008b).

Clouds produce very high extinction coefficients and often contaminate limb aerosol measurements. To reduce possible cloud
contamination, clouds are removed from the extinction profile before retrieval. The OMPS-LP algorithm retrieves the cloud
top height using a radiance gradient scheme (Chen et al., 2016).  Large aerosol particles with CR values of ~1 are uncommon
in the stratosphere because these particles settle out within days. More common is low-extinction sub-visible cirrus (SVC)
seen near the tropopause (Schoeberl et al., 2021). We found that selecting CR limit of < 1.1 is fairly effective at filtering out
cirrus that may appear above the reported cloud top height. Choosing 1.1 as the threshold is based on the SASKTRAN
simulations with aerosol model and ice particle model (Baum et al, 2014). At the same time, we restrict the retrieved aerosol
median radius to < 0.3 µm. Figure 2 displays a cross-sectional CR profile for a single OMPS-LP orbit on 09/13/2020. The
magenta line in the figure is the cloud top height from the OMPS-LP file. The figure shows that CR of ~ 1.1 is close to the
retrieved cloud top height as expected, and thus is a valid threshold for cloud screening.

There are at least three possible sources of uncertainty in our calculations.  The first source of uncertainty comes from OMPS-
LP extinction coefficient error which produces uncertainty in the CR. The second source of uncertainty comes from the
assumption that our size computation is independent of the assumed width of the aerosol size distribution.  The third source of





uncertainty is the variation in aerosol scattering phase function with particle radius. In the OMPS-LP V2.1 algorithm, a fixed phase function is assumed in the radiative transfer model (RTM) computation of the radiance. In our algorithm, size can vary,
and changes in the particle size may thus be inconsistent with the RTM phase function and this can be a source of error. In the next two sections we will assess these potential sources of uncertainty and an approach to estimating them.

**2.2 Color ratio and aerosol particle median radius uncertainties**

In the context of our retrieval method, assuming a fixed unimodal log-normal distribution width is a necessary step and a common approach used in current aerosol retrieval algorithms. The log-normal distribution (Chen et al., 2018, Eq. 4) is

$$n(r) = \frac{N}{r\sqrt{2\pi}\ln(s)} \exp\left(-\frac{1}{2}\left[\frac{\ln\left(\frac{r}{r_m}\right)}{\ln(s)}\right]^2\right) \tag{3}$$

where $r$ is the particle radius (μm), $n(r)$ is the number distribution, $r_m$ is the median radius, and $s$ is the standard deviation or width. We assumed the distribution width ($s$) is 1.6 for the aerosol model, which is considered reasonable for an unperturbed or background stratospheric aerosol retrievals (Bourassa et al., 2012, Rieger et al., 2018, Wrana et al., 2021). More discussions about the appropriate value of the distribution width compared to *in situ* balloon measurements are displayed at Section 3.1.
We have performed a series of experiments using the SASKTRAN Mie code to determine the sensitivity of the size to both color ratio errors and assumed distribution width.

Figure 3 shows how the median radius varies with color ratio and width (s) of the log-normal distribution. Given a color ratio (CR) of 3, for example, the median radius estimates vary from 0.05 to 0.3μm over a distribution width from 1.1 to 1.8. Given
a measured extinction coefficient, this radius range will produce a large change in the estimated aerosol concentration. However, Wrana et al. (2021) used a third extinction wavelength from SAGE III/ISS data to estimate the log-normal distribution width and found that most of the observations clustered between s = 1.4 and 1.6. This information also constrains our size distribution uncertainty.

We first define the average median radius uncertainty ($U_r$) as the standard deviation of particle median radii within the uncertainty domain of both CR and distribution width. The uncertainty in CR, $U_{CR}$, can be estimated from Taha et al. (2021, Fig. 6) comparison to SAGE III/ISS. The geometric sum of the uncertainty of each extinction components:

$$U_{CR} = \sqrt{U_{E\,\lambda1}^2 + U_{E\,\lambda2}^2} = \sqrt{\left(\frac{\delta_{\lambda1}}{E_{\lambda1}}\right)^2 + \left(\frac{\delta_{\lambda2}}{E_{\lambda2}}\right)^2} \tag{4}$$

For example, the extinction uncertainty, $U_{E\,869}$, at 20 km is about 5% and $U_{E\,510}$ is about 20% - both at the equator. The
uncertainty in the color ratio is then 21%. For the uncertainty in distribution width ($s$), we use fine mode parameters shown in Rieger et al. (2018, Fig. 6) which gives a width uncertainty of ~0.2, consistent with the Wrana et al. (2021) estimate. To estimate the median radius uncertainty, we compute the standard deviation for all the points within the domain $s\pm0.2$ and CR





±21% using Fig. 3. Using the uncertainty measurements for $s$ and CR, we can now vary $s$ and CR values to see how the $U$

varies. The results are shown in Fig. 4 with contours of median radius overlaid. Points that lie outside of the averaging domain

shown in Fig. 3 are not included. Fig. 4 shows that the radius uncertainty is largest for particles < 0.05μm and with common

distribution widths ($s$=1.4-1.6).

Figures 3 and 4 quantify the expected impact of CR uncertainty and distribution width uncertainty on radius using values from

Taha et al. (2021) and Rieger et al (2018). We can further quantify the uncertainty in the median radius using the color ratio

(510 nm/869 nm) uncertainty for ranges from 18 to 28 km (510 nm extinction coefficient uncertainty up to 10%-50% and 869

nm extinction coefficient uncertainty of 5%-20%) and the log-normal distribution width ($s$) uncertainty from 0.2 to 0.4. Then

for a width value of 1.6, for example, and averaging color ratios are between 2 and 4, our average median radius uncertainty

is ~20%. Using (1b) and assuming that effective cross section is proportional to the square of the mode radius leads to a number

density uncertainty of 44%. This example shows how the radius and number density uncertainty due to the distribution width

can be quantified.

Measurements following PyroCB or volcanic events show that the size distribution is not static, and our algorithm will have

higher uncertainty under those conditions. Fig. 11 (below) shows that some of the aerosol plumes are characterized by lower

CRs (between 1 and 2), but most of the aerosols still have CR values between 2 and 4. Figure 4 shows that the uncertainty in

this domain region is 20-30%.

To summarize, Figs. 3, 4 show that the distribution width error is largest for very small $r_m$ and larger values of $s$. An error in

the median radius creates a larger (square) error in the cross-section, and from (1b) a larger uncertainty in the concentration.

**2.3 The aerosol phase function uncertainty**

As stated above, to retrieve the aerosol extinction coefficient from limb scattering measurements, a fixed Mie scattering phase

function is assumed using a background aerosol model. This assumption can introduce uncertainties in retrieved particle

median radius when the particle radius differs from the background. To investigate the OMPS-LP aerosol color ratio sensitivity

to the algorithm's assumed phase function (Figure A1), we perturb the phase function parameters by ± 10% and run the OMPS-

LP retrieval algorithm at a range of scattering angles observed during a single orbit. This approach is similar to that used by

Chen et al. (2018). OMPS-LP aerosol retrieval algorithm assumes a gamma aerosol size distribution using fitted parameters

of $\alpha = 1.8$ and $\beta = 20.5$ where effective radius $r_{eff} = 0.185$μm using Chen et al., (2018).

$$r_{eff} = \frac{(\alpha + 2)}{\beta} \tag{5}$$

For an assumed distribution width, $s$=1.6, we can get a modal radius $r_m = 0.1$μm using,





$$r_m = r_{eff} \exp\left(-\frac{5}{2} \ln^2 s\right) \tag{6}$$

Changing $\alpha$ and $\beta$ effectively lead to changes in the median radius and the distribution width. Figure 5a shows that the phase function is more sensitive to $\beta$ changes than $\alpha$. A 10 % change of $\beta$ can produce $\pm$ 10 and 15% change in 510 nm and 869 nm phase functions, respectively, while a 10 % change of $\alpha$ results in a $\pm$ 3% to 5% change for both wavelengths. Chen et al. (2018) noted that increasing $\alpha$ increases the peak of the differential size distribution while increasing $\beta$ shifts the peak distribution to a larger particle radius. Figure 5b shows that the phase function perturbations produce anti-correlated but lesser

changes in aerosol extinction coefficient. The changes caused by $\beta$ perturbations are mostly within 5% and 3% for $\alpha$ for both wavelengths. The structural change in the extinction coefficient along the orbit is caused by scene reflectivity (Chen et al., 2018). The effect on CR (510/869) is shown in Fig. 5c for the same variations of phase functions. The CR perturbations are mostly within 3% between scattering angles of 65-125° and 5% outside that range, except for the very small scattering angles, which might be caused by scene reflectivity changes. For a typical color ratio of 3 with $s$ = 1.6 a 3% to 5% change in color

ratio perturbations lead to < 4% and < 7% difference in retrieved aerosol particle radius, respectively.

As noted above, the phase function error can be even larger for pyroCb or volcanic plumes. For those situations larger particles are more likely to occur. Changing $\beta$ to 11.0 gives $r_m$ = 0.2 µm (using Eq. 6) which results in 40% phase function modification (Figure 6). The resulting CR and $r_m$ uncertainty increase to $\pm$ 11-20% and $\pm$12-32% respectively for a typical CR of 3. Under

volcano eruptions, if we assume that the error in CR is a combined ~30% from $U_r$ then for a retrieved aerosol radius of $r_m$ = 0.2 µm, we get an uncertainty range of 0.15- 0.3 µm. The uncertainty is even larger for large scattering angles in the SH, which is caused by the reduced sensitivity of the OMPS-LP 510 nm measurements at these angles (Taha et al., 2021).

### 3 Verification and Validation

In this section we verify our OMPS-LP aerosol retrieved radius and number density algorithm using SAGE III/ISS

measurements using the same algorithm. We validate our results with *in situ* balloon measurements.

### 3.1 Comparison of OMPS-LP retrievals with SAGE III/ISS retrievals

SAGE III/ISS and predecessor solar occultation instruments provide multi-wavelength extinction profiles. Bourassa et al. (2012) compared limb scattering retrievals from Odin-OSIRIS with SAGE III and found good agreement at extra-tropical latitudes. Likewise, Taha et al. (2021) compared coincidence and zonal mean V2.1 OMPS-LP aerosol extinction coefficient

with SAGE III/ISS and found good agreement at equatorial latitudes, but larger relative differences above 25 km outside of the tropics. For example, at high altitudes and latitudes in the northern hemisphere, SAGE III/ISS compared to OMPS shows ~ 40% lower extinction coefficient for the shorter wavelengths (Taha et al., 2021, Fig. 6). This means that the SAGE III/ISS CR values will be smaller and particles relatively larger compared to OMPS-LP (Fig. 1b). With these caveats in mind, using





the SAGE III/ISS CR extinction coefficient provides a verification of our retrieval scheme and allows us to verify the more

complex extinction coefficient measurements made by OMPS-LP.

Note that OMPS-LP makes measurements near 1:30 PM local time while SAGE III/ISS measurements are at local sunrise and

sunset, thus for our comparisons, we follow Taha et al. (2021), and use the zonal median retrieval values for both from OMPS-

LP and SAGE III/ISS. For the SAGE III/ISS data, we use multiple profiles in the latitude range indicated, while OMPS-LP

measurements are selected to be near coincident with the location of the SAGE profiles. We apply our algorithm to SAGE

III/ISS data but use a different CR (521nm /869 nm). These wavelengths are the closest available wavelengths from SAGE

III/ISS that correspond closely to the two wavelengths used in our algorithm for OMPS-LP. Aside from comparing the retrieved

median radius and number density, we can also look at the influence of different OMPS-LP scattering angles in the retrievals.

In Fig. 7 the retrieved radius and number density are compared for four different scattering angle regimes. In the figures, the

measurement bands extend from the first quartile to the third quartile of the data, with a line at the zonal median. Some values

of the SAGE III/ISS aerosol extinction coefficient at higher altitudes are zero or negative due to the very low aerosol

concentrations during the observing period (Taha et al., 2021). This produces an artificially wide measurement uncertainty

band in the figures.

In general, Fig. 7 shows that the retrievals from OMPS-LP and from SAGE III/ISS are closely matched at all altitudes. The

typical particle radius is in the range 0.1 μm and 0.15 μm and does not show significant variations between the two retrievals.

The number density from OMPS-LP is slightly larger than that from SAGE III/ISS as expected from the discussion above.

There is no significant difference among each scattering angle range, except at the scattering angle of 33° above 24 km, where

OMPS-LP has a lower particle radius and higher number density. This result is consistent with the Bourassa et al. (2012) and

Taha et al. (2021) comparisons where poorer agreement between limb scattering retrievals and SAGE measurements were

found above 25 km, where the aerosol loading is very low. The larger tropical (71º-75º) difference below 18 km is mostly

caused by cloud contamination, as indicated by the large standard deviation.

The Fig. 7 comparisons verify our assertion that errors due to aerosol phase function variation with radius are minor (see

Section 2.3), and that the extinction coefficient estimates from the OMPS-LP L2 algorithm are robust. Besides under ambient

condition (background condition) in Fig. 7, the retrieval comparation is also conducted during Raikoke volcano eruption as

Fig. 8. Even larger particles exist during that period, the retrievals show favorable consistency as well, ~0.03 μm or 19%

difference in the retrieved radius. We also note that bias between SAGE III/ISS and OMPS-LP in the northern extra tropics is

consistent with Bourassa et al. (2012) and Taha et al. (2021) comparison between OMPS-LP and SAGE extinction coefficient

variations with wavelength.





## 3.2 Validation of OMPS-LP retrievals with *in situ* balloon measurements

Balloon *in situ* stratospheric aerosol measurements are routinely made using the University of Wyoming LASP Optical Particle Counter (LOPC) (Deshler et al. 2022; Kalnajs and Deshler, 2022). LOPC is usually flown with a condensation particle counter (CPC) which measures condensation nuclei (CN, Campbell and Deshler, 2014). The CPC measures particles from 0.075 to

0.005 µm which are not seen by the LOPC. The LOPC data is fit to a bimodal log-normal size distribution and the data files report the parameters to these fits. For particles smaller than 0.075 µm, the fits depend strongly on measurements from the CPC. The total LOPC particle concentration includes particles that are too small to scatter light and thus LOPC concentrations may be higher than satellite derived concentrations where there is an abundance of CN.

To minimize the effect of spatial and temporal differences in aerosol measurements, we use the OMPS-LP profiles that are nearest and have the closest time to balloon measurements. Figs. 9 and 10 show two characteristic comparison profiles. The specific observation time and location for each profile are noted in the figure captions. The data shown in Fig. 10 was taken shortly after the Raikoke volcano eruption. The particle radius and number density retrieved from the OMPS-LP profile are shown as red lines, the black lines are balloon observations with an uncertainty range. The retrievals assume different aerosol

size distribution widths ($s$ in Eq. 3) in each column. The uncertainty ranges of OMPS-LP retrievals are calculated from the uncertainty in the color ratio extinction coefficients (Eq. 5). To account for uncertainty in the assumed width we show the results for widths varying from $s=1.2$ to $1.8$. To better understand the consistency between OMPS-LP retrievals and balloon measurements, please refer to Fig. A2 for a detailed depiction of the absolute differences between the two profiles.

The comparisons in Fig. 9, 10 show the best agreement is for $s = 1.6$. This is consistent with analysis of Wrana et al (2021, 2023) for SAGE III/ISS observations at altitudes above ~ 20 km where the particle distributions are unimodal. Both the OMPS-LP and balloon data particle radius are ~0.1 µm at all altitudes. The particle radius decreases with increasing altitude for both OMPS-LP retrievals and in the balloon data above 20 km. This decrease can be explained by the fact that smaller particles will have slower settling speeds and thus a longer residence time at higher altitudes in the stratosphere, whereas the

larger particles will settle out to lower altitudes. At 16-18 km the satellite-derived number concentrations have substantial low-bias compared to the balloon-derived number concentrations, with the smaller particle radius there potentially consistent with ultra-fine aerosol particles invisible to the OMPS-LP becoming dominant. In Fig 10, the region where the balloon measured particle size distribution is strongly bimodal, our retrieval shows a jump in particle radius, which is not unexpected. The color ratio method does not give us enough information to generate parameters for a bimodal distribution, so the algorithm tries to

compensate by increasing the radius. The increase in radius produces a decrease in number density because the extinction is mostly due to fewer large particles (see Section 2). The reasonable agreement using $s = 1.6$ shows that this particle width works under both ambient and the aged volcanic plume conditions.





## 4 Background Aerosol Radius and Concentration and Volcanic Perturbations

As an application of our retrieval scheme, we show the variations in aerosol radius and number density retrieved during
background and two volcanic events, Raikoke and Hunga-Tonga. Recall that our retrieval cannot provide particle radius
information above ~0.3µm, - larger particles with CR values of 1.1 are identified as clouds and are removed before the
algorithm is applied. Figure 11 shows the extinction coefficient vs CR (a, c, e) and radius vs concentration (b, d, f) distributions
for these three situations. The data in Fig. 11a tends to split in two lobes that are identified in Thompson and Vernier (2013)
and Schoeberl et al. (2021) as aerosol and cloud. Aerosols have low extinction but high CRs whereas clouds have high
extinction and low CRs. A connecting tail between the two distributions are likely cloud-aerosol mixtures (Thomason and
Vernier, 2013) that occasionally show up in the observations. The labelled regions in Fig. 11a, c, e as aerosol and sub-visible
cirrus are based on the categorization in Schoeberl et al. (2021). Note that the analysis latitude range varies for each figure,
background -90° to 90° in Fig. 11a, b, 30° to 60° in Fig. 11c, d, -30° to 15° in Fig. 11e, f.

### 4.1 Background Aerosols

Figure11a shows that background shows a high concentration of measurements with extinction coefficients of ~ $4 \times 10^{-4}$ km$^{-1}$
with CR values of 2-3. The regions of lower extinction and higher CR may indicate the process of aerosol formation or
evaporation of larger particles. A high concentration region connects to an SVC region with lower CR and higher extinction.
The connection region may consist of larger aerosol particles in the accumulation mode or a mixture of aerosols and clouds.
Thomason and Vernier (2013) identified the region as small cloud particles. Figure 11b shows that the highest concentration
of aerosols observations is in the ~0.1 µm range with number density between 2 and 20 particles/cm$^3$ consistent with
observations shown in Section 3. The screening of CR < 1.1 has reduced the number of high extinction particles in the SVC
domain. The other panels in Fig. 11 are discussed below.

### 4.2 Aerosol changes after the Raikoke volcano eruption

Raikoke (48.29 °N, 153.25 °W) erupted on June 21, 2019, and injected materials to a height of 17-19 km (Muser et al., 2020;
Gorkavyi et al., 2021; Boone et al., 2022). Below ~18 km, the Raikoke plume moved both north and south, (Gorkavyi et al.,
2021, Fig. 8) following the eruption. The plume mostly consisted of sulfate aerosols (Boone et al., 2022) and ash (Bruckert et
al., 2022), but there is also evidence that the plume continued to self-loft after the eruption, reaching 27 km altitude (Khaykin
et al., 2022). Figure 11c and 11d show the distribution of aerosols for a 30-day period following the eruption. We restrict the
analysis to 30°-60°N where the southward spread of Raikoke aerosols occurred (see Fig. 8 in Gorkavyi et al., 2021). On top
of the background distribution (Fig. 11a), Fig. 11c shows a band of high extinction coefficient observations (> $10^{-3}$ km$^{-1}$) with
CR values ranging from 0.8 to 4.5. This band locates the volcanic aerosols. The aerosol particle radius and concentration plot
(Fig. 11d) show a shift toward larger particles with a concentrations > 50 particles/cm$^3$ compared to background.



Figures 12 and 13 show the time series of zonal median retrieved particle radius and number density, respectively, at different latitude regions following the eruption (day 0). The median tropopause height is also plotted in these figures. Note that extinction/radius/number density may be contaminated by ice-lofting summer convective systems near and below the tropopause that make it through our cloud screen. The eruption cloud is initially at 50° N, and as it moves southward the aerosols are detected at more southerly latitudes at a later time (Gorkavyi et al., 2021; Boone et al. 2022). At 50°-60°N (Fig.

12a) the aerosol radius begins to increase after day 20 between 10-15 km. The number density (Fig. 13a) also increases rapidly below 15 km. Between 30° and 50°N the upper boundary of the enhanced particle domain appears to increase in altitude consistent with self-lofting although southward and upward transport by the residual circulation may also have contributed to the altitude increase. At all latitudes after day 80, the particle radius at higher altitudes decreases consistent with aerosol settling. Initially, the volcanic signal is less discernible in the number density plots but becomes more evident after day 80.


The overall Raikoke plume morphology agrees with SAGE III/ISS measurements shown in Gorkavyi et al. (2021) and Schoeberl et al. (2021). Our result also consistent with the larger particles radius derived by Thomason et al., (2021) and Knepp et al. (2022). A recent study by Wells et al. (2023) used model simulation and OMPS-LP measurements to show that including ash in the model simulation produces better agreement with the measurements.


While our retrieval algorithm may have a larger uncertainty when it comes to volcanic aerosols compared to the background aerosols due to the potential variations in the width of the size distribution and the aerosol refractive index, The impact of the distribution width is limited. Especially, Fig. 10c suggest that $s = 1.6$ is a good approximation for the later stage of aerosol evolution. The change of scattering angle over times is also a considerable factor when discussing the time series of retrieved

aerosol properties. The scattering angle range is 30° - 85° (Figure A3), which results in ~23% algorithm uncertainty for a 50% phase function error (see section 2.3). Furthermore, Taha et al. (2021, Figure 3) showed that V2.1 retrievals have very little sensitivity to the assumed aerosol model errors due to the viewing geometry variations. Figure 13 a, b also shows a region with very large number density above 20 km which appears after day 40. This anomaly is due to an overestimate of the CR resulting a small median radius and then to match the extinction the algorithm generates a very high number density. We

suspect that this anomaly is due to issues with the OMPS-LP extinction coefficients retrieval under very low aerosol concentrations (see Taha et al., 2021, Fig. 6).

**4.3 Aerosol changes after Hunga Tonga-Hunga Ha'apai (HT) volcano eruption**

The Hunga Tonga-Hunga Ha'apai (hereafter HT) volcano (20.54 °S, 175.38 °W) erupted on January 13 and 15, 2022, and injected volcanic material high into the stratosphere. HT lofted a significant amount of water vapor into the stratosphere. The

impact and evolution of lofted water vapor and aerosols on dynamics and radiation can be found in Schoeberl et al. (2022, 2023) and Sellitto et al. (2022). Surprisingly, the amount of $SO_2$ injected was relatively low considering HT's volcanic



explosivity index of 5 (Carn et al., 2022; Millán et al., 2022; Sellitto et al. 2022). Nonetheless, the particles that formed after the injection were primarily sulfate aerosols (Taha et al., 2022; Bernath et al., 2023).

Figures 11e and 11f show the CR vs aerosol extinction coefficient distribution at HT eruption latitudes for Feb.-March, 2022. The HT aerosol plume has two distinct populations (Figure 11e), one lower extinction population with a CR of ~2.5 and a higher extinction population with a CR of ~1.5. Using the *in situ* measurements (Baron et al., 2023), we assume a log-normal particle size distribution width of 1.2 for this eruption event, which consistent with SAGE III/ISS measurements (Wrana et al., 2023). The two distinct populations translate into two size populations. A population at ~ 0.3 µm with a concentration of ~5

$cm^{-3}$ represents the higher altitude portion of the plume whereas the 0.23 µm population at 0.2 $cm^{-3}$ is represents the ambient distribution.

Figure 14 shows the time series of zonal median retrieved particle radius and number density between 12.5°S-17.5°S starting at Jan 1, 2022. A transition from a width 1.6 before the eruption to 1.2 following the eruption is applied. The extinction

coefficient (Fig. 14a) begins to increase immediately after eruption (Jan. 15) rising to $5x10^{-3}$ $km^{-1}$ by day 110. After the eruption, gaseous $SO_2$ rapidly converts to sulfate aerosol (Taha et al., 2022; Zhu et al., 2022; Sellitto et al., 2022) which produces a rapid rise in extinction coefficient. The particle radius between 22 and 25 km starts to increase from background 0.1 µm to over 0.3 µm. These changes in particle radius were also observed by Kloss et al. (2022) using a balloon-borne measurements in the end of January 2022. The median radius grows through day 30-80. The 0.4 µm peak in the median radius

appears near day 80 between 22 km and 25 km. After that, the median particle radius begins to decrease as the larger particles settle out. Both Legras et al. (2022) and Schoeberl et al. (2022) argue that the aerosol distribution settling rate is characteristic of 0.5 µm or larger particles. Note that 0.5 µm radius is above the retrieval limit for our algorithm, but the top of the aerosol cloud is clearly descending from day 120 to day 170.

Figure 15 shows vertical profiles of aerosol particle median radius (a-d) before the eruption (a) and three days (b-d) days following the eruption. The dates of the four profiles are indicated by the vertical dashed lines in Fig. 14. Prior to the eruption, the aerosol particle radius is less than 0.1 µm below 20 km; above 20 km, radius is roughly ~0.1 µm. Figure 15b-d shows aerosol radius after the eruption, and the subsequent profiles show the radius increase in the 22-26 km range. As noted above, the increase in size at ~26 km is due to rapid conversion $SO_2$ to sulfate. The scattering angle range is 85° - 130° (Figure A4)

indicates a less than 15% uncertainty for a 40% phase function error. The retrieved median radius of ~0.3 µm is consistent with Taha et al., (2022).



## 5 Summary and Conclusion

Satellite remote sensing of aerosols provides a global perspective on their role in climate change due to volcanic and pyroCb injections (Kremser et al., 2016) and as well as basic information on the potential use of aerosols in geo-engineering (NASEM, 380 2021). The particle radius and concentration of stratospheric aerosols are key quantities needed to determine volcanic aerosol impact on climate. Deriving aerosol particle radius and number density from satellite measurements provide constraints for the climatic effects of volcanic eruptions.

In this study, we have applied an existing technique to derive stratospheric aerosol radius and concentration from OMPS-LP 385 limb profiler satellite measurements. Using the Mie code inside the SASKATRAN radiative transfer model, we show that the ratio of extinction coefficient at two wavelengths (color ratio) is independent of the number density and only a function of the particle size for a log-normal size distribution. Thus, the color ratio can be used to estimate the particle median radius. With the median radius, we can then compute the cross section and use the extinction coefficient to determine the number density (Eq. 1). Our algorithm follows the approach of Bourossa et al. (2008b), Rieger et al, (2018) and Wrana et al. (2021). The 390 algorithm cannot distinguish radius greater than ~0.4 µm because the color ratio ($E_{510}/E_{879}$) approaches one. These particles are usually identified as large aerosols or clouds (Schoeberl et al., 2021). The OMPS-LP algorithm (Taha et al., 2021) assumes the same PSD for each wavelength and determines the extinction coefficient by matching the observed radiance with radiance computed using a radiative transfer model. The OMPS-LP algorithm then iterates the extinction coefficient until it matches the observed radiance. The free parameter in this approach is the number density, thus the OMPS-LP algorithm will retrieve 395 an aerosol number density that varies with wavelength. However, using the color ratio, we estimate the median radius of the PSD and then derive a consistent aerosol concentration using the extinction coefficient.

We verify our algorithm using SAGE III/ISS aerosol extinction coefficients and compare the results to OMPS-LP derived aerosol properties. Our results show that OMPS-LP measurements are consistent with SAGE III/ISS measurements as also 400 noted by Taha et al. (2021). Furthermore, we validate our retrieved aerosol median radius and number density with balloon measurements (Kalnajs and Deshler, 2022) under both ambient and volcanic conditions.

There are three major sources of uncertainty in our radius calculation: uncertainty in the measured radiance, uncertainty in assumptions about the log-normal size distribution width, and uncertainty in OMPS-LP assumed phase function. The retrievals 405 show the highest sensitivity to the assumed distribution width. We examined how the variation in retrieved aerosol radius with changes in distribution width during the comparison to balloon measurement. We find that the radius changes for different distribution widths can be substantial leading to non-negligible uncertainties in median radius and derived number concentration. However, the good agreement between our retrievals and *in situ* observations suggests a width of 1.6 is a reasonable value under both ambient conditions and the Raikoke volcanic eruption. This assessment is also consistent with





results from Rieger et al. (2018), Chen et al. (2018), Wrana et al. (2021), and Bourassa et al. (2012). The exception appears to be the Hunga-Tonga eruption where SAGE measurements support a smaller distribution width of 1.2 (Wrana et al., 2023) which leads to larger particle sizes consistent with balloon observations (Baron et al., 2023). We also found that the uncertainty caused by the algorithm's assumed phase function is minimized when using typical aerosol CRs although the uncertainty can reach ±12-±32% for volcanic aerosol.


Finally, we use the derived the median radius and particle number density to assess their spatio-temporal variation in the months after the 2019 Raikoke and 2022 Hunga-Tonga eruptions. Our retrieved particle sizes are consistent with the reported evolution of the aerosol clouds associated with those eruptions (Wells et al., 2022; Schoeberl et al., 2022; Taha et al., 2022; and Khaykin et al., 2022). At higher altitudes the radius decreases following the eruption is consistent with settling of larger
aerosols.

*Acknowledgments.* The authors would like to thank the OMPS-LP team for Level-2 data production and the in-situ balloon measurement team led by Terry Deshler at University of Wyoming for providing the data used in this study. We also acknowledge discussions with Daniel Zawada, Adam Bourassa, Zhong Chen, Pawan K. Bhartia, and Matthew T. Deland. The
reviewers of the original manuscript also helped us shape and improve the current version.

*Code and Data availability.* All the data sets used in this study can be downloaded at no cost although users must be registered. The OMPS-LP L2 data are available at https://snpp-omps.gesdisc.eosdis.nasa.gov/data/ (doi:10.5067/CX2B9NW6FI27). SAGE III/ISS data (https://doi.org/10.5067/ISS/SAGEIII/SOLAR_HDF4_L2-V5.1) data are accessible at the NASA
Atmospheric Sciences Data Center. The *in situ* balloon data are archived at https://doi.org/10.15786/21534894. Data analysis products shown here are available from the corresponding author.

*Author contributions.* YW and MS were responsible for the development of the aerosol retrieval algorithm, which is described in this paper. YW was responsible for code improvements and testing. YW and MS wrote the initial draft of the paper. GT
participated in the scientific discussion about OMPS-LP data and made substantial contributions to the paper. All authors reviewed the manuscript and provided advice on the text and figures.

*Competing interests*. The authors declare that they have no conflict of interest.

*Financial support.* This research has been supported by the National Aeronautics and Space Administration, Earth Science Division grant nos. 80NSSC21K1965 and 80NSSC22K0157.



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

**Appendix: Supplemental figures**

Plot of the aerosol and Rayleigh phase function used for OMPS-LP L2 product retrieval algorithm (Fig. A1). The plot of showing how close the OMPS-LP retrievals to balloon measurements (Fig. A2). The plot of the time series of single scattering angle of OMPS-LP central slit after Raikoke (Fig. A3) and Hunga Tonga-Hunga Ha'apai volcanic eruptions (Fig. A4).






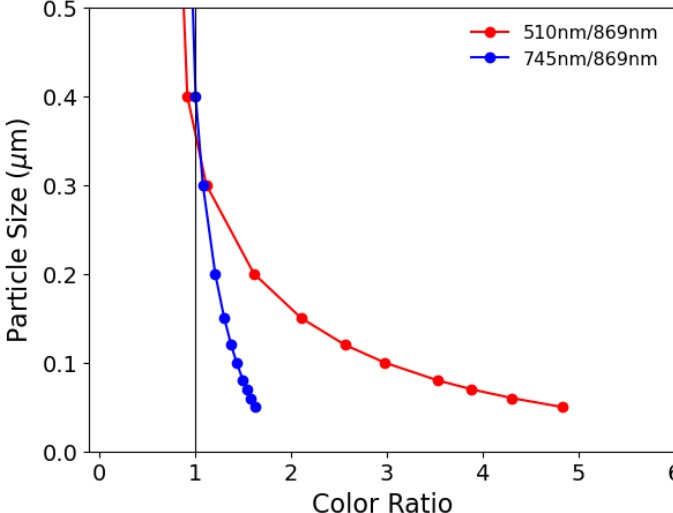

Figure 1: Aerosol particle radius (0.05 µm to 0.5 µm) as a function of aerosol extinction coefficient Color Ratio (CR) (510 nm/869 nm and 745 nm/869 nm) computed from SASKTRAN's Mie code based on a log-normally distributed aerosol with width (s) of 1.6. The black line denotes color ratio of one.




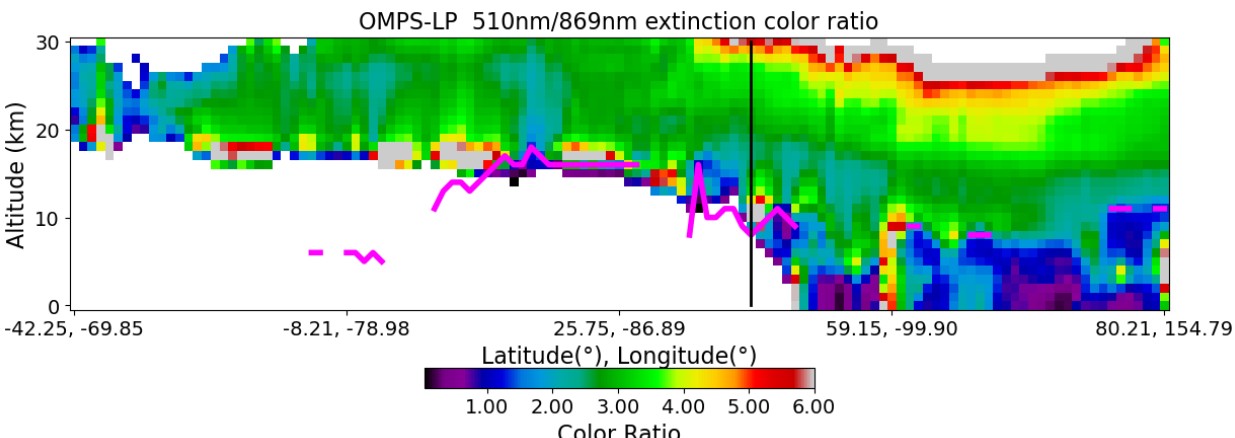


Figure 2: The OMPS-LP CR (510/869) on 9/13/2020. The magenta line shows the OMPS level 2 cloud height. The vertical black line refers to the profile shown in Figure 9.



Figure 3: The median radius (size) as a function of color ratio (510 nm/869 nm) and particle distribution width. Color contours are $\log_{10}$ of median radius in µm. The width is parameter s for the aerosol size distribution. Black contours are median radius in µm.



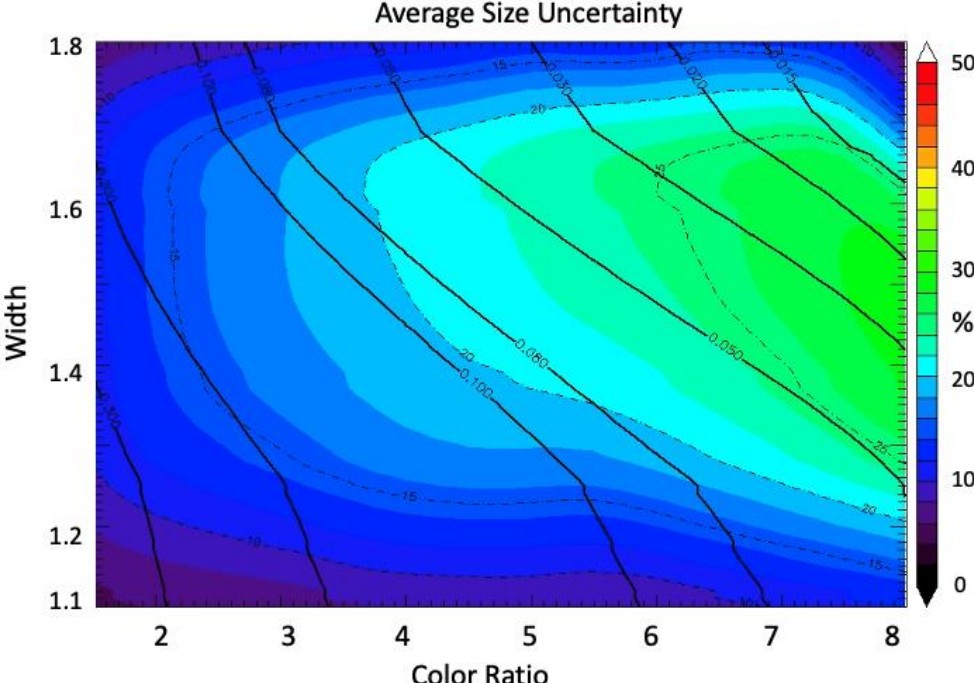

Figure 4. Average median radius (size) uncertainty in % as a function of distribution width and color ratio. The color contours and dashed contours are the normalized radius error in % for a distribution width (s) uncertainty ±0.2 and color ratio (510 nm/869 nm) uncertainty of 21%. The black contours are radius in µm.


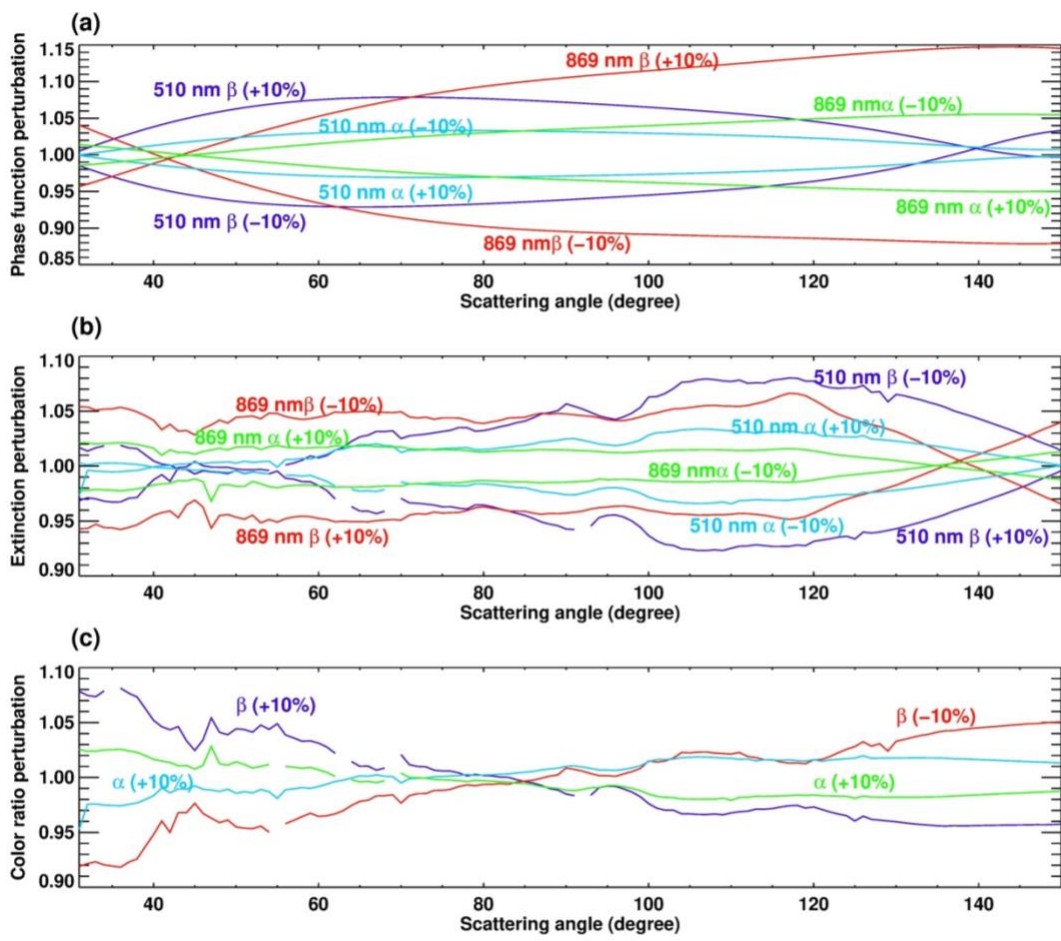

Figure 5: plot of simulated phase function (a), aerosol extinction (b), and color ratio (510 nm/869 nm) (c) perturbations as caused by gamma parameter changes of ± 10%. Perturbations are shown relative to OMPS-LP operational retrieval at various

scattering angles. Scattering angles represent the range of are in the northern hemisphere, while large scattering angles are in the southern hemisphere (Taha et al., 2021). The aerosol extinction was retrieved using a single orbit on 12 September 2016 at 20.5 km altitude. The fitted parameters for the gamma aerosol size distribution are $\beta$ = 20.5, $\alpha$=1.8, and median radius $r_m$=0.1µm.




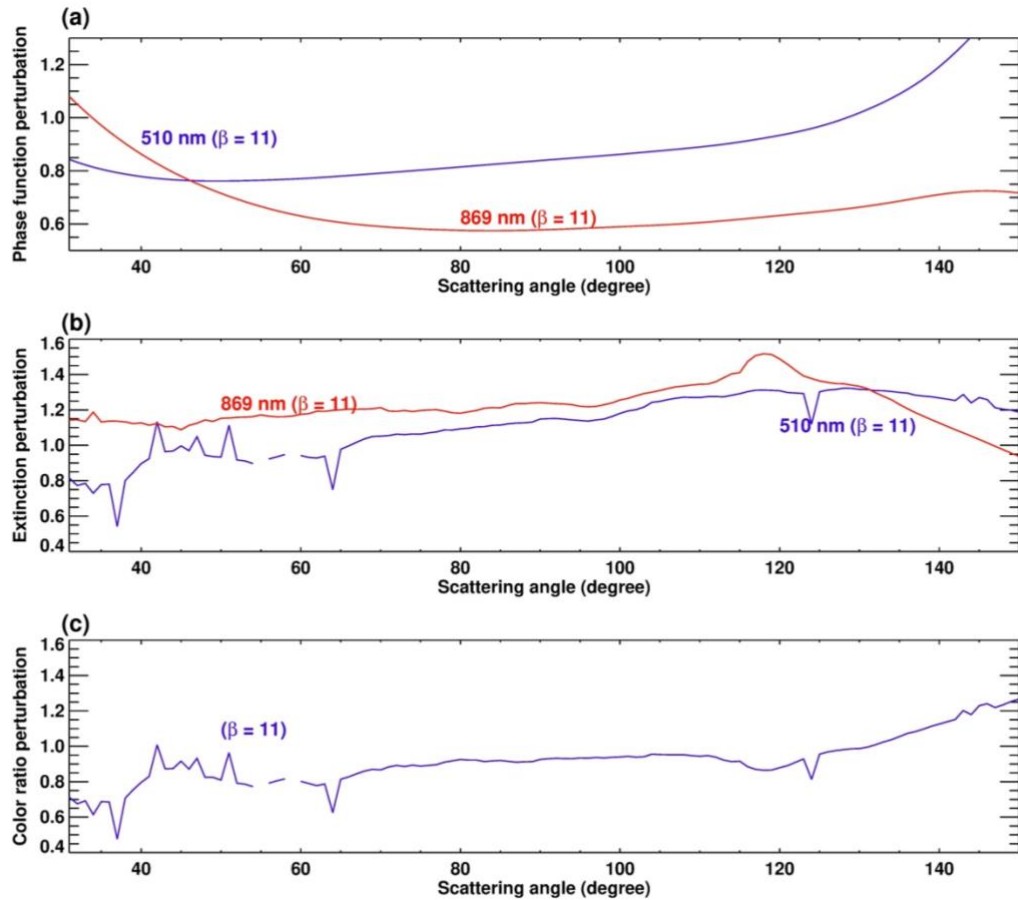

Figure 6: Same as Figure 5 except β = 11, α=1.8, $r_m$=0.2µm






Figure 7: The particle radius (upper row) and number density (bottom row) retrieved from OMPS-LP measurements (red color)
and SAGE III/ISS measurements (blue color). Band width extends from the first quartile to the third quartile of the data, with
a line at the zonal median. For the SAGE data, we use multiple profiles in the latitude range indicated. The OMPS-LP
measurements are selected to be near-coincident with SAGE profiles. The four sub-figures across the page show data for
different OMPS-LP scattering angle ranges.




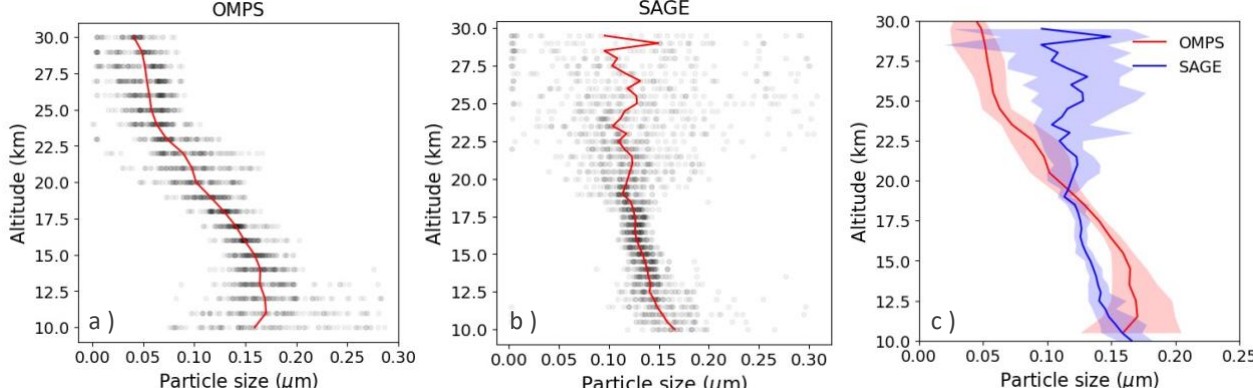

Figure 8: The retrieved aerosol particle radius from OMPS-LP (a) and SAGE III/ISS (b) between 45 °N and 47 °N during
Raikoke volcano eruption (Oct 31, 2019 – Nov 03, 2019). The red lines are median values. Comparisons between the two
retrievals (c) is shown with the band width extending from the first quartile to the third quartile of the data, along with a line
indicating the median.



Figure 9: Comparisons between OMPS-LP satellite-derived and CPC-LOPC balloon-derived particle radius and number density. Red lines show the particle radius (upper row) and number density (bottom row) retrieved from a single OMPS profile on 9/13/2020 19:00 UTC (42.03°N, 91.75°W) with different aerosol size distribution width (*s*) assumptions in each column. The single scattering angle of central slit is 57.35°. The shaded area is the uncertainty range. For the OMPS number density, we only show the upper limit of uncertainty range as the lower limit extends to zero. The black lines in each panel are the CPC-LOPC-derived particle radius and number density on the same day at 12:00 UTC (41.31°N, 105.65°W).



Figure 10: The same as Figure 9, but for the measurement on 8/28/2019 after the Raikoke eruption (6/21/2019). The CPC-LOPC balloon data is on 13:21 UTC at 41.31°N, 105.65W; The single OMPS-LP profile is on 20:04 UTC 42.65°N, 107.88°W, and the single scattering angle of central slit is 61.84 °.


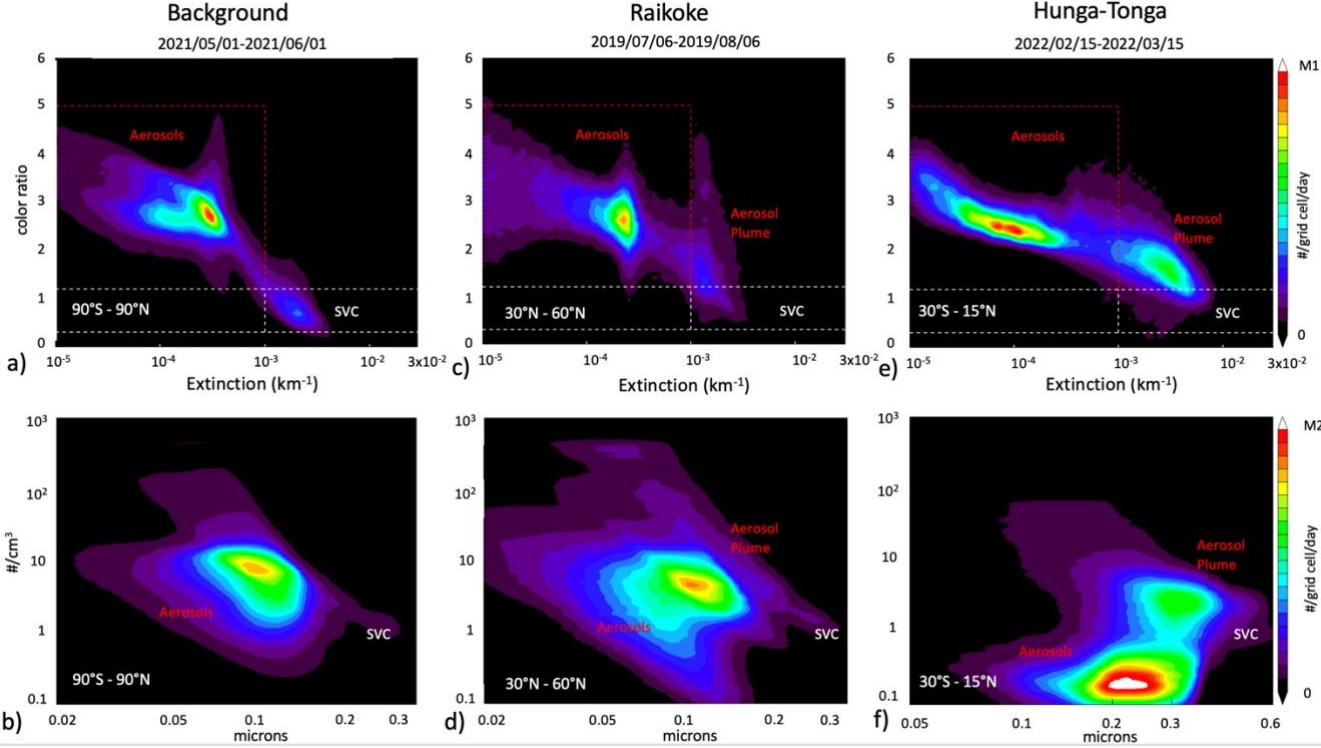

Figure 11: Measurement distributions during different observing conditions. The distributions are not normalized. They contain different number of measurements due to the changing observing period. Upper plots, OMPS-LP extinction vs CR (510nm/869nm), lower plots show particle number density vs median radius using our algorithm. Parts (a, b) non-volcanic background, Parts (c, d) period following Raikoke eruption (e, f) period following the Hunga-Tonga eruption. Date ranges are shown at the top of the figure; latitude range is shown inside each figure. Dashed lines on the top diagrams indicate approximate classification regions, aerosols, aerosol plumes, and subvisible cirrus (SVC). These regions are also identified in the lower figures. The color scale range is different for each figure M1=200, 30, 35 & M2=1200, 150, 300 for the color bars, respectively.






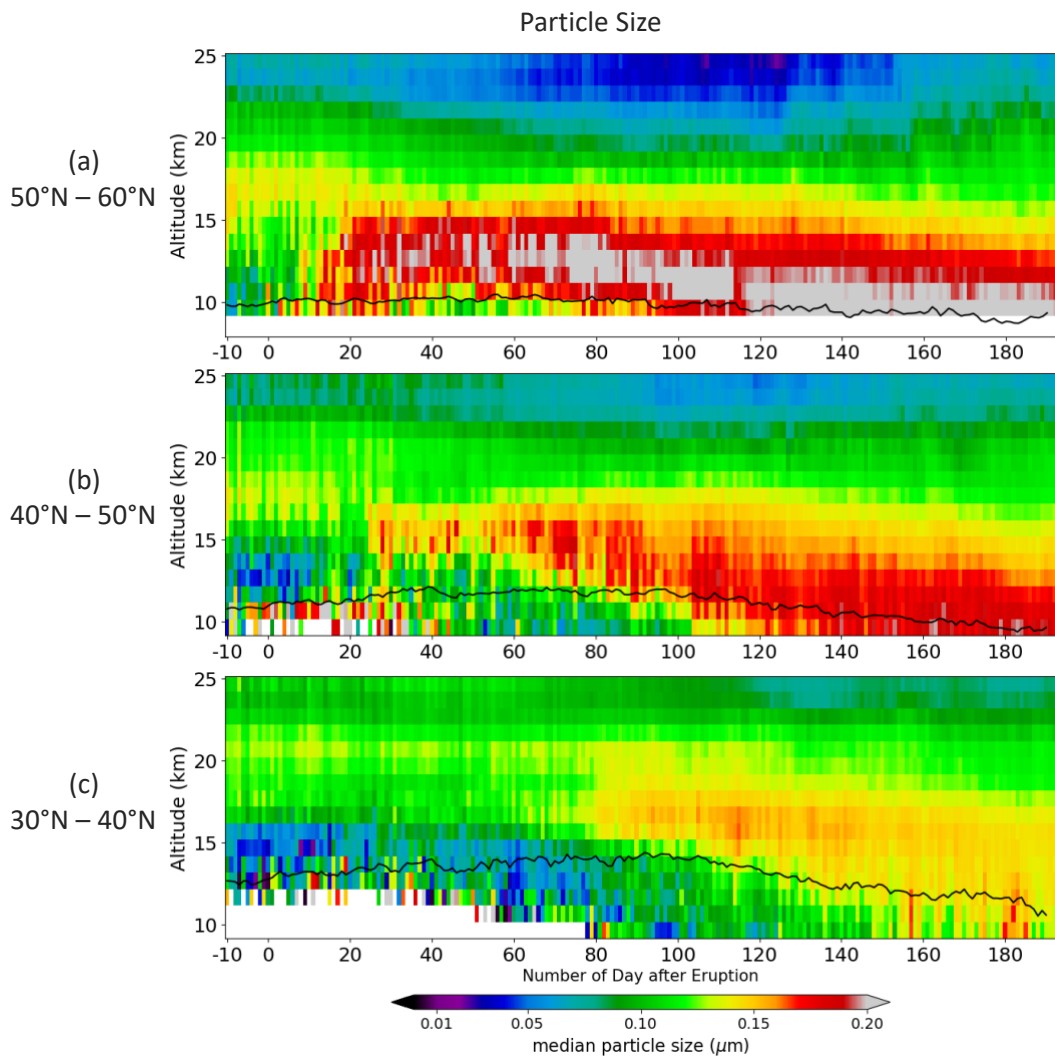

Figure 12: The time series of zonal median particle radius in three latitude ranges shown (a, b, c) after Raikoke volcanic eruption. Day 0 is June 21, 2019. Black line is the mean tropopause height.

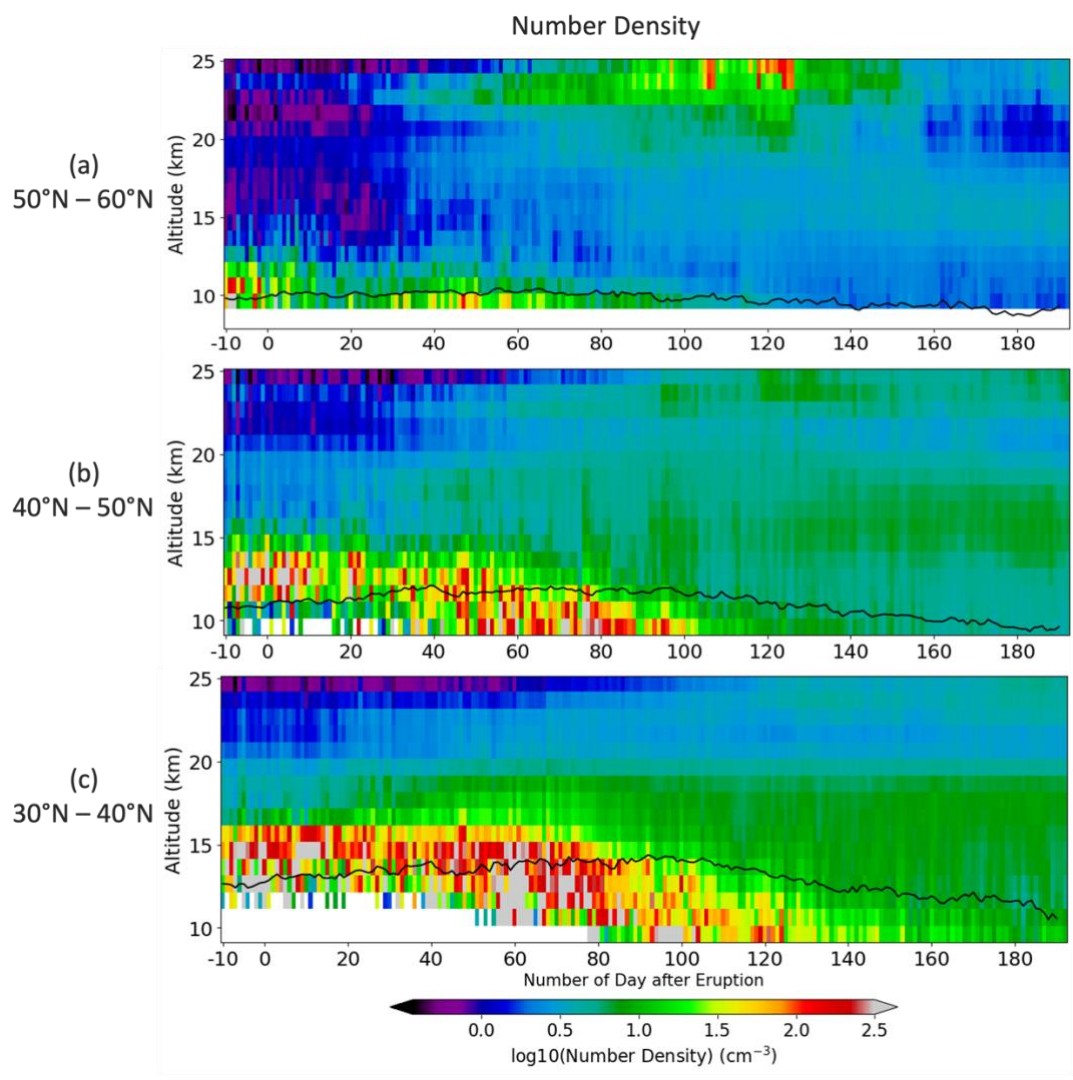


Figure 13: The same as Figure 12, but for number density.






Figure 14: Hunga Tonga-Hunga Ha'apai time series of zonal median (a) 869nm extinction coefficient, and (b) particle radius at 15° S beginning Jan 1, 2022. The 26 km height level is marked as a red line. A vertical line on day 15 indicates the eruption.
Four vertical dash lines refer to the four vertical profiles shown in Figures 15 (day 14, 45, 70, and 100 of 2022).

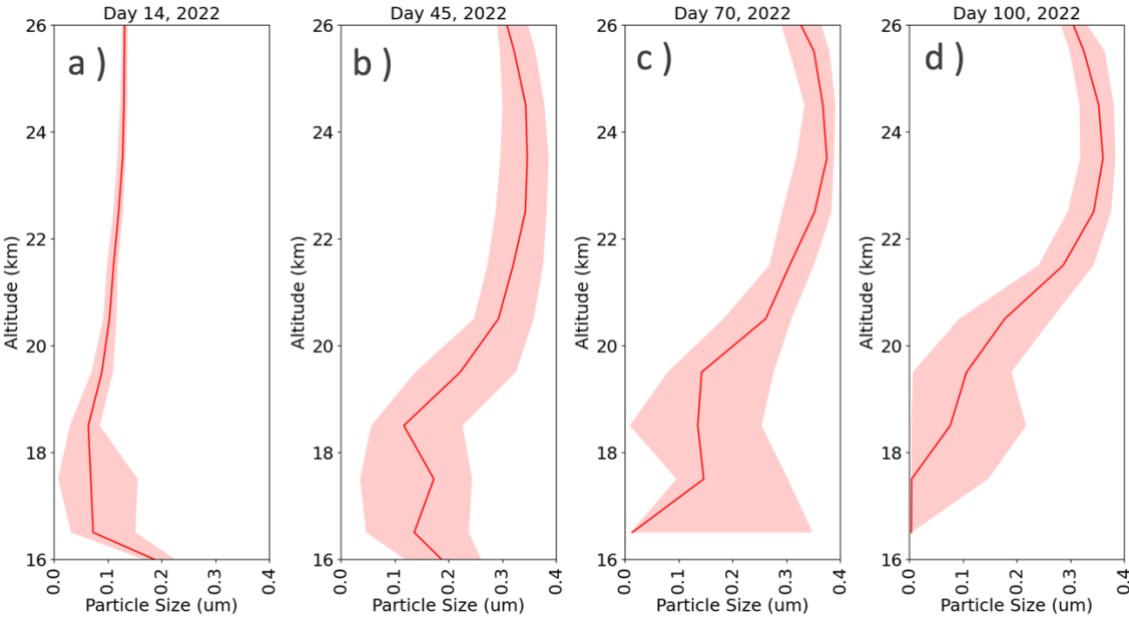

Figure 15: The particle radius retrieved from OMPS-LP between 12.5 °S and 17.5° S. The band width extends from the first quartile to the third quartile of the data, with a line at the zonal median. The four columns illustrate the different dates before and after the Hunga Tonga-Hunga Ha'apai eruption on Jan 15, 2022. Part (a), day 14, part (b) day 45, part (c) day 70, and part (d) day 100 of 2022.


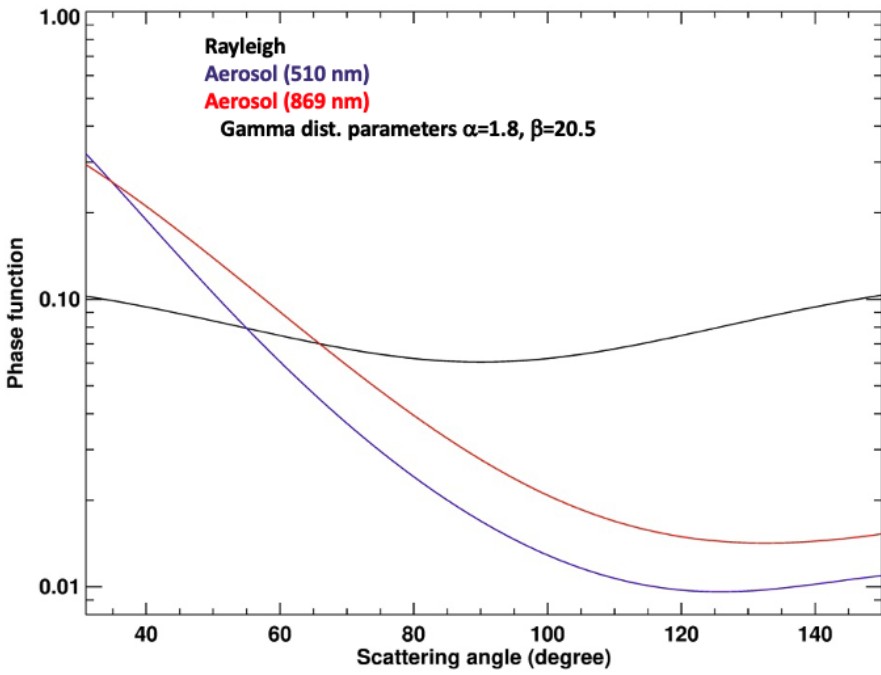

Figure A1: Plot of the aerosol phase function used for OMPS-LP L2 product retrieval algorithm (which assumes gamma size distribution) at 510 (blue) and 869 nm (red). Rayleigh phase function is also shown (black). Gamma function coefficients
are shown in the figure legend, see Chen et al., (2018, Eq 6).





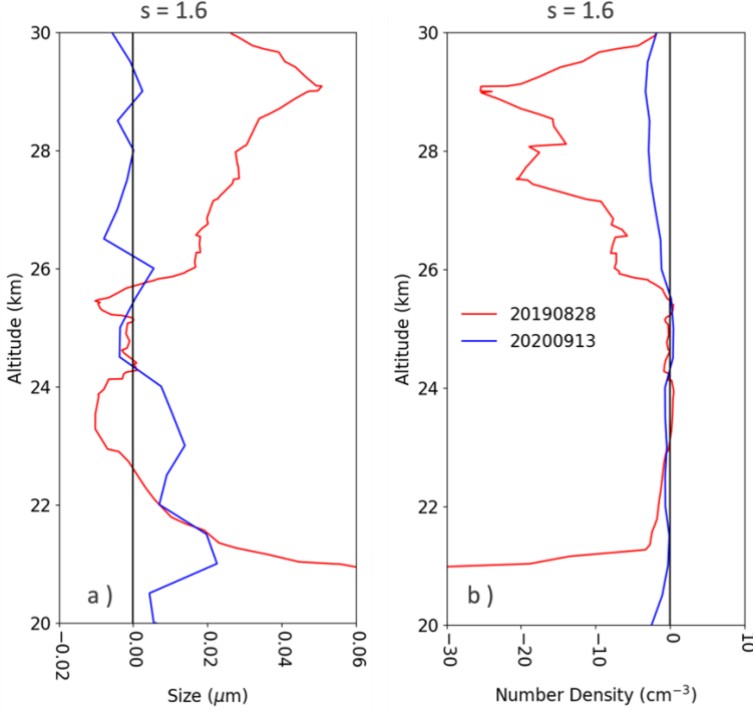

Figure A2: The value of OMPS-LP satellite-derived aerosol particle radius (a), and particle number density (b) minus CPC-LOPC balloon-measurements. Blue lines show the profile on 9/13/2020 (ambient/background condition), and red lines show 8/28/2019 (after Raikoke volcanic eruption – 6/21/2019). The profiles are smoothed with a 9-point moving average. The data processing details are described in Figure 9 and 10 captions.



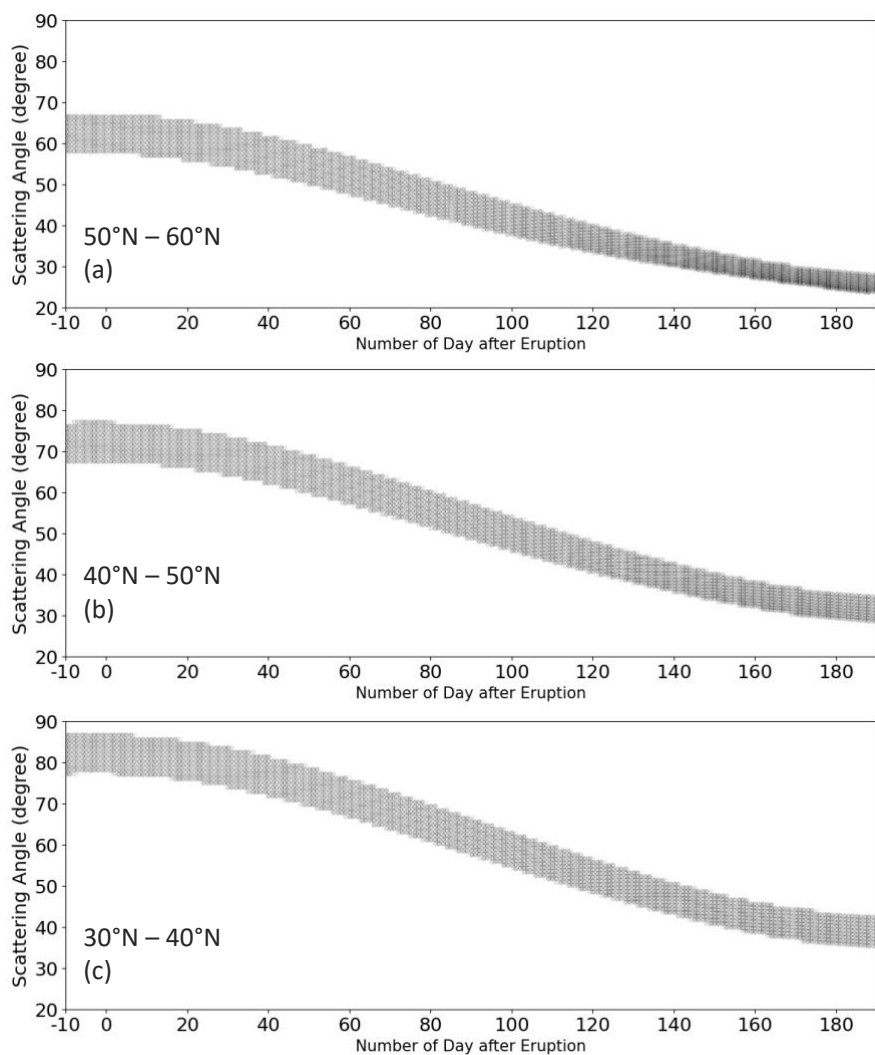

Figure A3: The time series of single scattering angle of central slit in three latitude ranges shown (a, b, c) after Raikoke volcanic
eruption. Day 0 is June 21, 2019.





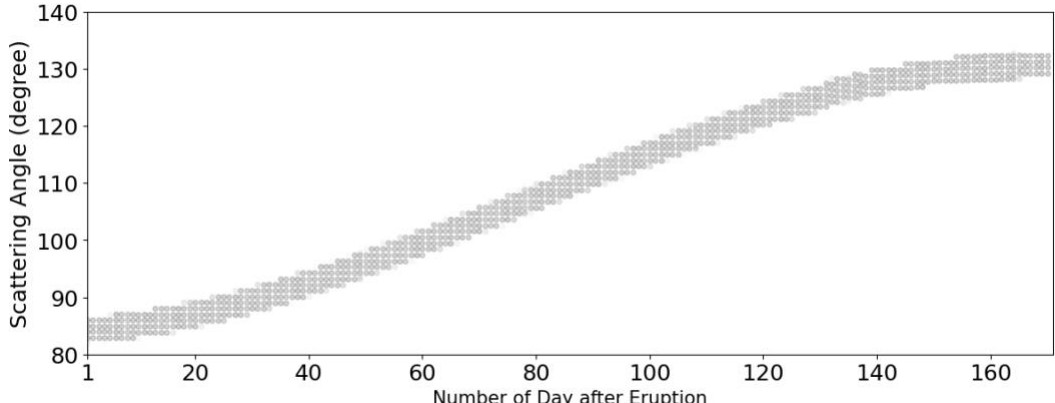

Figure A4: Hunga Tonga-Hunga Ha'apai time series of single scattering angle of central slit beginning Jan 1, 2022.