# Peer review of "Using OMPS-LP color ratio to extract stratospheric aerosol particle median radius and concentration with application to two volcanic eruptions"

_Atmospheric Measurement Techniques, 2023_

## Referee Comment (RC1)

**Review on Manuscript No. AMT-2023-267**

*Using OMPS-LP color ratio to extract stratospheric aerosol particle median radius and concentration with application to two volcanic eruptions* by Wang et al.

Wang et al. derive stratospheric aerosol microphysical parameters from OMPS using the color ratio. They discuss various sources of uncertainty and apply their algorithm to measurements from SAGEIII on ISS and validate their method with balloon data and then apply the algorithm to measurements made during two volcanic eruptions.

I personally do not have any problems with this study being published. Although the method itself of using two wavelengths for deriving microphysical parameters seems not to be new, I still think this study is of value for the community since it is applied to a different instrument and under different conditions than the ones used in previous studies. Especially when only measurements at two wavelengths are available such a method is clearly of value. However, I agree that the referencing should be done correctly, that the study should be better motivated and that the implications for the community should be more clearly stated. In addition, before resubmission the manuscript should be carefully checked for language and technical correctness. The present version is full of mistakes and could have definitely be prepared with more care.

**General comments:**

For me it did not become clear why you do everything with two wavelengths if there are more wavelengths available and one could do everything more accurately without making so many assumptions? Or are the observations at several wavelengths only available from SAGE III and not from OMPS? If you have only measurements at two wavelengths from OMPS I can totally understand why you develop the method you are presenting here. If there however are measurements of three or more wavelengths available from OMPS I would wonder why you make analyzing the data more complicated than necessary.

I think adding subsections describing the OMPS-LP and SAGE III measurements would be quite helpful for the reader. This is definitely missing in the current version of the manuscript. You cannot expect the reader to be an expert on both instruments. Further, I would suggest that you first describe the comparison to the balloon measurements since this is the main comparison (validation) of your method and then present the SAGE III comparison.

Why do you apply your method then to SAGE III? Why is it necessary to have a comparison to another satellite? Could you more clearly state what you try to achieve with this comparison? Couldn't you also use more than two wavelengths from SAGE III to also show what the difference to the more accurate way of deriving aerosol microphysical parameters is?

**Specific comments:**

P1, L15: "….assuming…… width of 1.6". You are not only assuming this with value. You also derive the best result/agreement with this value, thus I think you could/should clearly state this.

P1, L22: Instead of citing the text book from Seinfeld and Pandis (2016) I would suggest to cite Brock et al. (1995) and/or Kremser et al. (2016). Note, stratospheric (sulfuric acid) aerosols form mainly in the tropical upper troposphere (see Brock et al., 1995). Please rephrase/correct the sentence.

P1, L28: What is meant with "black" and "brown" aerosols? Please explain this to the reader. To my knowledge these terms are only used for tropospheric aerosol (e.g soot containing aerosols).

P1, L28: What is meant with "self loft"? Without explanations this sentence is useless. I would suggest to omit this sentence.

P2, L34: What about SAGE and SAGE II? I think a sentence about this two instruments should be added.

P2, L37-38: Is the stellar occultation technique also used to measure aerosols? Without any further explanations this sentence is useless and could be omitted.

P2, L51: Are there currently or in the future other missions measuring aerosols in the stratosphere? Or will OMPS be the only one irrespective of the measurement technique? When was OMPS launched and how long are the measurements expected to continue? Please add here some more information.

P2, L63-P3, L64: Without any further explanation this sentence is useless. What is the difference between your algorithm and the one from Thomason and Vernier (2013)? What are the differences between the algorithms? What has been changed?

P3, L69: Add here a section or sections describing the instruments/data used in this study (SAGE, OMPS, balloon)

P3, L77: What are  N, r_0 and s are. This has not been explained.

P3, L81: Why in this study? How is the color ration defined in other studies? Isn't the color ratio always defined the same way?

P3, L83: There is no Ångström exponent in Equation 1b.

P3, L8: Which wavelengths combinations are available and which ones have been used?

P4, L116: How "large"? Please give some numbers.

P4, L121: Why? Why this value? Do you derive this value from Fig. 1? Provide figure or reference for this value.

P5, L134: Using a log-normal distribution is quite common for stratospheric aerosols and thus here you should instead of a specific paper rather cite a textbook as e.g. Seinfeld and Pandis (2016).

P6, L176: If the error is large for small radii wouldn't this then cause difficulties when plume events are considered where many small particles are produced?

P6, L185-186: In both sentences you cite Chen et al. (2018). One of these citations could be skipped and the grammar of the second sentence should be corrected.

P7, L21-202: Not necessarily. You will have both small and large particles. If you make such a statement you should give a reason and/or provide a reference.

P7, 208ff: I would suggest to change the order and to first describe the validation with the balloon data and then the verification with SAGE. If you want to keep the current order you should provide a motivation or reasoning why this order is more logical.

P8, L227: Which wavelength pair has been chosen and why? How would the result differ if a different wavelength pair would be used?

P8, L231: Be more precise. Cleary state above which altitudes.

P8, L235: "matched at all altitudes". This is not correct. You do not have a match at all altitudes. Give the altitude range. Further, I see here a better agreement for particle size than for number density.

P9, L279: If this method does not give you information on bimodal distributions, isn't than that a significant drawback for investigating volcanic plumes? Especially shortly after the eruption the distribution will be bimodal for a certain time. Could you give an estimate in percent how large the deviation for the derived microphysical parameters are?

P9, L281: "……. Because the extinction is mostly due to fewer large particles…..". Sentence not clear. Please rephrase.

P10, L296: What is meant with "concentration"? A high number of measurements with extinction coefficients 4x10-4 km-1?

P10, L298: Which sizes? Give some numbers.

P10, L300: Also here. Give a number. How small?

P10, L309-310: This is not clear. What has the self-lofting of the plume to do with the composition?

P10, L313: Give a number.

P11, L316: Is the eruption equal to day 0? If yes, I would write or state this more clearly, e.g. eruption = day 0.

P11, L319:  at a later time -> add when exactly

P11, L323: at higher altitudes -> at which altitudes. Add a number.

P11, L323-324: the settling or sedimentation process is several times mentioned, but never explained.

P11, L326-327: Why referring here to other studies? Why don't you include a figure showing the same from SAGE?

P11, L328-329: This result is for this study not important and thus, the sentence does not make any sense here.

P11, L333-334: Why is the impact of the distribution width limited? The evolution of the width could be simulated with a box model. I assume that the small particles will quickly disappear (within a few days). You could check the literature for modelling studies of volcanic eruptions. I guess you could find there some numbers how quickly the distribution is back to the background distribution.

P12, L352: Sentence not clear. Check and rephrase. Further, I could not find your statement in Wrana et al. (2023). How did you derive this width? From the figures?

P12, L365: Don't just write larger. Give numbers.

P12, L366: Which settling rate? You mean the sedimentation rate of the particles. Do you mean that particles < 0.5 µm sediment out?

P12, L374-375: Sentence not correct. Please check and correct.

P13, L378-382: This paragraph is giving a motivation for the study and thus this paragraph should rather appear in the introduction than in the summary and conclusion section.

P13, L387:  Clearly state here that you refer here to log-normal distributions with one mode.

P13, L390-391: The two sentences should be rephrased and maybe combined. Add also why a CR of 1 is problematic.

P13, L403-405: I think before you stated the opposite. Further, the second sentence starting with "We examined……." is incomplete.

P13, L406-408: Add more details. Under which conditions does this happen?

P14, L420: What are the future implications? For what can the method be applied? The future OMPS measurements? Nevertheless, during volcanic eruptions you have large uncertainties and you need to rely on other studies to derive your input values?

Reference list: Check the style. Some journal names are written out. others not. Same with the author names. In some cases the entire first name is written, in most other cases the first initial. Check the ACP guidelines and prepare your reference list accordingly. Further, there are some references misplaced as e.g. Kremser and Yue and Deepak. The Box and Deepak paper is listed in the reference list, but not cited throughout the manuscript.

Figure 9 and 10: Add a legend to the figure. Is there no uncertainty range for the CPC data given or are these so low that these are not visible? What are the black zigzag lines on the right and left corners of the figure?

Figure 11: Wouldn't it make much more sense to compare the background at 30°-60°N with the distribution during Raikoke at 30°-60°N and the background at 30°S-15°N to the 30°S-15°N distribution during Hunga-Tonga? I really think the same regions should be compared.

Figure 14: I think it would be better to use black instead of red lines. Also omitting some lines would be helpful, e.g is the 26 km line really necessary? I think this one could be omitted. Why is here the eruption on day 15 and not on day 0 as before?

**Technical corrections:**

P1, L20: Introduce the abbreviation "Cb".

P1, L20: Reference of Kremser et al. (2016) is missing in the reference list.

P2, L42: Full stop before reference of Taha et al. (2021) obsolete.

P2, L57: PyroCB -> pyroCb. Use a consistent writing.

P3, L65: In the next section, we detail -> In the next section we describe in detail

P3, L79: scattering measurements -> scattering measurement

P3, L80: is the same as Wrana et al. (2021) Eq. 2 -> is the same as Eq. 2 in Wrana et al. (2021)

P4, L97: Add "the" -> we simulate the scattering….

P4, L103: show the how CR -> show how the CR

P4, L118: Add "a" -> We found that selecting a CR

P4, L119-120: Check sentence and correct sentence.

P5, L137-138: Either use singular or plural.

P6, L171: PyroCB -> pyroCb

P6, L186: Comma obsolete (reference of Chen et al.)

P6, L188: Comma obsolete.

P7, L197: "CR (510/869)" here you write it without adding nm. In other occasions nm is added. This should be done more consistently throughout the manuscript.

P7, L210: using the same algorithm -> the above described algorithm

P8, L246: comparation -> comparison

P8, L246:  Add "as shown" -> during Raikoke volcano eruption as shown in Fig. 8

P8, L248: Add "the" -> the bias

P8, L249: Add "the" -> "the comparisons" and move references at the end of the sentence.

P9, L275: settle out -> sediment

P10, L283: Use comma instead of writing twice "and" -> Background Aerosol Radius, Concentration and Volcanic Perturbations.

P10, L285: Add "conditions" so that it reads "background conditions"

P10, L288: Thompson -> Thomason

P10, L287: Move "Figure 11" before "(a, c, e)" so that it reads "The extinction coefficient vs CR is shown In Fig. 11 (a, c, e) and radius vs concentration distributions for these three situations in Fig. 11 (b, d, f).

P10, L291: Put a, c, e in parenthesis -> (a, c, e)

P10, L296: One "shows" obsolete. Write "background aerosols" instead of just "background".

P10, L301: aerosols -> aerosol and density -> densities

P10, L302: Section 3 -> Sect. 3

P10, L 306: Comma before reference of Gorkavyi obsolete.

P11, L318: screen -> screening

P11, L320: add "altitude" after 10-15 km

P11, L327: add "is" and particles should read particle -> Our result is also consistent with the larger particle radius……

P11, L332: After the comma "The" should be "the".

P12, L350: Feb.-March -> February to March

P12, L353: Add "is" -> which is consistent

P12, L355: Delete "is" before represents.

P12, L372: Add "the" -> the radius

P12, L373: Add "the" -> the aerosol radius

P12, L372: Add "altitude" -> the altitude range

P12, L374: Add "of" -> conversion of SO2

P13, L390: radius -> radii

P14, L416: Delete "the" before "median radius".

Figure 2 caption: Add "nm" after 501/869. Use a consistent writing style throughout the manuscript.

Figure 5 caption: plot -> Plot and add space before and after "="

Figure 6 caption: Add space before and after "="

**References:**

Brock, C. A., P. Hamill, J. C. Wilson, H. H. Jonsson, and K. R. Chan (1995), Particle formation in the upper tropical troposphere: A source of nuclei for the stratospheric aerosol, *Science*, 270(5242), 1650–1653, doi:10.2307/2887916.

Kremser, S., et al. (2016), Stratospheric aerosol—Observations, processes, and impact on climate, *Rev. Geophys.*, 54, 278–335, doi:10.1002/2015RG000511.

---

## Community Comment (CC1)

[supplement omitted: unrelated document]

---

## Community Comment (CC5)

**Overview**

The authors present a revised version of a paper that was originally submitted in 2023 (https://amt.copernicus.org/preprints/amt-2023-36/). Unfortunately the current incarnation of this manuscript fails to provide substantive changes from the previous submission and the authors fail to address any of the major challenges that were raised. To date, the methodology they present remains fundamentally flawed (this claim is supported by figures that the authors provide in their manuscript and will be expounded below), and the authors continue to gloss over these critical aspects without warning the reader of the impact of their assumptions. Rather, the authors "support" their assumptions by incorrectly citing various papers, ignoring other key papers that refute their assumptions, and ignoring the plethora of in situ data that invalidates their assumptions. An obvious case of this last point can be observed in their Fig. 10. If the authors were to plot the OPC-measured distribution width (herein referred to as either "distribution width" or $\sigma$) they would show the reader the large degree of variability $\sigma$ has within a single profile (for this flight $\sigma$ varied from 1.18 to 2.45). This plot alone would be enough to tell the reader that a static distribution width is not only ill advised, but wrong. Instead, this point is ignored throughout the text and the reader is left with an incorrect impression of the integrity of this method.

This is a major problem with this method: using a static distribution width inevitably forces the $r_m$ to a specific subset of the solution space which may, or may not, be close to the actual atmospheric conditions. The problem is, given this method, we will *never* know whether the assumed $\sigma$ value was right or wrong (and, by extension, we will have no confidence in the inferred $r_m$ estimate) unless we happen to be fortunate enough to have a coincident OPC flight that can be used to inform the authors' algorithm and confirm the output (if the authors were to provide a rigorous validation of their algorithm then we could certainly have more confidence). The authors demonstrate this by shifting their $\sigma$ value from 1.6 to 1.2 for the HTHH event; how would they know to do this without the OPC data? Further, the $\sigma$ in that profile changed, substantially, throughout the profile (again, see their Fig. 10). Shifting their assumed $\sigma$ from 1.6 to 1.2 made it a better guess in parts of the profile, but a *far worse* guess in other parts of the profile (again, without the OPC data, and in the absence of a robust validation effort, how would we know where the assumed $\sigma$ is "good" or "bad"?). In short, even when we have OPC data to inform our assumptions we must be very careful on how these assumptions are used because the atmosphere, especially after eruptions and/or wildfires, is not homogeneous.

Regarding the static distribution width, one may wonder how the authors justify their value ($\sigma =1.6$). They repeatedly, and incorrectly, refer to the Wrana et al. 2021 (`https://doi.org/10.5194/amt-14-2345-2021`) paper. I can only speculate how they misinterpret that paper (perhaps it is a misinterpretation of the Wrana et al. 2021, Fig. 4, which is for *a single profile*). The problem is that Wrana et al., 2021 did not provide a statistical evaluation of the distribution of $\sigma$ values (or any other size distribution parameter). However, Wrana et al. did state "The upper and lower boundaries of the colour bars in Fig. 6 roughly mark the ranges within which the values of the respective quantities fall for the data set between June 2017 and December 2019. The median values for this time frame at 20 km altitude are 130.6 nm for the median radius, 1.54 for the mode width $\sigma$,..." Granted, the median $\sigma$ value, at 20 km, from Wrana et al. 2021 was 1.54 (close to the authors' value of 1.6), but the authors must recognize that the color bar scale for $\sigma$ (Fig. 6 of Wrana et al. 2021) extended from $\approx$1.1 to 2.0. This is a direct refutation to the claim that Wrana et al. 2021 supports the selection of a static $\sigma$ of any value.

The authors make no mention of the bias in the OMPS extinction products under aerosol loading as describe in Bourassa et al., 2023 (`https://doi.org/10.1029/2022GL101978`), the socalled 1-D assumption and convergence problems. Bourassa et al. 2023 showed that, when the stratosphere is volcanically perturbed, the extinction is a factor of 2 too high at the aerosol peak, while, below the peak, the extinction product is a factor of 2 too low. While Bourassa et al. evaluated the performance of the 755 nm channel the conclusion of the paper is clear: *the bias is inherent in the overall retrieval methodology and there is no reason to believe this bias is limited to a single channel* (we note that the University of Saskatchewan's tomographic retrieval method alleviates, but does not obviate, this bias). Work done in our group (unpublished, but Mahesh Kovilakam suggested that he will provide figures in his review of this paper) indicates that this bias is not constant across wavelengths, but increases as you move to shorter wavelengths; hence, the bias does not cancel out in the ratio.

Finally, the authors fail to provide a convincing validation of their new product, which is essential. They provide a comparison with 2 balloon-based in situ measurements and declare both to be in "reasonable agreement" (line 281). However, the agreement in their Fig. 10, within the main aerosol layer, was off by at least a factor of 3 (i.e., the OPC measured $r_m$ remained steady at $\approx$50 nm from 16–21 km, while the OMPS method yielded $r_m$ estimates that ranged from $\approx$150 to >300 nm) and the profile shapes are not even close to being similar. If anything, this "validation" exercise proved the opposite of what they intended (i.e., it showed quite clearly that their product is not reliable outside background conditions). Can the authors justify calling this "reasonable agreement".

The authors go on to provide what they call an "evaluation" of their product by running OMPS and SAGE III/ISS data through their algorithm to determine how the derived $r_m$ values differed (e.g., their Figs. 7 and 8). What they fail to recognize is that the agreement is foreordained from the outset and that any agreement they show here is really a convoluted comparison of the 2 instruments' extinction ratios. *This in no way provides an evaluation of their algorithm. If the input numbers are similar then it is impossible for the output to differ.*

The authors miss an opportunity to compare their product with the recently release SAGE III/ISS particle size distribution (PSD) parameters (Knepp et al., 2024; `https://doi.org/10.5194/amt-17-2025-2024`). I do not fault them for this as our product was only made available within the last few months. Further, I do not suggest this as a shameless self promotion of our work, but the SAGE PSD parameters provide a unique data set that provide uncertainty estimates for the PSD parameters, all of which could be used to evaluate the performance of their algorithm. Unfortunately, we already know how their algorithm will perform since we did a similar evaluation of deriving PSD parameters from SAGE II (i.e., using a single extinction ratio of 520:1020 nm). This is shown in section 6.3 of Knepp et al. 2024 with the statistics summarized in Table 9 of that paper. What we observed is the $r_m$ estimates were off by up to -124%, -95%, -28%, 3%, and 21% at the 10[th], 25[th], 50[th], 75[th], and 90[th] percentiles. In short, 80% of our estimates were between 124% too low to 21% too high. There simply is not enough information within 2 channels (i.e., a single extinction ratio) to make an accurate estimation of $r_m$.

Finally, all papers require a purpose, the "so what" factor. After reading it, we should be able to answer the question "What did this paper tell us that is new?" or "How is the community now smarter?" The authors fail to deliver on this point. Herein, the authors put a new dataset (OMPS color ratios) through an aerosol size algorithm that was developed more than 40 years ago and this methodology is as limited in information content now as it was then. What did we learn? The method is not new and we do not gain any new insights into the atmosphere. Since the method is not new I would expect, at the bare minimum, a thorough demonstration of the validity of this method (i.e., a robust validation), but instead we get 2 comparisons with OPC profiles and an ill conceived evaluation with the SAGE III/ISS instrument. If the authors were to make this product

available to the community *it could not be used without the end users first doing the validation work that should have been done here.* In my view there is no scientific merit in this work.

It is for these reasons that I recommend that this paper be rejected for publication.

**Brief synopsis of the authors' work**

Herein, the authors demonstrate how the size (radius, herein referred to as $r_m$) of stratospheric aerosol can be estimated using extinction ratios (or color ratios, herein referred to as CR) using OMPS data. This method is not novel, and the authors appropriately cite the Yue and Deepak (1983) paper. This method is based on finding CR values in a Mie theory lookup table that fall within the expected uncertainties. From there, the authors average the radii of particles that fall within their solution space and return an estimated $r_m$. The authors then compare their estimates with those measured on 2 flights of the University of Wyoming's Optical Particle Counter (OPC) and then apply their method to OMPS data collected after 2 volcanic eruptions: Raikoke and HTHH. The authors also present a sensitivity study to determine the impact of different OMPS viewing geometries on their $r_m$ estimates.

**General Concerns with the Manuscript**

– This paper requires a thorough proof reading to correct the multitude of grammatical errors, incomplete sentences, and passages that are difficult to interpret.

– The authors make claims throughout the manuscript and provide references to bolster these claims. Many of the cited works do not support these claims. Some of these papers have nothing to do with what the authors are claiming. This is particularly egregious because many of these issues were raised in the first round of reviews for their original manuscript, yet they remain unchanged here. Please check all references and confirm they are correct.

– The authors cite other works that demonstrate a fundamental misunderstanding of the cited work. The Wrana et al. 2021 paper is particularly abused within this manuscript. These issues were raised during the review of the original paper and, to date, remain unchanged. This should be corrected before resubmission.

– The authors gloss over glaring issues within the OMPS data product and how these problems will impact their $r_m$ estimates. Bourassa et al., 2023 (doi.org10.10292022GL101978) and Gorkavyi et al., 2021 (doi.org10.5194amt-14-7545-2021) discuss the bias in the OMPS extinction product, but the authors never address this issue herein. I have discussed this with Mahesh Kovilakam, who will also submit a review that goes into more detail on this aspect, but this can be easily seen by comparing the global half-to-1-micron ratio of OMPS to that of SAGE.

– Previous work (Bourassa et al., 2007 `oi:10.1029/2006JD008079`) demonstrated that Rayleigh optical depths, for scattering instruments, are high enough at lower wavelengths (i.e., 510 nm) that limb scattering observations are insensitive to aerosol signatures. This relates back to the previous comment. Perhaps this is a misunderstanding on my part, but after having numerous conversations about this paper (conversations that involved people outside our

working group and outside our organization) I can confidently state that lack of fidelity in the OMPS 510 nm channel is a common perception (a perception that is supported by analysis involving the 510 channel). It would greatly benefit the reader if the authors were to demonstrate that the 510 nm channel is reliable and refute the claims of Bourassa et al. 2007.

**Specific comment on this manuscript**

The authors original statements will be presented in quotation marks (black font), while my comments and questions will be in blue.

1. lines 12–14: "We apply our algorithm to extinction coefficient measurements made by the Stratospheric Aerosol and Gas Experiment on the International Space Station (SAGE III/ISS) to verify our approach and find that our results are in good agreement."
   Whether you call this a "verification", "validation", or "evaluation", it does not matter because it was not done here. There was no validation/evaluation/verification of the algorithm. The authors used OMPS data to estimate $r_m$ and then used a coincident SAGE profile to estimate $r_m$. This provides nothing more than a roundabout comparison of the two extinction products for a few select profiles (none taken under volcanic aerosol loading, which avoids the issue pointed out by Bourassa et al., 2023). The authors should either perform a meaningful validation of their product or remove this claim from the manuscript.

2. line 36: Reference to Thomason et al., 2021
   This is an incorrect citation. This paper did not estimate particle sizes as the authors claim (other than to say the particles were "probably big" or "probably small"). This reference should be removed and all citations should be checked to ensure they are appropriate throughout.

3. line 55: Here the authors provide a brief enumeration of uncertainty sources in their products. Reference to the "arch effect" as discussed in Bourassa et al. 2023 and Gorkavyi et al. 2021 would be appropriate here as would the Rayleigh scattering optical depth issues as raised by Bourassa et al. 2007.

4. line 60: "Our algorithm is similar to the color ratio method developed by Thomason and Vernier's (2013)..."
   As written, this sounds like Thomason and Vernier 2013 determined particle sizes, which they did not. Please clarify.

5. line 75: It seems odd to refer to Q as scattering efficiency here since you are dealing with extinction efficiencies. This seems like a unnecessary detour. Please consider whether this nomenclature is necessary (your call on whether it is changed or not).

6. line 76 and Eq. 1a: The PSD equation is a function of r (array of particle radii over which Eq. 1a is integrated), $r_m$, and $\sigma$. Please update for clarity.

7. line 76: Why the jump from Eq. 1a to Eq. 3?

8. line 77: It would be helpful to the reader to define N, $r_0$, and s here.

9. line 80: "For occultation measurements p = 1 and Eq. (1a) is the same as Wrana et al. (2021) Eq. 2."
   It is unclear what the authors are trying to communicate here. As written, their Eq. 1a is identical to Wrana's Eq. 2, which makes this statement tautological. Please clarify.

10. Line 121 in regards to Fig. 2:
    Though not critical to the paper, why the discrepancy at -8.21/-78.98? Why would the OMPS retrieval stop >10 km above the cloud top?

11. Lines 146–148: "Wrana et al. (2021) used a third extinction wavelength from SAGE III/ISS data to estimate the log-normal distribution width and found that most of the observations clustered between s = 1.4 and 1.6. This information also constrains our size distribution uncertainty."
    The authors misinterpreted this paper. Wrana et al. 2021 does not support this assumption. Here, the authors ignore a wealth of data that refutes the assumed range of distribution widths. For example,

    (a) the Wyoming OPC record shows a lot more variability especially outside background conditions.

    (b) Wrana et al. 2021 (Fig. 6) showed far more variability, between 2017–2019, than the authors claim. From Wrana et al. 2021 "The upper and lower boundaries of the colour bars in Fig. 6 roughly mark the ranges within which the values of the respective quantities fall for the data set between June 2017 and December 2019." It is important to note that their colorbars extend from $\approx$1.1 to 2.0, which refutes the authors claim.

    (c) Wrana et al. 2023 showed more variability than what is claimed here.

    (d) Knepp et al., 2024 (`https://doi.org/10.5194/amt-17-2025-2024`) likewise showed more variability. This product is readily available for download and use.

    Ultimately, this results in the authors imposing an artificial constraint within their algorithm that preferentially steers their algorithm to a subset of solution spaces.

12. Lines 151–152: "The uncertainty in CR, $U_{CR}$, can be estimated from Taha et al. (2021, Fig. 6) comparison to SAGE III/ISS." As written, I do not understand what this sentence means. Please revise.

13. Line 156: "Rieger et al. (2018, Fig. 6) which gives a width uncertainty of $\approx$0.2"
    The authors misunderstand Fig. 6 of Rieger et al. 2018. This figure tells the reader the range of PSD values used in their simulations and is descriptive, not prescriptive. The authors are not using Rieger's Fig. 6 for its intended purpose and this figure does not support their $\sigma$ uncertainty.

14. Line 156: "...consistent with the Wrana et al. (2021) estimate."
    Again, the Wrana et al. 2021 paper does not support this claim.

15. Line 159: I do not understand what is meant by "outside of the averaging domain." Would the authors please clarify?

16. Lines 160–161: "...common distribution widths (s=1.4–1.6)."
    Defining this range of distribution widths as "common" is heretofore unsupported.

17. Lines 163–170: I apologize, but I do not understand what the purpose of this paragraph is or how it fits within the context of the surrounding text. Would the authors please check that this should be here and does not need revised?

18. Lines 169–170: "This example shows how the radius and number density uncertainty due to the distribution width can be quantified."
There is no doubt that uncertainty can be quantified, the fact remains that these values are calculated incorrectly. The range of $\sigma$ values has been artificially constrained to yield an artificially small uncertainty. If the authors were to use a reasonable range of values then their uncertainty estimates would be so large as to make their estimate of $r_m$ meaningless (especially under enhanced aerosol loading).

19. Line 173: The jump from Fig. 4 to 11 here is confusing. Please number figures and equations sequentially, in the order in which they appear in the text as this will aid the reader.

20. Lines 174–175, in reference to Fig. 4.
The content of this figure is misleading. As discussed above, the authors used an artificially small (and unjustified) uncertainty in $\sigma$. Second, the authors neglect to account for the bias inherent in the NASA OMPS retrieval algorithm (as discussed by Bourassa et al. 2023). Failing to account for these issues misleads the reader and skews your subsequent analysis and conclusions that are based on this figure. Please recreate this figure using correct uncertainties and accounting for bias under elevated aerosol loads.

21. Section 2.3
Here, the authors fail to account (or mention) the bias inherent in their retrieval as discussed by Bourassa et al. 2023. If the authors want this to be applicable to volcanic conditions, then this issue must be addressed.

22. Lines 202–207 Here, the authors seem to conflate bias and measurement uncertainty. What they show in Fig. 6(c) is a bias in their CR (an offset) that is caused by assuming an incorrect size distribution. Because this is an offset it cannot be used in place of an uncertainty in their error propagation. Please correct.

23. Line 206: "...we get an uncertainty range of 0.15–0.3 $\mu$m"
What is the range if the authors correctly calculate their errors (i.e., using realistic values of $\sigma$)?

24. Section 3.1
This section is unnecessary. The authors state "The Fig. 7 comparisons verify our assertion that errors due to aerosol phase function variation with radius are minor (see Section 2.3), and that the extinction coefficient estimates from the OMPS-LP L2 algorithm are robust." However, they previously stated that previous studies have evaluated the OMPS/SAGE extinction products so it is difficult for me to see the value of performing yet another comparison, much less one that is based on derived products (products that involve more assumptions and an additional level of abstraction). The purpose of this paper is to introduce a new method for inferring particle sizes from OMPS extinction coefficients and this section does not fit within the scope of that purpose. This in no way provides a validation of their method and cannot, under any condition, be interpreted as a validation or even a meaningful evaluation.

Despite how good Fig. 8 may look, it is ultimately meaningless because, again, you are not comparing 2 independent estimates of $r_m$, you are effectively comparing the 2 extinction products in a very roundabout way. As is, this provides no validation/evaluation and, for the purpose of this paper, is meaningless.

25. Line 244: "The Fig. 7 comparisons verify our assertion that errors due to aerosol phase function variation with radius are minor"
This presupposes that your $r_m$ estimates are correct, which has yet to be demonstrated. All agreement shown in this figure (or disagreement) is strictly due to the similarity in the 2 extinction products and in no way validates your $r_m$ estimate.

26. Lines 265–266: "...uncertainty ranges of OMPS-LP retrievals are calculated from the uncertainty in the color ratio extinction coefficients..."
As written, it seems that the authors did not include an uncertainty factor for the distribution width (please clarify). If they did not, then the uncertainty bounds in this figure are misleading and I ask that they include a full accounting of uncertainty.
I also read in the OMPS v2.1 readme "Loss of sensitivity of short wavelengths radiances to aerosols. This effect is caused by Rayleigh and aerosol attenuation of the limb scattered radiation, and becomes most pronounced below ≈17 km and in the southern hemisphere. We advise caution in using LP aerosol extinction data below 17 km and scattering angle greater than 145 degrees for wavelengths 675 nm or shorter. The error bars provided in the daily product data files do not include this term. This error can be reduced by using 745, 869, and 997 nm wavelengths." My concern is with this sentence: "The error bars provided in the daily product data files do not include this term." Given this statement, are CR-error estimates, which are then used to calculate the overall $r_m$ error, correct?

27. Lines 266–267: "To account for uncertainty in the assumed width we show the results for widths varying from s=1.2 to 1.8"
This demonstrates the sensitivity of your size estimate on the distribution width, which leads to the question of "Which distribution width is correct?" This figure alone provides irrefutable evidence that using a static distribution width is incorrect. This is observed by the disagreement between the 2 profiles (it does not matter what panel you look at) and by plotting the OPC-derived $\sigma$ profile for this flight. The OPC data show variation in $\sigma$ that ranged from 1.18 to 2.45 in the stratosphere. It is physically impossible to generate a meaningful $r_m$ estimate using a single $\sigma$ value throughout the profile.
As mentioned above, the authors have neglected the wealth of information on this topic and arbitrarily selected a value of 1.6 and incorrectly cited Wrana et al. 2021 in support of the 1.6 value, but failed to recognize that Wrana et al. showed (e.g., in their Fig. 6) that the distribution width has a broad range.
There is additional information available from the Wyoming OPC record (see my figure above) that can be useful in setting these boundaries. Recent work by our group shows that distribution width is generally between 1.2 and 1.9 (Knepp et al. 2024 `https://doi.org/10.5194/amt-17-2025-2024`), but this is highly dependent on where and when the measurements are made.

28. Line 270: "The comparisons in Fig. 9, 10 show the best agreement is for s=1.6. This is consistent with analysis of Wrana et al (2021, 2023)"
Neither of these papers support this conclusion.

29. Line 271: "...where the particle distributions are unimodal..."
Is this statement made strictly in regard to Fig. 9 & 10 or is this a more general statement. Looking at the Wyoming OPC record I see numerous examples of bimodal distributions well past 25 km. If we limit the analysis to the 2 profiles you showed here then we can fool ourselves into thinking the situation is better than it is (i.e., your analysis indicates that your algorithm performs better when the aerosol distributions are unimodal, but bimodal distributions are ubiquitous). It would greatly benefit the reader to expand this analysis to include more profiles that contain persistent bimodal distributions.

30. Lines 171–172: "Both the OMPS-LP and balloon data particle radius are ≈0.1 $\mu$m at all altitudes..."
This is categorically false and is refuted by the authors' own figures. A cursory look at the figures shows an overall range of 0.05–0.125 $\mu$m in Fig. 9 and a range of 0.05–0.3 $\mu$m in Fig. 10.

31. Lines 272–273: "The particle radius decreases with increasing altitude for both OMPS-LP retrievals and in the balloon data above 20 km."
This is not correct. The OPC $r_m$ value increased rapidly above 20 km and held steady until ≈ 21 km where it began to decrease.

32. Line 278: What is meant by "strongly bimodal"? Quantifying this would greatly aid the reader. Please quantify to remove ambiguity.

33. Lines 281–282: "The reasonable agreement using s=1.6 shows that this particle width works under both ambient and the aged volcanic plume conditions."
This is categorically false and I fail to see how the authors can make this claim. As is obvious from Fig. 10 this did not work well between 16–20 km where the 2 products differed by up to a factor of 6. Even at 18 km the disagreement remained a factor of 3. This is not good agreement. Please correct this statement.

34. Line 293: "...background -90° to 90° in Fig. 11a, b..."
For this plot, what altitudes were plotted?

35. Line 293: "...background -90° to 90° in Fig. 11a, b..."
It is difficult to understand the classification of this time period at "background" immediately after the La Soufriere eruption and when the atmosphere was still recovering from the ANY event (granted, much of that event already dissipated, but we still see traces in the SAGE data). Please clarify this designation

36. Line 331: While listing uncertainty sources the reader should be reminded of the issues in the OMPS product during elevated aerosol loading (per Bourassa et al. 2007, 2023).

37. Lines 332–334: "The impact of the distribution width is limited. Especially, Fig. 10c suggest that = 1.6 is a good approximation for the later stage of aerosol evolution."
This is wrong. Even a cursory look at these figures shows the distribution width plays a significant role in the size estimate. Contrary to the authors' claim, Figs. 9 & 10 demonstrate the importance of correctly selecting the distribution width. Further, the OMPS-derived $r_m$ in Fig. 10 (c) is 3–6 times larger than the OPC value and the OMPS profile shows a distinctly different structure from the OPC. I am sorry to say, but this difference is not inconsequential:

we cannot declare this a "good approximation". Can the authors please explain to the reader how being off by a factor of 3–6 can be viewed as good?
Please note that this poor behavior is a direct consequence of the static distribution width. What this demonstrates is how a constant $\sigma$ forces your algorithm toward a particular set of $r_m$ solutions. This is a clear demonstration of the inadequacy of this assumption.

38. Line 347: The reference to Taha et al. 2022 is not appropriate since that paper did not determine the aerosol composition.

39. Line 364: "The median radius grows through day 30-80. The 0.4 $\mu$m peak in the median radius..."
It is not clear to me how this is possible since the authors imposed a CR ratio cutoff (1.1) that, per their description, limited you to particles less than 0.3 $\mu$m?

40. Line 398: "We verify our algorithm using SAGE III/ISS..."
As discussed above this exercise in no way evaluated the performance of your $r_m$ algorithm. This only tested the agreement between the 2 extinction ratios and, knowing how well the 2 ratios agree. This is a meaningless "verification" since the result could have been guessed before the analysis began (i.e., if the 2 CRs are close then the inferred $r_m$ will be close; it proves nothing about the algorithm).

41. Line 400: "...we validate our retrieved aerosol median radius..."
Claiming your product is "validated" by comparing to 2 OPC profiles (one of which the agreement was quite poor) is misleading to the readers. Please revise to communicate this accurately.

42. Line 403: "There are three major sources of uncertainty in our radius calculation..."
Per Bourassa et al. 2007, 2023 and Gorkavyi et al. 2021 the authors should include the bias in their extinction product under elevated aerosol loading as a source of uncertainty/error.

43. Lines 408–409: "However, the good agreement between our retrievals and in situ observations suggests a width of 1.6 is a reasonable value under both ambient conditions and the Raikoke volcanic eruption."
I realize I have hit this point throughout the paper, but since the authors make this claim throughout the paper it seems they are genuinely unaware of how wrong this is. An assumed static distribution width of 1.6 is wrong and should not be done. The authors should know this from their Fig. 10, from Wrana et al. 2021, Wrana et al. 2023, Knepp et al. 2024, and the entire OPC record. This single assumption introduces a massive source of error and directs the algorithm to a predetermined subset of their solution space (this is clearly seen in Fig. 10).
I will note that assuming a value of 1.6 is a reasonable assumption during background periods, though you must still properly account for natural variation, which the authors have yet to do. However, background conditions are not particularly interesting. Further, if we assume $\sigma = 1.6$ as a "valid" assumption for distribution width then why not assume a mode radius of 85 nm? During background conditions the $r_m = 85$ nm assumption is just as valid as the $\sigma = 1.6$ assumption. Outside background conditions, both assumptions fail and effectively break the utility of the proposed algorithm.

44. Figure 10: I already spoke at length on Figure 10 and I believe this figure provides incontrovertible evidence that the static $\sigma$ assumption is wrong. Please consider plotting the OPC $\sigma$ value as well. Doing so will demonstrate 2 things: 1. a static $\sigma$ value is incorrect, 2. it will show you why your algorithm fails at reproducing the OPC profile (i.e., the information content of the OMPS data is too low and your estimate is being driven predominantly by the changing $\sigma$).

I thank the authors for taking the time to read my comments and look forward to their response.

---

## Community Comment (CC8)

The authors present a methodology for deriving particle median radius and concentration from OMPS-LP color ratio (CR). It's worth noting that this paper appears to be a resubmission of a previous work submitted last year (Wang et al., 2023), with the new version containing somewhat similar content (Wang et al., 2024). The core concept of CR remains consistent with the latest submission (Wang et al., 2024). Figure 1 mirrors the previous version submitted (Wang et al., 2023), albeit with the axes swapped. While utilizing CR seems a reasonable strategy for size distribution retrieval, it's essential to recognize that the OMPS-LP aerosol extinction coefficients at 510, 600, 675, 745, 869, and 997 nm are initially derived under the assumption of a constant Gamma distribution (Taha et al., 2021; Chen et al., 2018) . Consequently, these extinction coefficients primarily reflect background stratospheric aerosols, and using them for number density and concentration retrieval during volcanic eruptions lacks robustness.

Furthermore, a recent study (Bourassa et al., 2023) highlighted that the OMPS-LP NASA (referred to as OMPS(NASA)) extinction coefficient product exhibits a twofold increase compared to the OMPS extinction retrieved by the University of Saskatchewan (OMPS(USASK)) and SAGE III/ISS following the Tonga eruption. Notably, this study (Wang et al., 2024) employs the same OMPS(NASA) product to retrieve median radius and concentration, potentially impacting the derived quantities significantly. A thorough examination of OMPS(NASA) extinction coefficients at 510, 745, 869, and 997 nm, compared with alternative stratospheric extinction products, reveals a notable overestimation by the OMPS(NASA) product. Major concerns regarding the paper are outlined below:

**Major comments**

- Similar to Wang et al. (2023), the authors discuss CR (510/869 nm) and CR (745/869 nm). It appears that the authors utilized CR of 510/869 nm for the retrieval, but the rationale behind showcasing CR for 745/869 nm remains unclear. I am uncertain about the utility of CR of 745/869 nm, given the proximity of these two channels. The crucial point here is the significant bias observed in the 510 nm channel, as evidenced by available SAGE III/ISS measurements at a wavelength of 521 nm, close to 510 nm. Evaluation of the extinction product reveals that the OMPS(NASA) product at the 510 nm channel is unusable due to bias/noise in the data. Therefore, employing the extinction coefficient at 510 nm from OMPS(NASA) itself is not the appropriate approach to compute CR. We conducted an analysis of extinction coefficient data at 510, 745, 864, and 997 nm of OMPS with 521, 756, 869, and 1022 nm of SAGE III/ISS. Figure 1 depicts the zonally averaged stratospheric aerosol optical depth (SAOD) time series percent difference plot of OMPS(NASA) and SAGE III/ISS. It is evident from Figure 1 that the 510 nm channel of OMPS(NASA) exhibits a significant bias, irrespective of volcanic events. In addition to the substantial bias in the 510 nm channel, it is noteworthy that significant overestimation of SAOD is apparent in Figure 1 following the Tonga eruption in all four channels. This corroborates the findings of Bourassa et al. (2023) that OMPS(NASA) overestimates the extinction coefficient at 745 nm by a factor of two following the Tonga eruption. Therefore, users of OMPS(NASA) must exercise caution when using the OMPS(NASA) extinction coefficient product, particularly following volcanic

eruptions. Moreover, the use of the 510 nm channel is questionable, especially given the large bias in Figure 1a, regardless of any events. Additionally, other OMPS retrieval products from OMPS(USASK) and IUP Bremen (hereafter, OMPS(IUP)) are available to the public. It is worth noting that, in addition to the overestimation of aerosol extinction coefficient in OMPS(NASA) following the Tonga eruption, similar differences were noted following the Kelud (13 February 2014) and Calbuco (28 April 2015) eruptions at 745 (864) nm wavelengths when compared against OMPS(USASK) (OMPS(IUP)) products (not shown here). Therefore, following any events that perturb the stratosphere, OMPS(NASA) extinction coefficient product should be used with caution.

[Figure]

Figure 1: Latitude versus time dependence of zonally averaged stratospheric aerosol optical depth (SAOD) percent difference between OMPS(NASA) and SAGE III/ISS at (a) 510 , (b) 745 , (c) 864, and (d) 997 nm. For SAGE III/ISS the respective wavelengths used for computing percent differences are 521, 756, 869 and 1022 nm. Major volcanic eruptions (white) and wild fire events (green) with abbreviated two-letter code with their respective latitude and time of occurrence that are listed here. The event names shown are Canadian wildfire (Cw), Ambae (Am), Raikoke (Rk), Ulawun (Ul), Australian wildfire (Aw), California Creek Fire (Cc), La Soufriere (La), McKay Creek fire (Mc) and Hunga Tonga (Ht).

The authors employ the 510 nm channel to compute CR, leading to a biased CR that inevitably affects the retrieval of median radius and concentration. It's also worth noting that when SAOD is computed between the tropopause and about 21 km, the disparity between OMPS(NASA) and SAGE III/ISS is even more pronounced following the Tonga eruption. This discrepancy arises from the fact that OMPS(NASA) tends to overestimate extinction below the peak of the aerosol layer, as noted by Bourassa et al. (2023), as illustrated in Figure 2c,d. Additionally, OMPS(NASA) release notes (`https://disc.gsfc.nasa.gov/datasets/OMPS_NPP_LP_L2_AER_DAILY_2/summary?keywords=OMPS-NPP_LP-L2-AER`) state that low sensitivity of short wavelengths to aerosols may impact retrievals below 675 nm, advising caution in using LP aerosol extinction data below 17 km and scattering angles greater than 145 degrees

for wavelengths 675 nm or shorter.

While OMPS(NASA) extinction coefficients show improvement towards higher wavelengths compared to the 510 nm channel, it's important to highlight that the OMPS(NASA) extinction coefficients at 745, 869, and 997 nm significantly overestimate following the Tonga eruption and other events. Hence, the robustness of the OMPS(NASA) product, particularly following volcanic eruptions, is questionable.

Furthermore, we evaluated the OMPS(NASA) extinction product at 510 and 997 nm for a relatively unperturbed stratosphere (June 2017) and following the Tonga eruption (April 2024) (Figure 2) using GloSSAC version 2.21 data. The percent difference between OMPS(NASA) and GloSSAC clearly indicates a significant disparity at the 510 nm channel ($> 50\%$) below 24 km, even in June 2017 (Figure 2a). This underscores the bias in the 510 nm channel regardless of any perturbed event. However, the 997 nm channel appears reasonable during June 2017 (Figure 2b). However, following the Tonga eruption, both the 510 and 997 nm channels exhibit significant differences compared to GloSSAC (Figure 2)c,d.

[Figure]

Figure 2: Zonally monthly averaged percentage difference of OMPS(NASA) and GloSSAC (version 2.21) for 525 and 1020 nm on an altitude versus latitude plot. (a, b) for June 2017 and (c, d) for April 2022 following Hunga Tonga eruption.

- Since the 510 nm channel exhibits a significant bias, any CR computation involving this channel will inevitably yield incorrect ratios. Figures 3a and 3b illustrate the percent difference of SAOD between OMPS(NASA) and GloSSAC at 525 nm and 1020 nm, respectively (note that we utilized SAOD at 510 nm and 997 nm to calculate the percent difference with GloSSAC). It's apparent from Figure 3a that the OMPS 510 nm channel consistently exhibits a high bias, regardless of any perturbed event. Therefore, employing the 510 nm channel to retrieve particle size-related quantities would introduce bias and yield unreasonable results. While the

bias in the 510 nm channel persists throughout the record, Figure 3b shows some improvement when comparing the 997 nm channel against GloSSAC's 1020 nm channel. However, there is an overestimation of OMPS(NASA) SAOD following perturbed events such as the Canadian Wildfire, Ambae, Australian Wildfire, and the Tonga eruption.

[Figure]

Figure 3: Latitude versus time dependence of zonally averaged stratospheric aerosol optical depth (SAOD) percent differences from OMPS(NASA) and GloSSAC (v 2.21) (a,b) and SAOD ratios (c,d). OMPS(NASA) SAODs are computed for 510 and 997 nm, while for GloSSAC SAODs are computed at 525 and 1020 nm. Ratio between 510 and 997 nm of OMPS(NASA) AOD is shown in (c), while (d) shows ratio between 525 and 1020 nm of GloSSAC 2.21 for the same time period. Major volcanic eruptions (white) and wild fire events (green) are same as in Figure 1.

We also evaluated the SAOD ratio between the 510 nm and 997 nm channels, as well as between the 525 nm and 1020 nm channels of GloSSAC. This comparison offers insights into how two wavelengths can inform about aerosol extinction coefficient ratios, thereby aiding in the inference of aerosol particle sizes. It's essential to note that we focused on the time series from 2017 through 2022 due to the availability of SAGE III/ISS multi-wavelength measurements from 2017. Figure 3c clearly demonstrates that the OMPS(NASA) SAOD ratios do not provide meaningful information, indicating that these wavelengths cannot be utilized to infer size information. Notably, most of the bias stems from the 510 nm channel. In contrast, the aerosol SAOD ratio from GloSSAC offers valuable insights (Figure 3d) into how ratios change following each volcanic/fire event, particularly after events like the Canadian Wildfire, Ambae, Ulawun, Raikoke, Australian Wildfire, and Hunga Tonga. Each event behaves differently in terms of the aerosol SAOD ratio. For instance, the Canadian wildfire, Raikoke, Australian

wildfire, and Hunga Tonga show a smaller SAOD ratio, suggesting relatively larger particles, while Ambae and Ulawun eruptions exhibit a larger SAOD ratio, suggesting smaller particles.

Consequently, utilizing the OMPS(NASA) extinction product to retrieve size-related information would introduce bias and may not accurately represent the underlying aerosol size information.

**References**

Bourassa, A. E., Zawada, D. J., Rieger, L. A., Warnock, T. W., Toohey, M., and Degenstein, D. A.: Tomographic Retrievals of Hunga Tonga-Hunga Ha'apai Volcanic Aerosol, Geophysical Research Letters, 50, e2022GL101 978, https://doi.org/https://doi.org/10.1029/2022GL101978, URL `https://agupubs.onlinelibrary.wiley.com/doi/abs/10.1029/2022GL101978`, e2022GL101978 2022GL101978, 2023.

Chen, Z., Bhartia, P. K., Loughman, R., Colarco, P., and DeLand, M.: Improvement of stratospheric aerosol extinction retrieval from OMPS/LP using a new aerosol model, Atmospheric Measurement Techniques, 11, 6495–6509, https://doi.org/10.5194/amt-11-6495-2018, URL `https://amt.copernicus.org/articles/11/6495/2018/`, 2018.

Taha, G., Loughman, R., Zhu, T., Thomason, L., Kar, J., Rieger, L., and Bourassa, A.: OMPS LP Version 2.0 multi-wavelength aerosol extinction coefficient retrieval algorithm, Atmospheric Measurement Techniques, 14, 1015–1036, https://doi.org/10.5194/amt-14-1015-2021, URL `https://amt.copernicus.org/articles/14/1015/2021/`, 2021.

Wang, Y., Schoeberl, M., Taha, G., Zawada, D., and Bourassa, A.: Using OMPS-LP color ratio to extract stratospheric aerosol particle size and concentration with application to volcanic eruptions, Atmospheric Measurement Techniques Discussions, 2023, 1–25, https://doi.org/10.5194/amt-2023-36, URL `https://amt.copernicus.org/preprints/amt-2023-36/`, 2023.

Wang, Y., Schoeberl, M., and Taha, G.: Using OMPS-LP color ratio to extract stratospheric aerosol particle median radius and concentration with application to two volcanic eruptions, Atmospheric Measurement Techniques Discussions, 2024, 1–39, https://doi.org/10.5194/amt-2023-267, URL `https://amt.copernicus.org/preprints/amt-2023-267/`, 2024.

---

## Author Comment (AC1)

*Review on Manuscript No. AMT-2023-267*

*Using OMPS-LP color ratio to extract stratospheric aerosol particle median radius and concentration with application to two volcanic eruptions by Wang et al.*

*Wang et al. derive stratospheric aerosol microphysical parameters from OMPS using the color ratio. They discuss various sources of uncertainty and apply their algorithm to measurements from SAGEIII on ISS and validate their method with balloon data and then apply the algorithm to measurements made during two volcanic eruptions.*

*I personally do not have any problems with this study being published. Although the method itself of using two wavelengths for deriving microphysical parameters seems not to be new, I still think this study is of value for the community since it is applied to a different instrument and under different conditions than the ones used in previous studies. Especially when only measurements at two wavelengths are available such a method is clearly of value. However, I agree that the referencing should be done correctly, that the study should be better motivated and that the implications for the community should be more clearly stated. In addition, before resubmission the manuscript should be carefully checked for language and technical correctness. The present version is full of mistakes and could have definitely be prepared with more care.*

**Response:** Thanks for your comments. The new manuscript has revisions using all three reviewer's comments and some comments from the community reviewers as well. The response details can be found below.

*General comments:*

*For me it did not become clear why you do everything with two wavelengths if there are more wavelengths available and one could do everything more accurately without making so many assumptions? Or are the observations at several wavelengths only available from SAGE III and not from OMPS? If you have only measurements at two wavelengths from OMPS I can totally understand why you develop the method you are presenting here. If there however are measurements of three or more wavelengths available from OMPS I would wonder why you make analyzing the data more complicated than necessary.*

**Response:** The OMPS instrument has six channels with a wavelength range narrower than that of SAGE. The three-channel technique used in SAGE measurements relies on the wider range of wavelengths—specifically, 449 nm, 756 nm, and 1544 nm, as selected by Wrana et al. (2021). By comparison, the OMPS range of available wavelengths is 510-997 nm. Figure A1, added to the manuscript, shows that the shorter wavelength range makes it more difficult to derive the distribution width from two OMPS color ratios. Relevant discussions have been added to Section 2.2.

*I think adding subsections describing the OMPS-LP and SAGE III measurements would be quite helpful for the reader. This is definitely missing in the current version of the manuscript. You cannot expect the reader to be an expert on both instruments. Further, I would suggest that you first describe the comparison to the balloon measurements since this is the main comparison (validation) of your method and then present the SAGE III comparison.*

**Response:** We added a paragraph to Section 2.1 to describe the instrument. "As a space-based limb scattering measurement instrument, OMPS-LP on the S-NPP satellite (Flynn et al., 2014; Jaross et al., 2014), launched in October 2011, operates in a sun-synchronous ascending orbit with a 13:30 Equator crossing time. A second instrument was launched aboard the National Oceanic and Atmospheric Administration-21 (NOAA-21) satellite in 2023. Additionally, the OMPS LP is slated to be onboard the next Joint Polar Satellite System satellites, with JPSS-3 scheduled for 2027 and JPSS-4 for 2032. The OMPS-LP sensor utilizes three vertical slits to measure the Earth's limb by pointing aft along the spacecraft's flight path, capturing the sunlit portion of the globe without directly observing the sun. In this study, we selected the central slit. Our algorithm uses the aerosol extinction coefficient (E) from the L2 OMPS-LP products, V2.1 (Taha et al., 2021), which provides E at six wavelengths: 510, 600, 675, 745, 869, and 997 nm. The OMPS-LP aerosol extinction coefficient products have a vertical resolution of 1.6-1.8 km, reported every 1 km, and provides more than 7000 profiles per day."

In the original manuscript, we had a section that uses SAGE data. That section has been dropped. So there is no need to describe the SAGE instrument. In response to the strong opinions expressed in the community comments, we have decided to remove Section 3.1, which compared of OMPS-LP PSD retrievals with SAGE III/ISS. However, we will restore this section if the reviewers disagree.

*Why do you apply your method then to SAGE III? Why is it necessary to have a comparison to another satellite? Could you more clearly state what you try to achieve with this comparison? Couldn't you also use more than two wavelengths from SAGE III to also show what the difference to the more accurate way of deriving aerosol microphysical parameters is?*

**Response:** After considering the suggestions from the community reviewer, we have decided to remove the entire section comparing our OMPS retrievals to the retrievals from SAGE III from this manuscript. While we don't necessarily agree with the community reviewer opinions on the utility of comparing SAGE and OMPS profiles, the confusion about whether SAGE provides independent validation detracted from the overall focus of the paper.

*Specific comments:*

*P1, L15: "....assuming...... width of 1.6". You are not only assuming this with value. You also derive the best result/agreement with this value, thus I think you could/should clearly state this.*

**Response:** Rephrased the sentence to "Our results compare favorably to balloon borne particle size measurements and concentrations under ambient condition and 2019 Raikoke volcanic eruption assuming a log-normal particle size distribution width of 1.6, which also shows the best agreement with this value under these conditions."

*P1, L22: Instead of citing the text book from Seinfeld and Pandis (2016) I would suggest to cite Brock et al. (1995) and/or Kremser et al. (2016). Note, stratospheric (sulfuric acid) aerosols form mainly in the tropical upper troposphere (see Brock et al., 1995). Please rephrase/correct the sentence.*

**Response:** We have updated our references to cite more recent measurements by ballon and lidar soundings of stratospheric aerosols. Brock et al., (1995) is included.

*P1, L28: What is meant with "black" and "brown" aerosols? Please explain this to the reader. To my knowledge these terms are only used for tropospheric aerosol (e.g soot containing aerosols).*

**Response:** Basically, non-sulfate aerosols provide condensation centers for sulfate. Thus the aerosols are not transparent liquids but are also absorbing.

*P1, L28: What is meant with "self loft"? Without explanations this sentence is useless. I would suggest to omit this sentence.*

**Response:** Omitted.

*P2, L34: What about SAGE and SAGE II? I think a sentence about this two instruments should be added.*

**Response:** We have slightly expanded our discussion of SAGE I, II.

*P2, L37-38: Is the stellar occultation technique also used to measure aerosols? Without any further explanations this sentence is useless and could be omitted.*

**Response:** Deleted it.

*P2, L51: Are there currently or in the future other missions measuring aerosols in the stratosphere? Or will OMPS be the only one irrespective of the measurement technique? When was OMPS launched and how long are the measurements expected to continue? Please add here some more information.*

**Response:** Currently, there are two OMPS-LP instruments, one onboard S-NPP, and NOAA-21, with further two to be launched in 2027 and 2032. We added this to section 2.1.

*P2, L63-P3, L64: Without any further explanation this sentence is useless. What is the difference between your algorithm and the one from Thomason and Vernier (2013)? What are the differences between the algorithms? What has been changed?*

**Response:** The basic idea behind the algorithm is the same, but we are applying the technique to limb scattering measurements. The sentence has been modified to make that clear.

*P3, L69: Add here a section or sections describing the instruments/data used in this study (SAGE, OMPS, balloon)*

**Response:** We added a paragraph to Section 2.1 about the OMPS dataset.

*P3, L77: What are N, r_0 and s are. This has not been explained.*

**Response:** We added the explanation to the eq (1a).

*P3, L81: Why in this study? How is the color ration defined in other studies? Isn't the color ratio always defined the same way?*

**Response:** Yes, the color ratio can be any two wavelengths divided by each other. Writing it this way is intended to be more clear.

*P3, L83: There is no Ångström exponent in Equation 1b.*

**Response:** We have added a discussion of the relation of CR to the Ångström exponent .

*P3, L8: Which wavelengths combinations are available and which ones have been used?*

**Response:** Any two wavelengths can be used to calculate the color ratio. In this study, we selected the ratio of 510 nm to 869 nm. The relevant information has been added following Equation (2). The 997nm can also be used, but it does not extend over the whole OMPS measurement period.

*P4, L116: How "large"? Please give some numbers.*

**Response:** The large aerosol particles are > 0.4 μm. We added it to there.

*P4, L121: Why? Why this value? Do you derive this value from Fig. 1? Provide figure or reference for this value.*

**Response:** The color ratio of clouds is around of 1. Based on Figure 1 and Figure 9 (similar as Figure 3 in Schoeberl et al., 2021), we select color ratio of 1.1 as the threshold.

*P5, L134: Using a log-normal distribution is quite common for stratospheric aerosols and thus here you should instead of a specific paper rather cite a textbook as e.g. Seinfeld and Pandis (2016).*

**Response:** Since log-normal is so common, citing a text book seems appropriate to us.

*P6, L176: If the error is large for small radii wouldn't this then cause difficulties when plume events are considered where many small particles are produced?*

**Response:** Based on Figure 4, the affected aerosols have a radius smaller than 0.03 μm, which is smaller than typical aerosols even under ambient conditions.

*P6, L185-186: In both sentences you cite Chen et al. (2018). One of these citations could be skipped and the grammar of the second sentence should be corrected.*

**Response:** We rephrase to "This approach is similar to that used by Chen et al. (2018). OMPS-LP aerosol retrieval algorithm assumes a gamma aerosol size distribution with fitted parameters of $\alpha = 1.8$ and $\beta = 20.5$, resulting in effective radius $r_{eff}$ of 0.185μm."

*P7, L21-202: Not necessarily. You will have both small and large particles. If you make such a statement you should give a reason and/or provide a reference.*

**Response:** The initial eruption will include large particles, but as SO2 oxidizes and sulphate aerosols form, the size distribution shifts to small particles that slowly grow. This evolution is shown in ballon measurements and 2-color ratio retrievals in Wrana et al. (2023). PyroCBs show a more complex size distribution as shown by Fromm et al, (2008).

*P7, 208ff: I would suggest to change the order and to first describe the validation with the balloon data and then the verification with SAGE. If you want to keep the current order you should provide a motivation or reasoning why this order is more logical.*

*P8, L227: Which wavelength pair has been chosen and why? How would the result differ if a different wavelength pair would be used?*

*P8, L231: Be more precise. Cleary state above which altitudes.*

*P8, L235: "matched at all altitudes". This is not correct. You do not have a match at all altitudes. Give the altitude range. Further, I see here a better agreement for particle size than for number density.*

**Response:** Regarding to the comments from the community reviewers, we have decided to remove the entire section comparing our OMPS retrievals to the retrievals from SAGE III from this manuscript

*P9, L279: If this method does not give you information on bimodal distributions, isn't than that a significant drawback for investigating volcanic plumes? Especially shortly after the eruption the distribution will be bimodal for a certain time. Could you give an estimate in percent how large the deviation for the derived microphysical parameters are?*

**Response:** Yes it is a drawback as implied in our discussion. The percent estimate is highly dependent on the actual distribution e.g. the two modal radii and the width of each mode.

*P9, L281: "....... Because the extinction is mostly due to fewer large particles.....". Sentence not clear. Please rephrase.*

**Response:** Revised.

*P10, L296: What is meant with "concentration"? A high number of measurements with extinction coefficients 4x10-4 km-1?*

**Response:** Yes. We rephrased that sentence.

*P10, L298: Which sizes? Give some numbers.*

**Response:** Added 1.e-4

*P10, L300: Also here. Give a number. How small?*

**Response:** Deleted.

*P10, L309-310: This is not clear. What has the self-lofting of the plume to do with the composition?*

**Response:** The composition describes the plume in the previous sentence, and the sentence about self-lofting is meant to clarify that the plume may extend above 18 km in height. We rephrased the sentence to avoid any misunderstanding.

*P10, L313: Give a number.*

**Response:** Sentence revised to "The aerosol particle radius and concentration plot (Fig. 11d) shows a shift toward larger particles, with sizes above 0.1 µm and with a concentrations > 50 particles/cm3 compared to background."

*P11, L316: Is the eruption equal to day 0? If yes, I would write or state this more clearly, e.g. eruption = day 0.*

**Response:** Thanks. We take your suggestion and revised the sentence.

*P11, L319: at a later time -> add when exactly*

**Response:** We revised to "The eruption cloud is initially at 50° N, and as it moves southward the aerosols are detected at more southerly latitudes approximately a month later (Gorkavyi et al., 2021; Boone et al. 2022)."

*P11, L323: at higher altitudes -> at which altitudes. Add a number.*

**Response:** Added. Revised to "At all latitudes after day 80, the particle radius at higher altitudes (i.e., above 15 km) decreases consistent with gravitational settling (e.g English et al., 2013)."

*P11, L323-324: the settling or sedimentation process is several times mentioned, but never explained.*

**Response:** Gravitational settling is a well known process by which aerosols are removed from the stratosphere. Reference added.

*P11, L326-327: Why referring here to other studies? Why don't you include a figure showing the same from SAGE?*

**Response:** Our goal of this study focus on OMPS-LP measurements, so we refer to other studies. In addition, in this revision, we deleted the materials that using SAGE measurement in this manuscript.

*P11, L328-329: This result is for this study not important and thus, the sentence does not make any sense here.*

**Response:** Deleted.

*P11, L333-334: Why is the impact of the distribution width limited? The evolution of the width could be simulated with a box model. I assume that the small particles will quickly disappear (within a few days). You could check the literature for modelling studies of volcanic eruptions. I guess you could find there some numbers how quickly the distribution is back to the background distribution.*

**Response:** This paper (English et al., 2013) shows the small particles spend a couple of months to settle.

*P12, L352: Sentence not clear. Check and rephrase. Further, I could not find your statement in Wrana et al. (2023). How did you derive this width? From the figures?*

**Response:** We corrected the reference. Here is the revision. "Based on the in situ measurements from Baron et al., (2023), we assume a log-normal particle size distribution width of 1.2 for this eruption event, which is also consistent with SAGE III/ISS measurements (Duchamp et al., 2023)."

*P12, L365: Don't just write larger. Give numbers.*

**Response:** The settling rate increases with particle diameter according to the Stokes parameters. Since our retrievals are sensitive to particles < 0.5 µm larger means at the large end of the retrieval.

*P12, L366: Which settling rate? You mean the sedimentation rate of the particles. Do you mean that particles < 0.5 µm sediment out?*

**Response:** It means the aerosol plume setting rate is equivalent to the downward gravitational settling of aerosols with diameters of 0.5 µm. We revised that sentence to "Both Legras et al. (2022) and Schoeberl et al. (2022) argue that the aerosol distribution settling rate is characteristic of 0.5 µm or larger particles when assuming downward gravitational settling.".

*P12, L374-375: Sentence not correct. Please check and correct.*

**Response:** Revised to "The scattering angle variation range of 85o - 130° during the analyzed time period (Figure A6) indicates a less than 15% uncertainty for a 40% phase function error."

*P13, L378-382: This paragraph is giving a motivation for the study and thus this paragraph should rather appear in the introduction than in the summary and conclusion section.*

**Response:** Sentence moved to introduction.

*P13, L387: Clearly state here that you refer here to log-normal distributions with one mode.*

**Response:** Added.

*P13, L390-391: The two sentences should be rephrased and maybe combined. Add also why a CR of 1 is problematic.*

**Response:** Revised to "The algorithm cannot distinguish radii greater than ~0.4 µm because their color ratio (E510/E879) approaches one. Such particles are usually identified as large aerosols or aerosols mixed with clouds (Schoeberl et al., 2021)."

*P13, L403-405: I think before you stated the opposite. Further, the second sentence starting with "We examined......." is incomplete.*

**Response:** Revised to "We examined how variations in retrieved aerosol radius change with different assumptions about distribution width by comparing to balloon measurement."

*P13, L406-408: Add more details. Under which conditions does this happen?*

**Response:** We added details as follows "The size range can vary from 0.05 to 0.25 µm by adjusting the distribution width from 1.2 to 1.8, assuming a particle size of 0.1 µm with a distribution width of 1.6."

*P14, L420: What are the future implications? For what can the method be applied? The future OMPS measurements? Nevertheless, during volcanic eruptions you have large uncertainties and you need to rely on other studies to derive your input values?*

**Response:** As discussed in the introduction, this study focuses on testing whether OMPS-LP measurements can be used to retrieve particle size and number density. The results are reasonable for background conditions and the 2019 Raikoke volcano. However, for the 2022 Hunga-Tonga eruptions, the distribution width needs to be updated based on additional inputs.

*Reference list: Check the style. Some journal names are written out. others not. Same with the author names. In some cases the entire first name is written, in most other cases the first initial.*

**Response:** Thanks. Updated.

*Check the ACP guidelines and prepare your reference list accordingly. Further, there are some references misplaced as e.g. Kremser and Yue and Deepak. The Box and Deepak paper is listed in the reference list, but not cited throughout the manuscript.*

**Response:** Updated.

*Figure 9 and 10: Add a legend to the figure. Is there no uncertainty range for the CPC data given or are these so low that these are not visible? What are the black zigzag lines on the right and left corners of the figure?*

**Response:** We explained the two colored lines in the caption. The uncertainty range for the CPC data is too low to be visible. The black zigzag lines represent one of the bi-modal distributions.

*Figure 11: Wouldn't it make much more sense to compare the background at 30° -60°N with the distribution during Raikoke at 30°-60°N and the background at 30°S-15°N to the 30°S-15°N distribution during Hunga-Tonga? I really think the same regions should be compared.*

**Response:** Under background conditions, the aerosol retrievals are not significantly different, so we selected the globe to display the results.

*Figure 14: I think it would be better to use black instead of red lines. Also omitting some lines would be helpful, e.g is the 26 km line really necessary? I think this one could be omitted. Why is here the eruption on day 15 and not on day 0 as before?*

**Response:** Any color choice is going to have a downside.  Black lines would disappear at the top the figure, for example. We think the lines are visible enough.  The day number is the day number of the year in the Hunga case.

*Technical corrections we have also made:*

*P1, L20: Introduce the abbreviation "Cb".*

*P1, L20: Reference of Kremser et al. (2016) is missing in the reference list.*

*P2, L42: Full stop before reference of Taha et al. (2021) obsolete.*

*P2, L57: PyroCB -> pyroCb. Use a consistent writing.*

*P3, L65: In the next section, we detail -> In the next section we describe in detail*

*P3, L79: scattering measurements -> scattering measurement*

*P3, L80: is the same as Wrana et al. (2021) Eq. 2 -> is the same as Eq. 2 in Wrana et al. (2021)*

*P4, L97: Add "the" -> we simulate the scattering....*

*P4, L103: show the how CR -> show how the CR*

*P4, L118: Add "a" -> We found that selecting a CR*

*P4, L119-120: Check sentence and correct sentence.*

*P5, L137-138: Either use singular or plural.*

*P6, L171: PyroCB -> pyroCb*

*P6, L186: Comma obsolete (reference of Chen et al.)*

*P6, L188: Comma obsolete.*

*P7, L197: "CR (510/869)" here you write it without adding nm. In other occasions nm is added. This should be done more consistently throughout the manuscript.*

*P7, L210: using the same algorithm -> the above described algorithm*

*P8, L246: comparation -> comparison*

*P8, L246: Add "as shown" -> during Raikoke volcano eruption as shown in Fig. 8*

*P8, L248: Add "the" -> the bias*

*P8, L249: Add "the" -> "the comparisons" and move references at the end of the sentence.*

*P9, L275: settle out -> sediment*

*P10, L283: Use comma instead of writing twice "and" -> Background Aerosol Radius, Concentration and Volcanic Perturbations.*

*P10, L285: Add "conditions" so that it reads "background conditions"*

*P10, L288: Thompson -> Thomason*

*P10, L287: Move "Figure 11" before "(a, c, e)" so that it reads "The extinction coefficient vs CR is shown In Fig. 11 (a, c, e) and radius vs concentration distributions for these three situations in Fig. 11(b, d, f).*

*P10, L291: Put a, c, e in parenthesis -> (a, c, e)*

*P10, L296: One "shows" obsolete. Write "background aerosols" instead of just "background".*

*P10, L301: aerosols -> aerosol and density -> densities*

*P10, L302: Section 3 -> Sect. 3*

*P10, L 306: Comma before reference of Gorkavyi obsolete.*

*P11, L318: screen -> screening*

*P11, L320: add "altitude" after 10-15 km*

*P11, L327: add "is" and particles should read particle -> Our result is also consistent with the larger particle radius......*

*P11, L332: After the comma "The" should be "the".*

*P12, L350: Feb.-March -> February to March*

*P12, L353: Add "is"-> which is consistent*

*P12, L355: Delete "is" before represents.*

*P12, L372: Add "the"-> the radius*

*P12, L373: Add "the"-> the aerosol radius*

*P12, L372: Add "altitude"-> the altitude range*

*P12, L374: Add "of"-> conversion of SO2*

*P13, L390: radius -> radii*

*P14, L416: Delete "the" before "median radius".*

*Figure 2 caption: Add "nm" after 501/869. Use a consistent writing style throughout the manuscript.*

*Figure 5 caption: plot -> Plot and add space before and after "="*

*Figure 6 caption: Add space before and after "="*

*References:*

*Brock, C. A., P. Hamill, J. C. Wilson, H. H. Jonsson, and K. R. Chan (1995), Particle formation in the upper tropical troposphere: A source of nuclei for the stratospheric aerosol, Science, 270(5242),1650–1653, doi:10.2307/2887916.*

*Kremser, S., et al. (2016), Stratospheric aerosol—Observations, processes, and impact on climate, Rev. Geophys., 54, 278–335, doi:10.1002/2015RG000511.*

*Response: added these references*

**Response:** Thank you so much for your careful review. We have corrected the text as suggested.

References:

Chen, Z., Bhartia, P. K., Loughman, R., Colarco, P., and DeLand, M.: Improvement of stratospheric aerosol extinction retrieval from OMPS/LP using a new aerosol model, Atmos. Meas. Tech., 11, 6495–6509, https://doi.org/10.5194/amt-11-6495-2018, 2018.

English, Jason M., Owen B. Toon, and Michael J. Mills. Microphysical simulations of large volcanic eruptions: Pinatubo and Toba, J. Geophys. Res. Atmos., 118.4, 1880-1895. https://doi.org/10.1002/jgrd.50196, 2013.

Fromm, M., Shettle, E.P., Fricke, K.H., Ritter, C., Trickl, T., Giehl, H., Gerding, M., Barnes, J.E., O'Neill, M., Massie, S.T. and Blum, U., Stratospheric impact of the Chisholm pyrocumulonimbus eruption: 2. Vertical profile perspective, J. Geophys. Res. Atmos., 113.D8, https://doi.org/10.1029/2007JD009147, 2008

Wrana, F., von Savigny, C., Zalach, J., and Thomason, L. W.: Retrieval of stratospheric aerosol size distribution parameters using satellite solar occultation measurements at three wavelengths, Atmos. Meas. Tech., 14, 2345–2357, https://doi.org/10.5194/amt-14-2345-2021, 2021.

Wrana, F. U. Niemeier, L. W. Thomason, S. Wallis, and C. von Savigny: Stratospheric aerosol size reduction after volcanic eruptions, Atmos. Chem. Phys., 23, 9725–9743, https://doi.org/10.5194/acp-23-9725-2023, 2023

---

## Author Comment (AC5)

We are combining responses to the two community comments CC8 and CC5 below.  All three community comments come from the SAGE III team.

The gist of the CC8 comment is that the 510 nm channel is not very accurate compared to SAGE III (figures were attached) and CC5 makes a similar point. The color scale in the CC8 figures seems to indicate that all the channels compare poorly to SAGE with errors between 20-50 %.  Taha et al. (2021) Fig. 6 also shows the relative bias between SAGE and OMPS extinction which indicates a dependence in altitude as well as latitude. In the Taha figure, the agreement with SAGE is within ±25% above 18 km and over most of the latitude range. The 510 channel performs about the same as the other short wavelength OMPS channels.  Agreement with GloSSAC is better for channels at 745 nm and longer.

CC8 shows SAOD plots of the ratios of 510/997 and GloSSAC 535/1020. First, in our paper, we used the aerosol extinction rather than the reported SAOD. Furthermore, we restricted the extinction used to above 20 km altitude in the southern hemisphere (sh) tropics (Figure 14), as recommended by Taha et al. (2021).  The SAOD ratio in Figure 1 (from CC8) is wrong in the southern hemisphere and the tropics because the 510 nm useful altitude range is restricted to above 20 km (see Figure 2).  Note that the 869nm is mostly accurate for all stratospheric altitudes and latitudes.

[Figure]

Figure 1. A copy of Figure 3 (c) and (d) from CC8.

The argument CC8 makes is that the two SAODs are unlike each other.  The SAOD color ratio of ~ 5 for OMPS whereas the color ratio for GloSSAC is closer to 3.   The high OMPS ratio SAOD after Hunga is unlike the GloSSAC ratio. Note that we do not use the wavelengths shown in CC8's figure.

Below is a redo of the above plot using our own color scale and our wavelengths, and restricting the plot to 0°-60° N to avoid the anomalous 510 nm retrieval in the SH below 20 km. OMPS SAOD is computed by integrating from 2 km above the tropopause (to avoid clouds) to 40 km.  The SAOD color ratio between the OMPS and GloSSAC data sets is now more consistent except at high latitudes where OMPS appears to have lower values. This may be caused by the limited coverage of SAGE III at these latitudes compared to OMPS.  If

we use the wavelength ratio shown in CC8 (510/997) our results closer to the reviewer's. Notice that the SAOD shown here is only for comparison. For accurate color ratio, it is recommended to use the extinction coefficient at altitude ranges deemed accurate by (Taha et al., 2021) as we did in our paper.

The reviewer's point is that the paper should have more discussion of uncertainties in the OMPS extinction, and we concede the point. In the paper revisions, we add more discussion on the differences.

[Figure]

Figure 2. The figures are analogous to Figure 3 (c) and (d) from CC8, but with the extinction coefficient used to compute the color ratio, and the wavelengths adjusted to match those in our study. The colorbar scale has also been updated accordingly.

The CC5 reviewer makes two important points which we abstract below:

The first point has to do with available information in the OMPS extinction measurements. These issues are summarized in Knepp et al. (2024). Basically, there is insufficient information to accurately determine the concentration as well as characterize the PSD mode radius and distribution width with the OMPS wavelengths. This is fair point. Wrana et al. (2021, 2023) and Duchamp et al. (2024) retrieve a median radius and log-normal distribution width using two SAGE color ratios, the second employing the 1.543 extinction. Because of the narrower OMPS wavelength range OMPS wavelengths, we cannot retrieve both $r_m$ and s.

To estimate the log-normal size distribution width Knepp et al. (2024) takes a different approach from Wrana et al. (2022). Knepp varies log normal size distributions creating a solution space. Then they extract and equivalent radius and distribution width from the

color ratios from the observed extinctions.  This approach also yields a measure of uncertainty. We downloaded Knepp's files and generated a histogram of distribution widths for data between 45N and 45S between 20-30 km and before 2022 (to avoid Hunga). The normalized distribution of sigma values is shown below, the vertical line is the mean value.

[Figure]

Figure 3. The normalized distribution of sigma values (distribution widths) for data obtained from Knepp et al. (2024) within the latitude range of 45°N to 45°S and altitude range of 20-30 km, prior to 2022 (to exclude the Hunga event). The vertical line represents the mean value.

The Knepp et al retrievals suggest that assuming 1.6 probably widens the distribution too much, and a universal value is hardly justified.

Our approach is related to Knepp's in that we are trying to estimate the error by varying the distribution width. We first assume that we can extract $r_m$ from the color ratio.  As pointed out by CC3 and CC8, there is difference in the OMPS extinction compared to SAGE.  This uncertainty is reported in Taha et al. (2021) and noted in our text. CC8 argues that we should expand the discussion of the errors and we have done so.

In the discussion, the reviewer objects to using a PSD width of 1.6.  Under ambient conditions there is plenty of observational data suggesting that a width value of 1.6 is reasonable (see Wrana et al., 2021, Fig. 6). The figure (above) we generated from Knepp's files supports a slightly smaller value for s, but 1.6 is not unreasonable. We certainly agree that under volcanic conditions, however, the width may be much smaller as determined by Duchamp et al. (2023).  Using SAGE data, Duchamp retrieved Hunga particles with $r_m$ =0.35µm.  This agrees with our results using OMPS - our Fig 14.  Duchamp used the method developed by Wrana et al. (2021).

Validation of the OMPS product is difficult since few balloon measurements intercept OMPS locations exactly.  Aside from criticizing the lack of coincidences, the reviewer doesn't provide alternatives to validating the data. A comparison with Knepp's analysis is

beyond the scope of this current work, but we have started to make comparisons with Knepp's analysis (see above), and that will be the subject for a future paper.

We have gone through the other specific comments and made appropriate changes.

---

## Author Comment (AC6)

**Responses to one of the community reviewer's comments (CC5)**

We appreciate the feedback from all three community reviewers, who are all members of the SAGE III/ISS team. Their comments predominantly imply that SAGE III/ISS is the only satellite instrument capable of providing stratospheric aerosol particle size distribution (PSD) data. While we recognize the critical role of SAGE III/ISS, this perspective overlooks the need for alternative sources of PSD data beyond its operational lifetime, which is expected to conclude with the decommissioning of the ISS in 2030.

Additionally, the reviewers do not specifically address the PSD results presented in this paper, which align well with published literature despite larger uncertainties, inherent to the measurements.

The OMPS LP instrument stands out as a promising (if not the only) candidate for continuing PSD observations in the coming decades. This paper represents the first effort to leverage OMPS LP's multi-wavelength capabilities to derive PSD, a contribution we hope will encourage further refinements, both by our team and the broader community, to address the anticipated data gap once SAGE III is no longer operational or lacks coverage.

The following are our point-by-point responses to the community CC5, with his comments copied and pasted in italics.

*Overview*

*The authors present a revised version of a paper that was originally submitted in 2023 (https://amt.copernicus.org/preprints/amt-2023-36/). Unfortunately the current incarnation of this manuscript fails to provide substantive changes from the previous submission and the authors fail to address any of the major challenges that were raised. To date, the methodology they present remains fundamentally flawed (this claim is supported by figures that the authors provide in their manuscript and will be expounded below), and the authors continue to gloss over these critical aspects without warning the reader of the impact of their assumptions. Rather, the authors "sup- port" their assumptions by incorrectly citing various papers, ignoring other key papers that refute their assumptions, and ignoring the plethora of in situ data that invalidates their assumptions. An obvious case of this last point can be observed in their Fig. 10. If the authors were to plot the OPC-measured distribution width (herein referred to as either "distribution width" or σ) they would show the reader the large degree of variability σ has within a single profile (for this flight σ varied from 1.18 to 2.45). This plot alone would be enough to tell the reader that a static distribution width is not only ill advised, but wrong. Instead, this point is ignored throughout the text and the reader is left with an incorrect impression of the integrity of this method.*

**Response:** We appreciate your detailed feedback from the community reviewer. We understand that your primary concern is with our use of a static distribution width (σ) in our methodology. We have addressed this issue by performing a comprehensive uncertainty analysis, as presented in Section 2. This analysis was introduced in response to recommendations from earlier reviewers.

We acknowledge that some references were incorrectly cited in the initial submission. This was unintentional, and we have corrected all the errors highlighted by the reviewers. If there are any additional references you believe are still incorrect, we would appreciate it if you could point them out, and we will address them accordingly.

Regarding Figure 10, we would like to clarify that our study focuses on retrieving the aerosol particle median radius rather than the distribution width (σ). While we recognize the importance of the variability in σ, it falls outside the scope of our current study to compare this parameter directly with in-situ measurements. However, we are planning to address this aspect in a future study, which will specifically investigate the variability of σ under different stratospheric conditions.

We agree that a static distribution width introduces greater uncertainties compared to a dynamic approach. We have included a detailed explanation in the manuscript discussing our rationale for assuming a static distribution width for the OMPS-LP data. The main reason is the limited spectral range of the OMPS-LP measurements, which makes it challenging to distinguish variations in σ accurately.

*This is a major problem with this method: using a static distribution width inevitably forces the rm to a specific subset of the solution space which may, or may not, be close to the actual atmospheric conditions. The problem is, given this method, we will never know whether the assumed σ value was right or wrong (and, by extension, we will have no confidence in the inferred rm estimate) unless we happen to be fortunate enough to have a coincident OPC flight that can be used to inform the authors' algorithm and confirm the output (if the authors were to provide a rigorous validation of their algorithm then we could certainly have more confidence). The authors demonstrate this by shifting their σ value from 1.6 to 1.2 for the HTHH event; how would they know to do this without the OPC data? Further, the σ in that profile changed, substantially, throughout the profile (again, see their Fig. 10). Shifting their assumed σ from 1.6 to 1.2 made it a better guess in parts of the profile, but a far worse guess in other parts of the profile (again, without the OPC data, and in the absence of a robust validation effort, how would we know where the assumed σ is "good" or "bad"?). In short, even when we have OPC data to inform our assumptions we must be very careful on how these assumptions are used because the atmosphere, especially after eruptions and/or wildfires, is not homogeneous.*

*Regarding the static distribution width, one may wonder how the authors justify their value (σ =1.6). They repeatedly, and incorrectly, refer to the Wrana et al. 2021 (https://doi.org/10. 5194/amt-14-2345-2021) paper. I can only speculate how they misinterpret that paper (perhaps it is a misinterpretation of the Wrana et al. 2021, Fig. 4, which is for a single profile). The problem is that Wrana et al., 2021 did not provide a statistical evaluation of the distribution of σ values (or any other size distribution parameter). However, Wrana et al. did state "The upper and lower boundaries of the colour bars in Fig. 6 roughly mark the ranges within which the values of the respective quantities fall for the data set between June 2017 and December 2019. The median values for this time frame at 20 km altitude are 130.6 nm for the median radius, 1.54 for the mode width σ,..." Granted, the median σ value, at 20 km, from Wrana et al. 2021 was 1.54 (close to the authors' value of 1.6), but the authors must recognize that the color bar scale for σ (Fig. 6 of Wrana et al. 2021) extended from ≈1.1 to 2.0. This is a direct refutation to the claim that Wrana et al. 2021 supports the selection of a static σ of any value.*

*The authors make no mention of the bias in the OMPS extinction products under aerosol loading as describe in Bourassa et al., 2023 (https://doi.org/10.1029/2022GL101978), the so-called 1-D assumption and convergence problems. Bourassa et al. 2023 showed that, when the stratosphere is volcanically perturbed, the extinction is a factor of 2 too high at the aerosol peak, while, below the peak, the extinction product is a factor of 2 too low. While Bourassa et al. evaluated the performance of the 755 nm channel the conclusion of the paper is clear: the bias is inherent in the overall retrieval methodology and there is no reason to believe this bias is limited to a single channel (we note that the University of Saskatchewan's tomographic retrieval method alleviates, but does not obviate, this bias). Work done in our group (unpublished, but Mahesh Kovilakam suggested that he will provide figures in his review of this paper) indicates that this bias is not constant across wavelengths, but increases as you move to shorter wavelengths; hence, the bias does not cancel out in the ratio.*

*Finally, the authors fail to provide a convincing validation of their new product, which is essential. They provide a comparison with 2 balloon-based in situ measurements and declare both to be in "reasonable agreement" (line 281). However, the agreement in their Fig. 10, within the main aerosol layer, was off by at least a factor of 3 (i.e., the OPC measured rm remained steady at ≈50 nm from 16–21 km, while the OMPS method yielded rm estimates that ranged from ≈150 to >300 nm) and the profile shapes are not even close to being similar. If anything, this "validation" exercise proved the opposite of what they intended (i.e., it showed quite clearly that their product is not reliable outside background conditions). Can the authors justify calling this "reasonable agreement".*

**Responses:** The first point has to do with available information in the OMPS extinction measurements.  These issues are summarized in Knepp et al. (2024).  Basically, there is insufficient information to accurately determine the concentration as well as characterize

the PSD mode radius and distribution width (σ) with the OMPS wavelengths. This is fair point. Wrana et al. (2021, 2023) and Duchamp et al. (2024) retrieve a median radius and log-normal distribution width using two SAGE color ratios, the second employing the 1.543 extinction. Because of the narrower OMPS wavelength range, we cannot independently retrieve both $r_m$ and s.

To estimate the log-normal size distribution width Knepp et al. (2024) takes a different approach from Wrana et al. (2022). Knepp varies log normal size distributions creating a solution space. Then they extract and equivalent radius and distribution width from the color ratios from the observed extinctions. This approach also yields a measure of uncertainty. We downloaded Knepp's files and generated a histogram of distribution widths for data between 45N and 45S between 20-30 km and before 2022 (to avoid Hunga). The normalized distribution of sigma values is shown below, the vertical line is the mean value.

[Figure]

Figure 1. The normalized distribution of sigma values (distribution widths) for data obtained from Knepp et al. (2024) within the latitude range of 45°N to 45°S and altitude range of 20-30 km, prior to 2022 (to exclude the Hunga event). The vertical line represents the mean value.

Our approach is related to Knepp et al. (2024) in that we are trying to estimate the error by varying the distribution width. We first assume that we can extract $r_m$ from the color ratio. As pointed out by CC3 and CC8, there is difference in the OMPS extinction compared to SAGE. This uncertainty is reported in Taha et al. (2021) and noted in our text. CC8 argues that we should expand the discussion of the errors and we have done so.

The Knepp et al (2024) retrievals suggest that assuming 1.6 probably widens the distribution too much, and, we agree with the reviewer that a universal value is hardly justified. Under ambient conditions there is plenty of observational data suggesting that a width value of 1.6 is reasonable (see Wrana et al., 2021, Fig. 6 and discussion in Reiger et al. 2014; Table 1). The figure (above) we generated from Knepp's files supports a slightly

smaller value for σ, but 1.6 is not unreasonable. We certainly agree that under volcanic conditions, however, the width may be much smaller as determined by Duchamp et al. (2023). Using SAGE data, Duchamp retrieved Hunga particles with $r_m$ =0.35μm. This agrees with our results using OMPS when we adjust σ - our Fig 14. Duchamp used the method developed by Wrana et al. (2021).

Validation of the OMPS product is difficult since few balloon measurements intercept OMPS locations exactly. Aside from criticizing the lack of coincidences, the reviewer doesn't provide alternatives to validating the data. A comparison with Knepp's analysis is beyond the scope of this current work, but we have started to make comparisons with Knepp's analysis (see above), and that will be the subject for a future paper.

*The authors go on to provide what they call an "evaluation" of their product by running OMPS and SAGE III/ISS data through their algorithm to determine how the derived rm values differed (e.g., their Figs. 7 and 8). What they fail to recognize is that the agreement is foreordained from the outset and that any agreement they show here is really a convoluted comparison of the 2 instruments' extinction ratios. This in no way provides an evaluation of their algorithm. If the input numbers are similar then it is impossible for the output to differ.*

*The authors miss an opportunity to compare their product with the recently release SAGE III/ISS particle size distribution (PSD) parameters (Knepp et al., 2024; https://doi.org/10. 5194/amt-17-2025-2024). I do not fault them for this as our product was only made available within the last few months. Further, I do not suggest this as a shameless self promotion of our work, but the SAGE PSD parameters provide a unique data set that provide uncertainty estimates for the PSD parameters, all of which could be used to evaluate the performance of their algorithm. Unfortunately, we already know how their algorithm will perform since we did a similar evaluation of deriving PSD parameters from SAGE II (i.e., using a single extinction ratio of 520:1020 nm). This is shown in section 6.3 of Knepp et al. 2024 with the statistics summarized in Table 9 of that paper. What we observed is the rm estimates were off by up to -124%, -95%, -28%, 3%, and 21% at the 10th, 25th, 50th, 75th, and 90th percentiles. In short, 80% of our estimates were between 124% too low to 21% too high. There simply is not enough information within 2 channels (i.e., a single extinction ratio) to make an accurate estimation of rm.*

**Responses:** In response to the strong opinions expressed in your above comments and all community comments that all from your SAGE III/ISS team, we have decided to remove Section 3.1, which compared of OMPS-LP PSD retrievals with SAGE III/ISS. Whereas we did make it clear that comparison to SAGE was not validation but algorithm verification, CC5

and other community reviewers also misconstrued the point. Since other readers will likely also be confused we removed the section.

*Finally, all papers require a purpose, the "so what" factor. After reading it, we should be able to answer the question "What did this paper tell us that is new?" or "How is the community now smarter?" The authors fail to deliver on this point.*

**Responses:** This an opinion and not an actual comment on the methodology or results in the paper, but we will address these questions further below.

*Herein, the authors put a new dataset (OMPS color ratios) through an aerosol size algorithm that was developed more than 40 years ago and this methodology is as limited in information content now as it was then.*

**Responses:** We are well aware that color ratio has been used before and added references as such. However, overall application of this technique to the OMPS-LP data set has not been done before. This is new research. Also new is the sensitivity of the retrievals to assumptions about the distribution widths.

*What did we learn? The method is not new and we do not gain any new insights into the atmosphere. Since the method is not new I would expect, at the bare minimum, a thorough demonstration of the validity of this method (i.e., a robust validation), but instead we get 2 comparisons with OPC profiles and an ill conceived evaluation with the SAGE III/ISS instrument.*

**Responses:** As we noted above the comparison with SAGE was verification of the algorithm not validation, and we clearly stated that in the text. However, we have removed that section. There is little truly independent validation data available, and we have made use of what we have. Note that SAGE II aerosols were validated by OPC predecessor instruments (Wang et al.,1989) using a similar approach and just seven coincidences. Specific to the validation shown in the paper, the comparisons with balloon-borne OPC measurements are crucial, especially since they include analyses under two different conditions, ambient background and post-volcanic eruption scenarios. These comparisons provide a good understanding of the algorithm's performance across varying atmospheric conditions. While the validation is not exhaustive, we believe it is a meaningful demonstration of the method's capability and lays the groundwork for further validation studies.

*If the authors were to make this product available to the community it could not be used without the end users first doing the validation work that should have been done here.*

**Responses:** Here was an opportunity for the CC5 to point us at additional validation material that he would recommend – and yet ...

*In my view there is no scientific merit in this work.*

**Responses:** As stated, this is a 'view' and not a fact. We strongly disagree.

Returning to the reviewer's overview comments.

*"What did this paper tell us that is new?"*

The paper applies the color ratio algorithm to OMPS data which has not been done before. The paper shows the uncertainty in assumptions about the size distribution reflect on the uncertainty in the retrievals.

*"How is the community now smarter?"*

The community is now smarter in that users should apply color ration algorithms cautiously to OMPS measurements especially during extreme events. Indirectly, we show that OMPS wavelengths are insufficient to retrieve both rm and s .

The reviewer clearly recognizes that the two-wavelength retrieval algorithm has a long history and is still widely used in the scientific community because of its robustness and applicability in retrieving aerosol properties from satellite observations.

Our study demonstrates why this well-established algorithm is particularly suited to the OMPS-LP instrument.

Applying this conventional particle size retrieval algorithm to the OMPS-LP is indeed significant. It highlights OMPS-LP's capability to provide valuable information on stratospheric aerosols, which is crucial for monitoring long-term changes in aerosol loading and understanding their impact on climate and atmospheric chemistry.

Moreover, we have conducted a detailed uncertainty analysis that introduces a novel approach to quantify uncertainties in aerosol size retrievals. This analysis considers the uncertainties arising from measurement errors, the assumed distribution width, and phase function assumptions. We believe this contributes new insights into the limitations and potential of aerosol size retrievals from satellite observations.

We are committed to making this dataset available to the community with clear documentation on its limitations and intended applications.

*Brief synopsis of the authors' work*

*Herein, the authors demonstrate how the size (radius, herein referred to as rm) of stratospheric aerosol can be estimated using extinction ratios (or color ratios, herein referred to as CR) using OMPS data. This method is not novel, and the authors appropriately cite the Yue and Deepak (1983) paper. This method is based on finding CR values in a Mie theory lookup table that fall within the expected uncertainties. From there, the authors average the radii of particles that fall within their solution space and return an estimated rm. The authors then compare their estimates with those measured on 2 flights of the University of Wyoming's Optical Particle Counter (OPC) and then apply their method to OMPS data collected after 2 volcanic eruptions: Raikoke and HTHH. The authors also present a sensitivity study to determine the impact of different OMPS viewing geometries on their rm estimates.*

*General Concerns with the Manuscript*

*− This paper requires a thorough proof reading to correct the multitude of grammatical errors, incomplete sentences, and passages that are difficult to interpret.*

*− The authors make claims throughout the manuscript and provide references to bolster these claims. Many of the cited works do not support these claims. Some of these papers have nothing to do with what the authors are claiming. This is particularly egregious because many of these issues were raised in the first round of reviews for their original manuscript, yet they remain unchanged here. Please check all references and confirm they are correct.*

**Responses:** We sincerely apologize for any grammatical errors and unclear passages in the manuscript. We have conducted a thorough proofreading and have corrected all identified issues to ensure clarity and readability throughout the text.

Regarding the references, we acknowledge that some were incorrectly cited in the previous version. We have carefully reviewed and updated all citations to ensure they accurately support our claims. If there are specific references you believe still do not align with the statements made in the manuscript, we would appreciate further guidance so that we can make the necessary corrections.

*− The authors cite other works that demonstrate a fundamental misunderstanding of the cited work. The Wrana et al. 2021 paper is particularly abused within this manuscript. These issues were raised during the review of the original paper and, to date, remain unchanged. This should be corrected before resubmission.*

**Responses:** We appreciate your feedback regarding the use of Wrana et al. (2021) in our manuscript. As we replied above, we cited this work specifically to support our assumption that a distribution width of 1.6 is appropriate under ambient conditions, as indicated in their Figure 6 and corroborated by balloon measurements. For scenarios involving volcanic eruptions, we have referenced Duchamp et al. (2023), which provides relevant information on distribution width under these conditions.

We understand the importance of accurately representing cited works and have carefully reviewed our citations to ensure they are used correctly. If there are specific aspects of Wrana et al. (2021) that you believe have been misinterpreted, we would appreciate further clarification so we can address this concern comprehensively.

*— The authors gloss over glaring issues within the OMPS data product and how these problems will impact their rm estimates. Bourassa et al., 2023 (doi.org10.10292022GL101978) and Gorkavyi et al., 2021 (doi.org10.5194amt-14-7545-2021) discuss the bias in the OMPS extinction product, but the authors never address this issue herein. I have discussed this with Mahesh Kovilakam, who will also submit a review that goes into more detail on this aspect, but this can be easily seen by comparing the global half-to-1-micron ratio of OMPS to that of SAGE.*

*— Previous work (Bourassaetal.,2007oi:10.1029/2006JD008079) demonstrated that Rayleigh optical depths, for scattering instruments, are high enough at lower wavelengths (i.e., 510 nm) that limb scattering observations are insensitive to aerosol signatures. This relates back to the previous comment. Perhaps this is a misunderstanding on my part, but after having numerous conversations about this paper (conversations that involved people outside our working group and outside our organization) I can confidently state that lack of fidelity in the OMPS 510 nm channel is a common perception (a perception that is supported by anal- ysis involving the 510 channel). It would greatly benefit the reader if the authors were to demonstrate that the 510 nm channel is reliable and refute the claims of Bourassa et al. 2007.*

**Responses:** We acknowledge the concerns regarding the quality of the OMPS data product and its potential impact on our retrievals of aerosol properties. While the detailed evaluation of OMPS data biases is beyond the primary scope of our study, we have used the OMPS aerosol extinction coefficient product as provided and focused on deriving aerosol properties based on this dataset.

We understand the significance of addressing these issues, as highlighted by Bourassa et al. (2023) and Gorkavyi et al. (2021) and the sensitivity of the 510 nm channel by Bourassa et al. (2007). In our response to Comment CC8, we have discussed these data quality concerns in more detail.

*Specific comment on this manuscript*

*The authors original statements will be presented in quotation marks (black font), while my comments and questions will be in blue.*

*1.       lines 12–14: "We apply our algorithm to extinction coefficient measurements made by the Stratospheric Aerosol and Gas Experiment on the International Space Station (SAGE III/ISS) to verify our approach and find that our results are in good agreement."*
*Whether you call this a "verification", "validation", or "evaluation", it does not matter be- cause it was not done here. There was no validation/evaluation/verification of the algorithm. The*

*authors used OMPS data to estimate rm and then used a coincident SAGE profile to es- timate rm. This provides nothing more than a roundabout comparison of the two extinction products for a few select profiles (none taken under volcanic aerosol loading, which avoids the issue pointed out by Bourassa et al., 2023). The authors should either perform a meaningful validation of their product or remove this claim from the manuscript.*

**Responses:** The whole section related to SAGE III/ISS is deleted in this revision.

*2.      line 36: Reference to Thomason et al., 2021*

*This is an incorrect citation. This paper did not estimate particle sizes as the authors claim (other than to say the particles were "probably big" or "probably small"). This reference should be removed and all citations should be checked to ensure they are appropriate throughout.*

**Responses:** Deleted.

*3.      line 55: Here the authors provide a brief enumeration of uncertainty sources in their products. Reference to the "arch effect" as discussed in Bourassa et al. 2023 and Gorkavyi et al. 2021 would be appropriate here as would the Rayleigh scattering optical depth issues as raised by Bourassa et al. 2007.*

**Responses:** added.

*4.      line 60: "Our algorithm is similar to the color ratio method developed by Thomason and Vernier's (2013)..."*
*As written, this sounds like Thomason and Vernier 2013 determined particle sizes, which they did not. Please clarify.*

**Responses:** Modified.

*5.      line 75: It seems odd to refer to Q as scattering efficiency here since you are dealing with extinction efficiencies. This seems like a unnecessary detour. Please consider whether this nomenclature is necessary (your call on whether it is changed or not).*

*6.      line 76 and Eq. 1a: The PSD equation is a function of r (array of particle radii over which Eq. 1a is integrated), rm, and σ. Please update for clarity.*

**Responses:** Updated.

*7.      line 76: Why the jump from Eq. 1a to Eq. 3?*

**Responses:** It is better to show the log-normal distribution details in Section 2.2.

*8.      line 77: It would be helpful to the reader to define N, r0, and s here.*

**Responses:** Updated.

*9.      line 80: "For occultation measurements p = 1 and Eq. (1a) is the same as Wrana et al. (2021) Eq. 2."*
*It is unclear what the authors are trying to communicate here. As written, their Eq. 1a is identical to Wrana's Eq. 2, which makes this statement tautological. Please clarify.*

**Responses:** Deleted.

*10.      Line 121 in regards to Fig. 2:*

*Though not critical to the paper, why the discrepancy at -8.21/-78.98? Why would the OMPS retrieval stop >10 km above the cloud top?*

**Responses:** That profile is the closest match to the balloon measurement shown in Figure 9. The retrieval issue you mentioned is outside the scope of this study.

*11.      Lines 146–148: "Wrana et al. (2021) used a third extinction wavelength from SAGE III/ISS data to estimate the log-normal distribution width and found that most of the observations clustered between s = 1.4 and 1.6. This information also constrains our size distribution uncertainty."*

*The authors misinterpreted this paper. Wrana et al. 2021 does not support this assumption. Here, the authors ignore a wealth of data that refutes the assumed range of distribution widths. For example,*

*(a)  the Wyoming OPC record shows a lot more variability especially outside background conditions.*

*(b)  Wrana et al. 2021 (Fig. 6) showed far more variability, between 2017–2019, than the authors claim. From Wrana et al. 2021 "The upper and lower boundaries of the colour bars in Fig. 6 roughly mark the ranges within which the values of the respective quantities fall for the data set between June 2017 and December 2019." It is important to note that their colorbars extend from ≈1.1 to 2.0, which refutes the authors claim.*

*(c)  Wrana et al. 2023 showed more variability than what is claimed here.*

*(d)  Knepp et al., 2024 (https://doi.org/10.5194/amt-17-2025-2024) likewise showed*

*more variability. This product is readily available for download and use.*

*Ultimately, this results in the authors imposing an artificial constraint within their algorithm that preferentially steers their algorithm to a subset of solution spaces.*

*12.      Lines 151–152: "The uncertainty in CR, UCR, can be estimated from Taha et al. (2021, Fig. 6) comparison to SAGE III/ISS." As written, I do not understand what this sentence means. Please revise.*

*13.    Line 156: "Rieger et al. (2018, Fig. 6) which gives a width uncertainty of ≈0.2"*
*The authors misunderstand Fig. 6 of Rieger et al. 2018. This figure tells the reader the range of*
*PSD values used in their simulations and is descriptive, not prescriptive. The authors are not*
*using Rieger's Fig. 6 for its intended purpose and this figure does not support their σ uncertainty.*

*14.    Line 156: "...consistent with the Wrana et al. (2021) estimate."*

*Again, the Wrana et al. 2021 paper does not support this claim.*

**Responses:** For comments 11 to 14, please refer to our earlier response regarding the
distribution width.

*15.    Line 159: I do not understand what is meant by "outside of the averaging domain."*
*Would the authors please clarify?*

**Responses:** We mean Points located outside the averaging domain depicted in Fig. 3 are
excluded.

*16.    Lines 160–161: "...common distribution widths (s=1.4–1.6)."*

*Defining this range of distribution widths as "common" is heretofore unsupported.*

**Responses:** Please refer to the response above for the main comment.

*17.    Lines 163–170: I apologize, but I do not understand what the purpose of this paragraph*
*is or how it fits within the context of the surrounding text. Would the authors please check that*
*this should be here and does not need revised?*

**Responses:** This paragraph provides further explanation of Figs. 3 and 4 and includes an
example to demonstrate the application of the figures.

*18.    Lines 169–170: "This example shows how the radius and number density uncertainty due*
*to the distribution width can be quantified."*
*There is no doubt that uncertainty can be quantified, the fact remains that these values are*
*calculated incorrectly. The range of σ values has been artificially constrained to yield an*
*artificially small uncertainty. If the authors were to use a reasonable range of values then their*
*uncertainty estimates would be so large as to make their estimate of rm meaningless (especially*
*under enhanced aerosol loading).*

**Responses:** We do not agree that the range of σ values has been artificially constrained to
produce a smaller uncertainty. In fact, the range of σ is from 1.1 to 1.8, as shown in Figure
4, which is comparable to the range in Wrana et al. (2021), from 1.1 to 2.

*19.    Line 173: The jump from Fig. 4 to 11 here is confusing. Please number figures and*
*equations sequentially, in the order in which they appear in the text as this will aid the reader.*

**Responses:** Figure 9 is more appropriate in Section 4, where aerosol properties under different conditions are discussed. Mentioning Figure 9 here is optional and simply serves to provide readers with an example to help understand our concept. The current Section 2.2 primarily focuses on evaluating retrieval uncertainties.

*20.     Lines 174–175, in reference to Fig. 4.*

*The content of this figure is misleading. As discussed above, the authors used an artificially small (and unjustified) uncertainty in σ. Second, the authors neglect to account for the bias inherent in the NASA OMPS retrieval algorithm (as discussed by Bourassa et al. 2023). Failing to account for these issues misleads the reader and skews your subsequent analy- sis and conclusions that are based on this figure. Please recreate this figure using correct uncertainties and accounting for bias under elevated aerosol loads.*

**Responses:** Please refer to the response above for the comment 18.

*21.     Section 2.3*

*Here, the authors fail to account (or mention) the bias inherent in their retrieval as discussed by Bourassa et al. 2023. If the authors want this to be applicable to volcanic conditions, then this issue must be addressed.*

*22.     Lines 202–207 Here, the authors seem to conflate bias and measurement uncertainty. What they show in Fig. 6(c) is a bias in their CR (an offset) that is caused by assuming an incorrect size distribution. Because this is an offset it cannot be used in place of an uncertainty in their error propagation. Please correct.*

*23.     Line 206: "...we get an uncertainty range of 0.15–0.3 μm"*
*What is the range if the authors correctly calculate their errors (i.e., using realistic values of σ)?*

**Responses:** For comments 21 to 23, please refer to the response above for the main comment.

*24.     Section 3.1*

*This section is unnecessary. The authors state "The Fig. 7 comparisons verify our assertion that errors due to aerosol phase function variation with radius are minor (see Section 2.3), and that the extinction coefficient estimates from the OMPS-LP L2 algorithm are robust." However, they previously stated that previous studies have evaluated the OMPS/SAGE extinction products so it is difficult for me to see the value of performing yet another com- parison, much less one that is based on derived products (products that involve more as- sumptions and an additional level of abstraction). The purpose of this paper is to introduce a new method for inferring particle sizes from OMPS extinction coefficients and this section does not fit within the scope of that purpose. This in no way provides a validation of their method and cannot, under any condition, be interpreted as a validation or even a meaningful evaluation.*

*Despite how good Fig. 8 may look, it is ultimately meaningless because, again, you are not comparing 2 independent estimates of rm, you are effectively comparing the 2 extinction products in a very roundabout way. As is, this provides no validation/evaluation and, for the purpose of this paper, is meaningless.*

*25.     Line 244: "The Fig. 7 comparisons verify our assertion that errors due to aerosol phase function variation with radius are minor"*
*This presupposes that your rm estimates are correct, which has yet to be demonstrated. All agreement shown in this figure (or disagreement) is strictly due to the similarity in the 2 extinction products and in no way validates your rm estimate.*

**Responses:** For comments 24 and 25, the whole Section 3.1 is deleted.

*26.     Lines 265–266: "...uncertainty ranges of OMPS-LP retrievals are calculated from the uncer- tainty in the color ratio extinction coefficients..."*
*As written, it seems that the authors did not include an uncertainty factor for the distribu- tion width (please clarify). If they did not, then the uncertainty bounds in this figure are misleading and I ask that they include a full accounting of uncertainty.*

*I also read in the OMPS v2.1 readme "Loss of sensitivity of short wavelengths radiances to aerosols. This effect is caused by Rayleigh and aerosol attenuation of the limb scattered radiation, and becomes most pronounced below ≈17 km and in the southern hemisphere. We advise caution in using LP aerosol extinction data below 17 km and scattering angle greater than 145 degrees for wavelengths 675 nm or shorter. The error bars provided in the daily product data files do not include this term. This error can be reduced by using 745, 869, and 997 nm wavelengths." My concern is with this sentence: "The error bars provided in the daily product data files do not include this term." Given this statement, are CR-error estimates, which are then used to calculate the overall rm error, correct?*

**Responses:** The uncertainty in distribution width is not included in this figure, nor should it be, as the figures are meant to compare which distribution width values best fit the balloon measurements.

Regarding wavelength sensitivity, both figures focus on the Northern Hemisphere and do not discuss altitudes below 16 km.

*27.     Lines 266–267: "To account for uncertainty in the assumed width we show the results for widths varying from s=1.2 to 1.8"*
*This demonstrates the sensitivity of your size estimate on the distribution width, which leads to the question of "Which distribution width is correct?" This figure alone provides irrefutable evidence that using a static distribution width is incorrect. This is observed by the disagreement between the 2 profiles (it does not matter what panel you look at) and by plotting the OPC-derived σ profile for this flight. The OPC data show variation in σ that ranged from 1.18 to 2.45*

*in the stratosphere. It is physically impossible to generate a meaningful rm estimate using a single σ value throughout the profile.*

*As mentioned above, the authors have neglected the wealth of information on this topic and arbitrarily selected a value of 1.6 and incorrectly cited Wrana et al. 2021 in support of the 1.6 value, but failed to recognize that Wrana et al. showed (e.g., in their Fig. 6) that the distribution width has a broad range.*

*There is additional information available from the Wyoming OPC record (see my figure above) that can be useful in setting these boundaries. Recent work by our group shows that distribution width is generally between 1.2 and 1.9 (Knepp et al. 2024 https://doi. org/10.5194/amt-17-2025-2024), but this is highly dependent on where and when the measurements are made.*

**Responses:** These figures are used to validate our assumption about distribution width. The distribution widths beyond the displayed range perform even worse compared to the measurements, so there is no need to include them. Additionally, a dynamic distribution width is not the focus of this study.

*28.    Line 270: "The comparisons in Fig. 9, 10 show the best agreement is for s=1.6. This is consistent with analysis of Wrana et al (2021, 2023)"*
*Neither of these papers support this conclusion.*

**Responses:** Please refer to the response above for the main comment.

*29.    Line 271: "...where the particle distributions are unimodal..."*

*Is this statement made strictly in regard to Fig. 9 & 10 or is this a more general statement. Looking at the Wyoming OPC record I see numerous examples of bimodal distributions well past 25 km. If we limit the analysis to the 2 profiles you showed here then we can fool ourselves into thinking the situation is better than it is (i.e., your analysis indicates that your algorithm performs better when the aerosol distributions are unimodal, but bimodal distributions are ubiquitous). It would greatly benefit the reader to expand this analysis to include more profiles that contain persistent bimodal distributions.*

**Responses:** Thank you for providing this information. The bimodal distribution will be part of our next investigation. Note that Knepp et al. (2024) suggests that is very difficult to retrieve a bimodal distribution using SAGE data.

*30.    Lines 171–172: "Both the OMPS-LP and balloon data particle radius are ≈0.1 μm at all altitudes..."*

*This is categorically false and is refuted by the authors' own figures. A cursory look at the figures shows an overall range of 0.05–0.125 μm in Fig. 9 and a range of 0.05–0.3 μm in Fig. 10.*

**Responses:** We mean that most values are around 0.1 µm, but that sentence has been deleted.

*31.     Lines 272–273: "The particle radius decreases with increasing altitude for both OMPS-LP retrievals and in the balloon data above 20 km."*
*This is not correct. The OPC rm value increased rapidly above 20 km and held steady until ≈ 21 km where it began to decrease.*

**Responses:** We changed the 20 km to 21 km.

*32.     Line 278: What is meant by "strongly bimodal"? Quantifying this would greatly aid the reader. Please quantify to remove ambiguity.*

**Responses:** We modified the sentence.

*33.     Lines 281–282: "The reasonable agreement using s=1.6 shows that this particle width works under both ambient and the aged volcanic plume conditions."*
*This is categorically false and I fail to see how the authors can make this claim. As is obvious from Fig. 10 this did not work well between 16–20 km where the 2 products differed by up to a factor of 6. Even at 18 km the disagreement remained a factor of 3. This is not good agreement. Please correct this statement.*

**Responses:** There is a bimodal distribution between 16 and 20 km, so it cannot simply be described as a factor of 6. Additionally, among the various distribution widths, s=1.6 shows the closest fit to the measurements.

*34.     Line 293: "...background -90° to 90° in Fig. 11a, b..." For this plot, what altitudes were plotted?*

**Responses:** Measurement distributions during different observing conditions from the tropopause to 35 km.

*35.     Line 293: "...background -90° to 90° in Fig. 11a, b..."*
*It is difficult to understand the classification of this time period at "background" immediately after the La Soufriere eruption and when the atmosphere was still recovering from the ANY event (granted, much of that event already dissipated, but we still see traces in the SAGE data). Please clarify this designation*

**Responses:** It is difficult to find any purely quiescent – there will always be some small uncertainty. Below is from GloSSAC and shows that there are constant aerosol perturbations in the stratosphere. Below is the GloSSAC SAOD with the periods where the analysis takes place.  While not perfect, these regions are reasonably representative of background.

[Figure]

Figure 2. The time series of GloSSAC SAOD with the periods where the analysis takes place.

*36.      Line 331: While listing uncertainty sources the reader should be reminded of the issues in the OMPS product during elevated aerosol loading (per Bourassa et al. 2007, 2023).*

**Responses:**  Bourassa et al. (2023) introduced a tomographic retrieval technique using multiple profiles to remove or reduce the 'arch effect' where the standard OMPS algorithm extends an isolated aerosol anomaly downward at the edges increasing the total aerosol optical depth.  This can create an anomalously high SAOD. Bourassa suggested this explains that difference between SAGE and OMPS-LP retrievals.  The correction introduced by the Saskatchewan team reduced the SAOD to values consistent with SAGE early in the Hunga eruption period.  However, the differences between SAGE and OMPS persist even after the eruption aerosol has dispersed - when the 'edges' have disappeared. The present consensus is that the differences between NASA OMPS and USask OMPS are due to assumptions about the size distribution.  This explanation is more in line with our results showing the sensitivity retrieved aerosol properties to the width of the distribution.  As a result we feel no need to bring this issue up until it is better resolved.

*37.      Lines 332–334: "The impact of the distribution width is limited. Especially, Fig. 10c suggest that = 1.6 is a good approximation for the later stage of aerosol evolution."*
*This is wrong. Even a cursory look at these figures shows the distribution width plays a significant role in the size estimate. Contrary to the authors' claim, Figs. 9 & 10 demonstrate the importance of correctly selecting the distribution width. Further, the OMPS-derived rm in Fig.*

*10 (c) is 3–6 times larger than the OPC value and the OMPS profile shows a distinctly different structure from the OPC. I am sorry to say, but this difference is not inconsequential:*

*we cannot declare this a "good approximation". Can the authors please explain to the reader how being off by a factor of 3–6 can be viewed as good?*
*Please note that this poor behavior is a direct consequence of the static distribution width. What this demonstrates is how a constant σ forces your algorithm toward a particular set of rm solutions. This is a clear demonstration of the inadequacy of this assumption.*

**Responses:** Please referring to the comment 33.

*38.    Line 347: The reference to Taha et al. 2022 is not appropriate since that paper did not determine the aerosol composition.*

**Responses:** Deleted.

*39.    Line 364: "The median radius grows through day 30-80. The 0.4 μm peak in the median radius..."*

*It is not clear to me how this is possible since the authors imposed a CR ratio cutoff (1.1) that, per their description, limited you to particles less than 0.3 μm?*

**Responses:** The limitation due to the color ratio cutoff has been corrected to 0.4 μm in the text

*40.    Line 398: "We verify our algorithm using SAGE III/ISS..."*

*As discussed above this exercise in no way evaluated the performance of your rm algorithm. This only tested the agreement between the 2 extinction ratios and, knowing how well the 2 ratios agree. This is a meaningless "verification" since the result could have been guessed before the analysis began (i.e., if the 2 CRs are close then the inferred rm will be close; it proves nothing about the algorithm).*

**Responses:** Deleted.

*41.    Line 400: "...we validate our retrieved aerosol median radius..."*

*Claiming your product is "validated" by comparing to 2 OPC profiles (one of which the agreement was quite poor) is misleading to the readers. Please revise to communicate this accurately.*

**Responses:** We responded to this comment above.

*42.    Line 403: "There are three major sources of uncertainty in our radius calculation..."*

*Per Bourassa et al. 2007, 2023 and Gorkavyi et al. 2021 the authors should include the bias in their extinction product under elevated aerosol loading as a source of uncertainty/error.*

**Responses:** See response on line 36.

*43.      Lines 408–409: "However, the good agreement between our retrievals and in situ observations suggests a width of 1.6 is a reasonable value under both ambient conditions and the Raikoke volcanic eruption."*
*I realize I have hit this point throughout the paper, but since the authors make this claim throughout the paper it seems they are genuinely unaware of how wrong this is. An assumed static distribution width of 1.6 is wrong and should not be done. The authors should know this from their Fig. 10, from Wrana et al. 2021, Wrana et al. 2023, Knepp et al. 2024, and the entire OPC record. This single assumption introduces a massive source of error and directs the algorithm to a predetermined subset of their solution space (this is clearly seen in Fig. 10).*

*I will note that assuming a value of 1.6 is a reasonable assumption during background periods, though you must still properly account for natural variation, which the authors have yet to do. However, background conditions are not particularly interesting. Further, if we assume σ =1.6 as a "valid" assumption for distribution width then why not assume a mode radius of 85 nm? During background conditions the rm=85 nm assumption is just as valid as the σ =1.6 assumption. Outside background conditions, both assumptions fail and effectively break the utility of the proposed algorithm.*

*44. Figure 10: I already spoke at length on Figure 10 and I believe this figure provides incontro-vertible evidence that the static σ assumption is wrong. Please consider plotting the OPC σ value as well. Doing so will demonstrate 2 things: 1. a static σ value is incorrect, 2. it will show you why your algorithm fails at reproducing the OPC profile (i.e., the information content of the OMPS data is too low and your estimate is being driven predominantly by the changing σ).*

*I thank the authors for taking the time to read my comments and look forward to their response.*

**Responses:** For the discussion regarding static distribution width in comments 43 and 44, please refer to the response above for the main comment.

---

## Author Comment (AC7)

**Responses to one of the community reviewer's comments (CC8)**

The following are our responses to the community reviewer Mahesh Kovilakam's comments (CC8). His comments are copied and pasted here in italics. Since his comments are written as several paragraphs, not numbering, and primarily address one issue, we provided a response after his entire comment rather than using a point-by-point format.

*The authors present a methodology for deriving particle median radius and concentration from OMPS-LP color ratio (CR). It's worth noting that this paper appears to be a resubmission of a previous work submitted last year (Wang et al., 2023), with the new version containing somewhat similar content (Wang et al., 2024). The core concept of CR remains consistent with the latest submission (Wang et al., 2024). Figure 1 mirrors the previous version submitted (Wang et al., 2023), albeit with the axes swapped. While utilizing CR seems a reasonable strategy for size distribution retrieval, it's essential to recognize that the OMPS-LP aerosol extinction coefficients at 510, 600, 675, 745, 869, and 997 nm are initially derived under the assumption of a constant Gamma distribution (Taha et al., 2021; Chen et al., 2018) . Consequently, these extinction coefficients primarily reflect background stratospheric aerosols, and using them for number density and concentration retrieval during volcanic eruptions lacks robustness.*

*Furthermore, a recent study (Bourassa et al., 2023) highlighted that the OMPS-LP NASA (re-ferred to as OMPS(NASA)) extinction coefficient product exhibits a twofold increase compared to the OMPS extinction retrieved by the University of Saskatchewan (OMPS(USASK)) and SAGE III/ISS following the Tonga eruption. Notably, this study (Wang et al., 2024) employs the same OMPS(NASA) product to retrieve median radius and concentration, potentially impacting the de- rived quantities significantly. A thorough examination of OMPS(NASA) extinction coefficients at 510, 745, 869, and 997 nm, compared with alternative stratospheric extinction products, reveals a notable overestimation by the OMPS(NASA) product. Major concerns regarding the paper are outlined below:*

*Major comments*

*• Similar to Wang et al. (2023), the authors discuss CR (510/869 nm) and CR (745/869 nm). It appears that the authors utilized CR of 510/869 nm for the retrieval, but the rationale behind showcasing CR for 745/869 nm remains unclear. I am uncertain about the utility of CR of 745/869 nm, given the proximity of these two channels. The crucial point here is the significant bias observed in the 510 nm channel, as evidenced by available SAGE III/ISS mea- surements at a wavelength of 521 nm, close to 510 nm. Evaluation of the extinction product reveals that the OMPS(NASA) product at the 510 nm channel is unusable due to bias/noise in the data. Therefore, employing the extinction coefficient at 510 nm from OMPS(NASA) itself is not the appropriate approach to compute CR. We conducted an analysis of extinction coefficient data at 510, 745, 864, and 997 nm of OMPS with 521, 756, 869, and 1022 nm of SAGE III/ISS. Figure 1 depicts the zonally averaged stratospheric aerosol optical depth (SAOD) time series percent difference plot of OMPS(NASA) and SAGE III/ISS. It is evi- dent from Figure 1 that the 510 nm channel of OMPS(NASA) exhibits a significant bias, irrespective of volcanic events. In addition to the substantial bias in the 510 nm channel, it is noteworthy that significant overestimation of*

SAOD is apparent in Figure 1 following the Tonga eruption in all four channels. This corroborates the findings of Bourassa et al. (2023) that OMPS(NASA) overestimates the extinction coefficient at 745 nm by a factor of two following the Tonga eruption. Therefore, users of OMPS(NASA) must exercise caution when using the OMPS(NASA) extinction coefficient product, particularly following volcanic eruptions. Moreover, the use of the 510 nm channel is questionable, especially given the large bias in Figure 1a, regardless of any events. Additionally, other OMPS retrieval products from OMPS(USASK) and IUP Bremen (hereafter, OMPS(IUP)) are available to the public. It is worth noting that, in addition to the overestimation of aerosol extinction coefficient in OMPS(NASA) following the Tonga eruption, similar differences were noted following the Ke- lud (13 February 2014) and Calbuco (28 April 2015) eruptions at 745 (864) nm wavelengths when compared against OMPS(USASK) (OMPS(IUP)) products (not shown here). There- fore, following any events that perturb the stratosphere, OMPS(NASA) extinction coefficient product should be used with caution.

[Figure]

Figure 1: Latitude versus time dependence of zonally averaged stratospheric aerosol optical depth (SAOD) percent difference between OMPS(NASA) and SAGE III/ISS at (a) 510 , (b) 745 , (c) 864, and (d) 997 nm. For SAGE III/ISS the respective wavelengths used for computing percent differences are 521, 756, 869 and 1022 nm. Major volcanic eruptions (white) and wild fire events (green) with abbreviated two-letter code with their respective latitude and time of occurrence that are listed here. The event names shown are Canadian wildfire (Cw), Ambae (Am), Raikoke (Rk), Ulawun (Ul), Australian wildfire (Aw), California Creek Fire (Cc), La Soufriere (La), McKay Creek fire (Mc) and Hunga Tonga (Ht).

The authors employ the 510 nm channel to compute CR, leading to a biased CR that in- evitably affects the retrieval of median radius and concentration. It's also worth noting that when SAOD is computed between the tropopause and about 21 km, the disparity between OMPS(NASA)

and SAGE III/ISS is even more pronounced following the Tonga eruption. This discrepancy arises from the fact that OMPS(NASA) tends to overestimate extinction below the peak of the aerosol layer, as noted by Bourassa et al. (2023), as illustrated in Figure 2c,d. Additionally, OMPS(NASA) release notes (https://disc.gsfc.nasa.gov/datasets/OMPS_ NPP_LP_L2_AER_DAILY_2/summary?keywords=OMPS-NPP_LP-L2-AER) state that low sensitiv- ity of short wavelengths to aerosols may impact retrievals below 675 nm, advising caution in using LP aerosol extinction data below 17 km and scattering angles greater than 145 degrees for wavelengths 675 nm or shorter.

While OMPS(NASA) extinction coefficients show improvement towards higher wavelengths compared to the 510 nm channel, it's important to highlight that the OMPS(NASA) extinction coefficients at 745, 869, and 997 nm significantly overestimate following the Tonga eruption and other events. Hence, the robustness of the OMPS(NASA) product, particularly following volcanic eruptions, is questionable.

Furthermore, we evaluated the OMPS(NASA) extinction product at 510 and 997 nm for a relatively unperturbed stratosphere (June 2017) and following the Tonga eruption (April 2024) (Figure 2) using GloSSAC version 2.21 data. The percent difference between OMPS(NASA) and GloSSAC clearly indicates a significant disparity at the 510 nm channel (> 50%) below 24 km, even in June 2017 (Figure 2a). This underscores the bias in the 510 nm channel regardless of any perturbed event. However, the 997 nm channel appears reasonable during June 2017 (Figure 2b). However, following the Tonga eruption, both the 510 and 997 nm channels exhibit significant differences compared to GloSSAC (Figure 2)c,d.

[Figure]

Figure 2: Zonally monthly averaged percentage difference of OMPS(NASA) and GloSSAC (version 2.21) for 525 and 1020 nm on an altitude versus latitude plot. (a, b) for June 2017 and (c, d) for April 2022 following Hunga Tonga eruption.

• *Since the 510 nm channel exhibits a significant bias, any CR computation involving this channel will inevitably yield incorrect ratios. Figures 3a and 3b illustrate the percent difference of SAOD between OMPS(NASA) and GloSSAC at 525 nm and 1020 nm, respectively (note that we utilized SAOD at 510 nm and 997 nm to calculate the percent difference with GloSSAC). It's apparent from Figure 3a that the OMPS 510 nm channel consistently exhibits a high bias, regardless of any perturbed event. Therefore, employing the 510 nm channel to retrieve particle size-related quantities would introduce bias and yield unreasonable results. While the bias in the 510 nm channel persists throughout the record, Figure 3b shows some improvement when comparing the 997 nm channel against GloSSAC's 1020 nm channel. However, there is an overestimation of OMPS(NASA) SAOD following perturbed events such as the Canadian Wildfire, Ambae, Australian Wildfire, and the Tonga eruption.*

[Figure]

*Figure 3: Latitude versus time dependence of zonally averaged stratospheric aerosol optical depth (SAOD) percent differences from OMPS(NASA) and GloSSAC (v 2.21) (a,b) and SAOD ratios (c,d). OMPS(NASA) SAODs are computed for 510 and 997 nm, while for GloSSAC SAODs are computed at 525 and 1020 nm. Ratio between 510 and 997 nm of OMPS(NASA) AOD is shown in (c), while (d) shows ratio between 525 and 1020 nm of GloSSAC 2.21 for the same time period. Major volcanic eruptions (white) and wild fire events (green) are same as in Figure 1.*

*We also evaluated the SAOD ratio between the 510 nm and 997 nm channels, as well as between the 525 nm and 1020 nm channels of GloSSAC. This comparison offers insights into*

*how two wavelengths can inform about aerosol extinction coefficient ratios, thereby aiding in the inference of aerosol particle sizes. It's essential to note that we focused on the time series from 2017 through 2022 due to the availability of SAGE III/ISS multi-wavelength measure- ments from 2017. Figure 3c clearly demonstrates that the OMPS(NASA) SAOD ratios do not provide meaningful information, indicating that these wavelengths cannot be utilized to infer size information. Notably, most of the bias stems from the 510 nm channel. In contrast, the aerosol SAOD ratio from GloSSAC offers valuable insights (Figure 3d) into how ratios change following each volcanic/fire event, particularly after events like the Canadian Wildfire, Am- bae, Ulawun, Raikoke, Australian Wildfire, and Hunga Tonga. Each event behaves differently in terms of the aerosol SAOD ratio. For instance, the Canadian wildfire, Raikoke, Australian wildfire, and Hunga Tonga show a smaller SAOD ratio, suggesting relatively larger particles, while Ambae and Ulawun eruptions exhibit a larger SAOD ratio, suggesting smaller particles.*

*Consequently, utilizing the OMPS(NASA) extinction product to retrieve size-related infor- mation would introduce bias and may not accurately represent the underlying aerosol size information.*

**References**

*Bourassa, A. E., Zawada, D. J., Rieger, L. A., Warnock, T. W., Toohey, M., and Degenstein, D. A.: Tomographic Retrievals of Hunga Tonga-Hunga Ha'apai Volcanic Aerosol, Geophysical Research Letters, 50, e2022GL101 978, https://doi.org/https://doi.org/10. 1029/2022GL101978, URL https://agupubs.onlinelibrary.wiley.com/doi/abs/10.1029/ 2022GL101978, e2022GL101978 2022GL101978, 2023.*

*Chen, Z., Bhartia, P. K., Loughman, R., Colarco, P., and DeLand, M.: Improvement of strato- spheric aerosol extinction retrieval from OMPS/LP using a new aerosol model, Atmospheric Mea- surement Techniques, 11, 6495–6509, https://doi.org/10.5194/amt-11-6495-2018, URL https: //amt.copernicus.org/articles/11/6495/2018/, 2018.*

*Taha, G., Loughman, R., Zhu, T., Thomason, L., Kar, J., Rieger, L., and Bourassa, A.: OMPS LP Version 2.0 multi-wavelength aerosol extinction coefficient retrieval algorithm, At- mospheric Measurement Techniques, 14, 1015–1036, https://doi.org/10.5194/amt-14-1015-2021, URL https://amt.copernicus.org/articles/14/1015/2021/, 2021.*

*Wang, Y., Schoeberl, M., Taha, G., Zawada, D., and Bourassa, A.: Using OMPS-LP color ratio to extract stratospheric aerosol particle size and concentration with application to vol- canic eruptions, Atmospheric Measurement Techniques Discussions, 2023, 1–25, https://doi.org/ 10.5194/amt-2023-36, URL https://amt.copernicus.org/preprints/amt-2023-36/, 2023.*

*Wang, Y., Schoeberl, M., and Taha, G.: Using OMPS-LP color ratio to extract stratospheric aerosol particle median radius and concentration with application to two volcanic eruptions, Atmospheric Measurement Techniques Discussions, 2024, 1–39, https://doi.org/10.5194/amt-2023-267, URL https://amt.copernicus.org/preprints/amt-2023-267/, 2024.*

**Response:** The gist of the CC8 comment is that the 510 nm channel is not very accurate compared to SAGE III (figures were attached) and CC5 makes a similar point. The color scale in the CC8 figures seems to indicate that all the channels compare poorly to SAGE with errors between 20-50 %.  Taha et al. (2021) Fig. 6 also shows the relative bias between SAGE and OMPS extinction which indicates a dependence in altitude as well as latitude. In

the Taha figure, the agreement with SAGE is within ±25% above 18 km and over most of the latitude range. The 510 channel performs about the same as the other short wavelength OMPS channels.  Agreement with GloSSAC is better for channels at 745 nm and longer. CC8 shows SAOD plots of the ratios of 510/997 and GloSSAC 535/1020.

First, in our paper, we used the aerosol extinction rather than the reported SAOD. Furthermore, we restricted our analysis to above 20 km in the southern hemisphere (SH) tropics (Figure 14), as recommended by Taha et al. (2021).  The SAOD ratio in Figure 1 (from CC8) is incorrect in the southern hemisphere and the tropics because the 510 nm useful altitude range should be restricted to above 20 km (see Figure 2).  The OMPS-LP 869nm is generally accurate for all stratospheric altitudes and latitudes.

[Figure]

Figure 1. A copy of Figure 3 (c) and (d) from CC8.

The argument CC8 makes is that the two SAODs (Fig. 1) are unlike each other therefore OMPS SAOD generated by CC8 is incorrect.  Note that we do not use the wavelengths shown in CC8's figure.

Below is a redo of the above plot using our own color scale and our wavelengths, and restricting the plot to 0°-60° N to avoid the anomalous 510 nm retrieval in the SH below 20 km. OMPS SAOD is computed by integrating from 2 km above the tropopause (to avoid clouds) to 40 km.  The SAOD color ratio between the OMPS and GloSSAC data sets is now more consistent except at high latitudes where OMPS appears to have lower values. This may be caused by the limited coverage of SAGE III at these latitudes compared to OMPS.  If we use the wavelength ratio shown in CC8 (510/997) our results closer to the reviewer's. Notice that the SAOD shown here is only for comparison. For accurate color ratio, it is recommended to use the extinction coefficient at altitude ranges deemed accurate by (Taha et al., 2021) as we did in our paper.

The reviewer's point is that the paper should have more discussion of uncertainties in the OMPS extinction, and we concede the point.  In the paper revisions, we add more discussion on the differences.

[Figure]

Figure 2. The figures are analogous to Figure 3 (c) and (d) from CC8, but with the extinction coefficient used to compute the color ratio, and the wavelengths adjusted to match those in our study. The color bar scale has also been updated accordingly.